# Continuous-time Riemannian SGD and SVRG Flows on Wasserstein Probabilistic Space

**Mingyang Yi**[1]*, **Bohan Wang**[2]
[1] Renmin University of China
[2] Alibaba Group
yimingyang@ruc.edu.cn
bhwangfy@gmail.com

## Abstract

Recently, optimization on the Riemannian manifold have provided valuable insights to the optimization community. In this regard, extending these methods to to the Wasserstein space is of particular interest, since optimization on Wasserstein space is closely connected to practical sampling processes. Generally, the standard (continuous) optimization method on Wasserstein space is Riemannian gradient flow (i.e., Langevin dynamics when minimizing KL divergence). In this paper, we aim to enrich the family of continuous optimization methods in the Wasserstein space, by extending the gradient flow on it into the stochastic gradient descent (SGD) flow and stochastic variance reduction gradient (SVRG) flow. By leveraging the property of Wasserstein space, we construct stochastic differential equations (SDEs) to approximate the corresponding discrete Euclidean dynamics of the desired Riemannian stochastic methods. Then, we obtain the flows in Wasserstein space by Fokker-Planck equation. Finally, we establish convergence rates of the proposed stochastic flows, which align with those known in the Euclidean setting.

## 1 Introduction

As a valuable extension of Euclidean space optimization, generalizing the parameter space to a Riemannian manifold—such as a matrix manifold [1] or a probability measure space [10]—has greatly enriched the toolbox of the optimization community. Technically, optimization on a manifold can be derived from Euclidean techniques by defining corresponding Riemannian gradients and transport rules [1]. For example, several widely used gradient-based optimization methods have been generalized to this setting, including gradient descent (GD) [53], stochastic gradient descent (SGD) [4], and stochastic variance reduced gradient (SVRG) [52].

On the other hand, optimization over probability measure spaces has attracted considerable attention due to its connection to sampling processes [17]. Interestingly, for a specific type of probability measure space—the second-order Wasserstein space—techniques from Riemannian optimization can be directly applied [9], owing to its manifold-like geometric structure. For example, minimizing the KL divergence via Riemannian gradient flow is equivalent to employing Langevin diffusion [35], a standard method for sampling from a target distribution. Therefore, advancing Riemannian optimization in the Wasserstein space may lead to the development of novel sampling techniques.

To this end, we focus on Riemannian SGD [25, 20] and SVRG [33] in Wasserstein space, which are standard stochastic optimization methods known for their lower computational complexity in Euclidean settings [5]. Specifically, we investigate the continuous counterparts (flows) of these two previously unexplored Riemannian methods, ***as continuous formulations provide a powerful***

---

*Corresponding to: Mingyang Yi and Bohan Wang

39th Conference on Neural Information Processing Systems (NeurIPS 2025).

*framework for analyzing the properties of optimization algorithms* [13, 14]. In Euclidean space, such continuous stochastic flows are characterized by SDEs [25, 20, 33]. However, these SDEs do not readily generalize to Riemannian manifolds, as their definitions rely on Brownian motion—a concept that is considerably more complex to define on Riemannian manifolds [38].

In general, continuous optimization methods are derived by taking the limit of the step size in corresponding discrete optimization dynamics (e.g., transitioning from GD to gradient flow [40]). Naturally, we seek to apply this approach to discrete Riemannian SGD and SVRG [4, 52]. Unfortunately, such an extension is nontrivial for two reasons: (1) the linear structure that underpins the aforementioned continuous dynamics does not necessarily exist on manifolds; (2) describing the randomness inherent in stochastic methods is challenging in a manifold setting. Fortunately, the dynamics of probability measures in Wasserstein space can be equivalently described by the dynamics of random vectors in Euclidean space. By taking the step size to zero, the corresponding discrete dynamics in Euclidean space converge to an SDE, which characterizes the evolution of probability measures in Wasserstein space through the Fokker-Planck (F-P) equation [32].

Through this approach, we successfully establish continuous Riemannian SGD and SVRG flows in the Wasserstein space for minimizing the KL divergence as intended. Notably, the existing MCMC methods—stochastic gradient Langevin dynamics [49] and stochastic variance reduction Langevin dynamics [56, 7]—are precisely the discrete counterparts of the two proposed flows. Furthermore, under appropriate regularity conditions, we prove convergence rates for these continuous stochastic flows. Specifically, for non-convex problems, the convergence rates (measured by the first-order stationarity criterion [6]) of the Riemannian SGD flow and Riemannian SVRG flow are $\mathcal{O}(1/\sqrt{T})$ and $\mathcal{O}(N^{2/3}/T)$, respectively, where $N$ denotes the number of component functions. Additionally, under an extra Riemannian Polyak-Łojasiewicz (PL) inequality [23, 11]—equivalent to the log-Sobolev inequality [44]—the two methods achieve global convergence rates of orders $\mathcal{O}(1/T)$ and $\mathcal{O}(e^{-\gamma T/N^{2/3}})$, respectively, matching their Euclidean counterparts [33].

## 2    Related Work

**Riemannian Optimization.**    Unlike continuous methods on Riemannian manifolds [1], discrete methods have been extensively studied. Examples include Riemannian GD [1, 6, 53], Riemannian Nesterov-type methods [54, 28, 26], Riemannian SGD [4, 39], Riemannian SVRG [52], and some other Algorithms [3, 55, 50, 12]. As established in the literature, the standard approach in Euclidean space for linking discrete dynamics to their continuous counterparts involves taking the limit to the step size in the discrete dynamics, which yields a differential equation [43, 40, 27]. However, this derivation does not extend directly to arbitrary Riemannian manifolds. Such extrapolation has only been achieved in specific spaces, such as the Wasserstein space, where the resulting curve satisfies the Fokker–Planck equation [40], thereby establishing a connection between the Wasserstein and Euclidean spaces. Nevertheless, only a limited number of discrete optimization methods in the Wasserstein space have been generalized to their continuous counterparts—for example, gradient flow [11], Nesterov accelerated flow [48], and Newton flow [47]. Unfortunately, these extrapolation techniques have not been applied to stochastic dynamics. Moreover, their methodologies are restricted to specific algorithms, in contrast to the more general framework proposed in this paper.

**Stochastic Sampling.**    Standard sampling methods, such as MCMC [22], typically construct (stochastic) dynamics that converge to the target distribution, with convergence measured by a probability distance or divergence. Consequently, sampling can be viewed as an optimization problem in the space of probability measures [10]. However, existing literature has primarily focused on discrete Langevin dynamics [35] and their stochastic variants [49, 15, 56, 7, 57, 24], particularly analyzing their convergence rates under different criteria—such as KL divergence [8], R'enyi divergence [11, 2, 30], or Wasserstein distance [16]. By contrast, exploring their connections with continuous Riemannian optimization methods, as undertaken in this paper, has received relatively little attention.

## 3    Preliminaries

supported on $\mathcal{X} = \mathbb{R}^d$, where $\mathcal{P}$ is the Wasserstein metric space endowed with the second-order Wasserstein distance [45] (abbreviated as Wasserstein distance), defined as $\mathsf{W}_2^2(\pi, \mu) =$

$\inf_{\Pi \in \Gamma(\pi,\mu)} \int \|\boldsymbol{x} - \boldsymbol{y}\|^2 d\Pi(\boldsymbol{x}, \boldsymbol{y})$, where $\Gamma(\pi, \mu)$ denotes the set of joint probability measures with marginals $\pi$ and $\mu$, respectively. Notably, the Wasserstein space possesses a geometric structure analogous to that of a Riemannian manifold [9], allowing us to apply Riemannian optimization techniques. We now introduce several key definitions. As in $\mathbb{R}^d$, the Wasserstein space (similar to Riemannian manifold) is equipped with an "inner product", resulting the Riemannian metric $\langle \cdot, \cdot \rangle_\pi : \mathcal{T}_\pi \mathcal{P} \times \mathcal{T}_\pi \mathcal{P} \to \mathbb{R}$. Here, $\mathcal{T}_\pi \mathcal{P}$ is the tangent space of $\mathcal{P}$ at $\pi$ (see [9], Section 1.3). Using this Riemannian metric, we define the Riemannian (Wasserstein) gradient $\mathrm{grad} F(\pi) \in \mathcal{T}_\pi \mathcal{P}$ of a function $F : \mathcal{P} \to \mathbb{R}$ as the unique element satisfying $\lim_{t \to 0} \frac{F(\pi_t) - F(\pi_0)}{t} = \langle \mathrm{grad} F(\pi_0), \boldsymbol{v}_0 \rangle_{\pi_0}$, for every curve $\pi_t$ in $\mathcal{P}$ with tangent vector $\boldsymbol{v}_0 \in \mathcal{T}_{\pi_0}$ on $\pi_0$. The transportation of $\pi_t \in \mathcal{P}$ in $\mathcal{T}_\pi \mathcal{P}$ is determined by the exponential map $\mathrm{Exp}_\pi : \mathcal{T}_\pi \mathcal{P} \to \mathcal{P}$. Then, the discrete Riemannian GD $\{\pi_n\}$ with learning rate $\eta$ is defined as

$$\pi_{n+1} = \mathrm{Exp}_{\pi_n}[-\eta \mathrm{grad} F(\pi_n)], \tag{1}$$

to minimize $F(\pi)$. We refer readers for more details about this dynamics to [1, 6, 53, 9]. An illustration of Riemannian GD is in Figure 1.

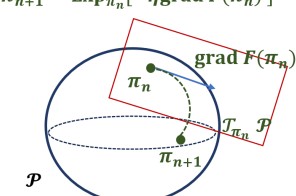

Next, we instantiate these definitions in the context of the Wasserstein space. First, a curve $\pi_t$ in $\mathcal{P}$ is characterized by the continuity equation (also known as the Fokker–Planck equation) [32]:

$$\frac{\partial \pi_t}{\partial t} + \nabla \cdot (\pi_t \boldsymbol{v}_t) = 0, \tag{2}$$

Figure 1: Riemannian GD

where $\pi_t \in \mathcal{P}$ and $\boldsymbol{v}_t \in \mathcal{T}_{\pi_t} \mathcal{P}$ [9] is a vector field mapping $\mathcal{X} \to \mathcal{X}$. The Fokker–Planck equation implies that $\pi_t$ describes the probability distribution of the stochastic ordinary differential equation (SODE) $d\boldsymbol{x}_t = \boldsymbol{v}_t(\boldsymbol{x}_t)dt$, where the randomness originates from the initial condition $\boldsymbol{x}_0$. Consequently, the tangent vector to the curve $\pi_t$, i.e., the direction of curve is given by $\boldsymbol{v}_t$ in the F-P equation (2). Besides, for $\boldsymbol{u}, \boldsymbol{v} \in \mathcal{T}_\pi \mathcal{P}$ the Riemannian metric [10] in Wasserstein space is $\langle \boldsymbol{u}, \boldsymbol{v} \rangle_\pi = \int \langle \boldsymbol{u}, \boldsymbol{v} \rangle d\pi$, and $\|\boldsymbol{u}\|_\pi^2 = \langle \boldsymbol{u}, \boldsymbol{u} \rangle_\pi$, where $\langle \cdot, \cdot \rangle$ is the inner product in Euclidean space. Finally, the exponential map in Wasserstein space is

$$\mathrm{Exp}_\pi[\boldsymbol{v}] = (\boldsymbol{v} + \mathrm{id})_{\#\pi}, \tag{3}$$

where $(\boldsymbol{v} + \mathrm{id})_{\#\pi}$ is the probability distribution of random variable $\boldsymbol{x} + \boldsymbol{v}(\boldsymbol{x})$ with $\boldsymbol{x} \sim \pi$ ([9], page 44). In this paper, we mainly explore minimizing the KL divergence [44] to a target probability measure $\mu$

$$\min_{\pi \in \mathcal{P}} F(\pi) = \min_{\pi \in \mathcal{P}} D_{KL}(\pi \| \mu) = \min_{\pi \in \mathcal{P}} \int \log \frac{d\pi}{d\mu} d\pi. \tag{4}$$

Here, the target distribution $\mu$ is assumed to satisfy $\mu \propto \exp(-V(\boldsymbol{x}))$ for some potential function $V(\boldsymbol{x})$ [10]. For the minimization problem (4), global convergence results have been established under specific regularity conditions, such as the log-Sobolev inequality [9, 44]. For the KL divergence $F(\pi) = D_{\mathrm{KL}}(\pi \| \mu)$, this inequality takes the form

$$D_{\mathrm{KL}}(\pi \| \mu) \le \frac{1}{2\gamma} \int \left| \nabla \log \frac{d\pi}{d\mu} \right|^2 d\pi = \frac{1}{2\gamma} |\mathrm{grad}_\pi D_{\mathrm{KL}}(\pi \| \mu)|_\pi^2, \tag{5}$$

for some $\gamma > 0$ and all $\pi$, where the equality follows from Proposition 2. In what follows, we write $\mathrm{grad}_\pi D_{\mathrm{KL}}(\pi \| \mu)$ as $\mathrm{grad} D_{\mathrm{KL}}(\pi \| \mu)$ to simplify notation. In fact, the log-Sobolev inequality in the Wasserstein space generalizes the PL inequality [23, 11], thereby guaranteeing global convergence of optimization methods. Further details can be found in Section D of the Appendix.

In this paper, we need the following lemma from [29, 42], which connects the SODE and SDE.

**Lemma 1.** *[29, 42] The SDE $d\boldsymbol{x}_t = \boldsymbol{b}(\boldsymbol{x}_t, t)dt + \boldsymbol{G}(\boldsymbol{x}_t, t)dW_t$, has the same density with SODE*

$$d\boldsymbol{x}_t = \boldsymbol{b}(\boldsymbol{x}_t, t) - \frac{1}{2} \nabla \cdot \left[ \boldsymbol{G}(\boldsymbol{x}_t, t) \boldsymbol{G}^\top(\boldsymbol{x}_t, t) \right] - \frac{1}{2} \boldsymbol{G}(\boldsymbol{x}_t, t) \boldsymbol{G}^\top(\boldsymbol{x}_t, t) \nabla \log \pi_t(\boldsymbol{x}_t) dt, \tag{6}$$

*where $\pi_t$[2] is the corresponded probability measure of $\boldsymbol{x}_t$.*

As shown by this lemma and (2), the direction of $\pi_t$ can be directly linked to a SDE. For further details on these preliminaries, we refer readers to [1, 6, 9].

---

[2]We simplify $\log (d\pi_t/d\boldsymbol{x})(\boldsymbol{x}_t)$ as $\log \pi_t(\boldsymbol{x}_t)$ if there is no obfuscation in sequel.

# 4 Riemannian Gradient Flow

In this section, we investigate the continuous gradient flow for minimizing the KL divergence in the Wasserstein space through the framework of Riemannian manifold optimization. Although this continuous optimization method has been studied previously [28, 40, 9], **a systematic analysis from the perspective of manifold optimization remains lacking**. We demonstrate that our approach provides valuable insights for the subsequent development of Riemannian SGD and SVRG flows.

## 4.1 Constructing Riemannian Gradient Flow

We first calculate the Riemannian gradient of KL divergence (proved in Appendix A).

**Proposition 1.** *The Riemannian gradient of $F(\pi) = D_{KL}(\pi \parallel \mu)$ in Wasserstein space is*

$$\mathrm{grad}F(\pi) = \mathrm{grad}D_{KL}(\pi \parallel \mu) = \nabla \log \frac{d\pi}{d\mu}. \tag{7}$$

With the defined Riemannian gradient of the KL divergence and exponential map (3), we can implement the discrete Riemannian gradient descent as in (1).

As noted in Section 1, we aim to establish a correspondence between discrete dynamics and their continuous counterparts. In Euclidean space, the GD dynamics $(\boldsymbol{x}_{n+1} - \boldsymbol{x}_n)/\eta = -\nabla F(\boldsymbol{x}_n)$ leads to an ODE (gradient flow) $d\boldsymbol{x}_t = -\nabla F(\boldsymbol{x}_t)dt$ in the limit as $\eta \to 0$. However, this approach does not directly extend to the manifold setting, since the Riemannian GD dynamics $\pi_n$ in (1) does not induce a linear structure. Fortunately, the probability measure $\pi_n$ corresponds to random vectors $\boldsymbol{x}_n \in \mathbb{R}^d$ such that $\boldsymbol{x}_n \sim \pi_n$ and $\boldsymbol{x}_{n+1} = \boldsymbol{x}_n - \eta \nabla \log \frac{d\pi_n}{d\mu}(\boldsymbol{x}_n)$. Consequently, the dynamics induced by Riemannian GD (1) can be used to construct a differential equation in the limit $\eta \to 0$. The corresponding Fokker–Planck equation for this differential equation in the Wasserstein space then yields the gradient flow. The conceptual framework is illustrated in Figure 2, and the formal result is stated in the following proposition.

**Assumption 1.** *For probability measure $\mu$, the $\log \mu$ and $\nabla \log \mu$ are all Lipschitz continuous with coefficient $L_1$ and $L_2$, respectively.* [3]

Figure 2: Our idea to bridge the discrete dynamics to its continuous counterparts.

**Proposition 2.** *Under Assumption 1 and $1 \leq n \leq \mathcal{O}(\lfloor 1/\eta \rfloor)$, the discrete Riemannian GD (1) approximates continuous Riemannian gradient flow $\pi_t$*

$$\frac{\partial}{\partial t}\pi_t = \nabla \cdot (\pi_t \mathrm{grad}D_{KL}(\pi_t \parallel \mu)) = \nabla \cdot \left( \pi_t \nabla \log \frac{d\pi_t}{d\mu} \right), \tag{8}$$

*by $\mathbb{E}[\|\boldsymbol{x}_n - \hat{\boldsymbol{x}}_{n\eta}\|^2] \leq \mathcal{O}(\eta)$, where $\boldsymbol{x}_n \sim \pi_n$ in (1), $\hat{\boldsymbol{x}}_{n\eta} \sim \pi_{n\eta}$ in (8), for $\boldsymbol{x}_0 = \hat{\boldsymbol{x}}_0$.*

This proposition shows that the discrete Riemannian GD approximates the flow (8) for any finite product $n\eta$ in the limit as $\eta \to 0$. Therefore, the flow (8) indeed corresponds to the Riemannian gradient flow. Moreover, by combining Lemma 1 with the F-P equation (2), the measure $\pi_t$ in (8) is the distribution of Langevin dynamics, which serves as a standard stochastic sampling algorithm [35] given by $d\boldsymbol{x}_t = \nabla \log \mu(\boldsymbol{x}_t)dt + \sqrt{2}dW_t$. This connection is also discussed in [10, 40, 26].

**Remark 1.** *Notably, existing literature [40] has also established that (8) corresponds to the Riemannian gradient flow in the Wasserstein space for minimizing the KL divergence. **However, their results are not derived from a limiting procedure on the learning rate in discrete Riemannian optimization methods, unlike our approach.** The authors [40] obtain the curve $\pi_t$ by solving the variational problem $\pi_{n+1} = \arg\min_\pi \left[ D_{KL}(\pi \parallel \mu) + W_2^2(\pi, \pi_n)/2\eta \right]$ in the limit $\eta \to 0$. Unfortunately, unlike our method, this approach cannot be extended to the stochastic optimization setting in Section 5, as the aforementioned minimization problem does not readily incorporate stochastic gradients.*

## 4.2 Convergence of Riemannian GD Flow

In this section, we prove the convergence of the Riemannian gradient flow (8). For non-convex problems in Euclidean space, convergence is typically measured by the first-order stationarity criterion

---

[3] Since $\mu \propto \exp(-V(\boldsymbol{x}))$, the condition means $V(\boldsymbol{x})$ and $\nabla V(\boldsymbol{x})$ are all Lipschitz continuous.

---

**Algorithm 1** Discrete Riemannian SGD

---

**Input:** Exponential map Exp, initialized $\pi_0$, learning rate $\eta$, steps $M$.

 1: **for** $n = 0, \cdots, M - 1$ **do**
 2:     Sample $\xi_n \sim \xi$ independent with $\pi_n$;
 3:     Update $\pi_{n+1} = \mathrm{Exp}_{\pi_n} \left[ -\eta \mathrm{grad} D_{KL}(\pi \parallel \mu_{\xi_n}) \right]$;
 4: **end for**
 5: **Return:** $\pi_M$.

---

[19, 6, 51]. Accordingly, in the Wasserstein space, we analogously analyze the convergence rate of the Riemannian gradient norm $|\mathrm{grad} D_{\mathrm{KL}}(\pi_t \parallel \mu)|\pi_t^2$ [4], following the approach of Boumal et al. [6], Balasubramanian et al. [2]. Moreover, the PL inequality [23] ensures global convergence in Euclidean space; for instance, under this condition, the gradient flow converges exponentially to a global minimum [23]. As discussed in Section 3, in the Wasserstein space, the Riemannian PL inequality is generalized by the log-Sobolev inequality [44], which likewise guarantees an exponential global convergence rate for the Riemannian gradient flow, as stated in the following theorem.

**Theorem 1.** *[[9]] Let $\pi_t$ follows the Riemannian gradient flow* (8)*, then for any $T > 0$, we have*

$$\frac{1}{T} \int_0^T \|\mathrm{grad} D_{KL}(\pi_t \parallel \mu)\|_{\pi_t}^2 dt \leq \frac{D_{KL}(\pi_0 \parallel \mu)}{T}. \tag{9}$$

*Moreover, if the log-Sobolev inequality* (5) *(Riemannian PL inequality) is satisfied for $\mu$, then*

$$D_{KL}(\pi_t \parallel \mu) \leq e^{-2\gamma t} D_{KL}(\pi_0 \parallel \mu). \tag{10}$$

We prove it in Appendix A. The convergence rate of Riemannian gradient flow is also proved in [9] and it matches the results in Euclidean setting [43, 23] as expected. We prove this theorem here to illustrate the criteria of convergence rates under different conditions.

## 5 Riemannian Stochastic Gradient Flow

In practice, SGD is preferred over GD due to its lower computational complexity. However, unlike in Euclidean space [20, 25], the continuous SGD flow in Wasserstein space has not been explored yet. Next, our goal is to generalize the Riemannian gradient flow (8) to the Riemannian SGD flow.

### 5.1 Constructing Riemannian Stochastic Flow

The stochastic algorithm is developed to minimize the stochastic optimization problem such that

$$\min_{\pi} \mathbb{E}_\xi \left[ f_\xi(\pi) \right] = \min_{\pi} \mathbb{E}_\xi [D_{KL}(\pi \parallel \mu_\xi)], \tag{11}$$

where the expectation is taken over $\xi$ parameterizes a set of probability measures $\mu_\xi$ [5]. For objective (11), we can get the optima of it by the following proposition proved in Appendix B.

**Proposition 3.** *The global optima of problem* (11) *is $\mu \propto \exp\left(\mathbb{E}_\xi \left[\log \mu_\xi\right]\right)$*

Since we can explicitly get the optima $\mu$ defined in Proposition (3), the goal of Riemannian SGD flow should be moving towards it. Thus, the target distribution becomes $\mu$ Proposition (3) in the sequel.

To solve the problem (11), one may use the standard discrete method, Riemannian SGD [4, 50] as outlined in Algorithm 1. By analogy with our derivation of the Riemannian GD flow in Section 4, the Riemannian SGD flow is naturally posited as its continuous counterpart. To derive this flow, we first construct the Euclidean-space dynamics $\boldsymbol{x}_n$ of Algorithm 1. Then, following a procedure similar to that in Proposition 2, we approximate the dynamics of $\boldsymbol{x}_n$ by a continuous SDE. The corresponding F-P equation then yields the continuous Riemannian SGD flow. Below is the result.

**Assumption 2.** *For any $\xi$ and probability measure $\mu_\xi$, $\log \mu_\xi$ and $\nabla \log \mu_\xi$ are Lipschitz continuous with coefficient $L_1$ and $L_2$ respectively.* [6]

---

[4]$|\mathrm{grad} D\mathrm{KL}(\pi_t \parallel \mu)|_{\pi_t}^2 \to 0$ does not imply $D(\pi_t, \mu) \to 0$ for other probability distances or divergences $D(\cdot, \cdot)$, such as the total variation distance. Further details can be found in Balasubramanian et al. [2].
[5]$\mu_\xi(\boldsymbol{x})$ is assumed to be $\mu_\xi(\boldsymbol{x}) \propto \exp(-V_\xi(\boldsymbol{x}))$.
[6]As $\mu_\xi \propto \exp(-V_\xi)$, the assumption implies $V_\xi$ and its gradient are Lipschitz continuous.

**Proposition 4.** *Under Assumption 2, let $f_\xi(\pi) = D_{KL}(\pi \parallel \mu_\xi)$, the discrete Riemannian SGD Algorithm 1 with $1 \le n \le \mathcal{O}(\lfloor 1/\eta \rfloor)$ approximates the continuous Riemannian stochastic gradient flow*

$$\frac{\partial}{\partial t}\pi_t = \nabla \cdot \left[ \pi_t \left( \nabla \log \frac{d\pi_t}{d\mu} - \frac{\eta}{2}\nabla \cdot \Sigma_{\text{SGD}} - \frac{\eta}{2}\Sigma_{\text{SGD}}\nabla \log \pi_t \right) \right], \tag{12}$$

*by $\mathbb{E}[\|\boldsymbol{x}_n - \hat{\boldsymbol{x}}_{n\eta}\|^2] \le \mathcal{O}(\eta)$, where $\boldsymbol{x}_n \sim \pi_n$ in Algorithm 1, $\hat{\boldsymbol{x}}_{n\eta} \sim \pi_{n\eta}$ in (12), for $\boldsymbol{x}_0 = \hat{\boldsymbol{x}}_0$. Here*

$$\Sigma_{\text{SGD}}(\boldsymbol{x}) = \mathbb{E}_\xi[(\nabla \log \mu_\xi(\boldsymbol{x}) - \nabla\mathbb{E}_\xi[\log \mu_\xi(\boldsymbol{x})])(\nabla \log \mu_\xi(\boldsymbol{x}) - \nabla\mathbb{E}_\xi[\log \mu_\xi(\boldsymbol{x})])^\top]. \tag{13}$$

As can be seen, similar to Proposition 2, we approximate the discrete Riemannian SGD with continuous flow (12), so that it is Riemannian SGD flow as desired.

One may observe that the Riemannian "stochastic" gradient flow (12) is a deterministic curve in the Wasserstein space $\mathcal{P}$. This is not inconsistent with the randomness in the index $\xi_n$ for $\pi_n$ in discrete Riemannian SGD. This is because the stochasticity introduced by $\xi_n$ when obtaining $\pi_n$ (with $\boldsymbol{x}_n \sim \pi_n$) in Algorithm 1 is implicitly captured in the corresponding $\boldsymbol{x}_n$. In other words, the randomnesses from the random vector $\boldsymbol{x}_n$ itself and the random index $\xi_n$ are fully incorporated into $\boldsymbol{x}_n$ and manifested in its continuous approximation $\hat{\boldsymbol{x}}_{n\eta} \sim \pi_{n\eta}$, where $\pi_t \in \mathcal{P}$ is the deterministic curve (12) in the Wasserstein space. Thus, all randomness in the Riemannian SGD is accounted in this formulation.

Besides, we can find that implementing the discrete Riemannian SGD is non-trivial, since it requires Riemannian gradient $\nabla \log (d\pi_t/d\mu)$. However, during proving Proposition 4, we show the discrete stochastic gradient Langevin dynamics (SGLD) [49]

$$\boldsymbol{x}_{n+1} = \boldsymbol{x}_n + \eta\nabla \log \mu_{\xi_n}(\boldsymbol{x}_n) + \sqrt{2\eta}\boldsymbol{\epsilon}_n \tag{14}$$

approximates [7] the corresponded dynamics of $\{\boldsymbol{x}_n\}$ in discrete Riemannian SGD (Algorithm 1)

$$\boldsymbol{x}_{n+1} = \boldsymbol{x}_n + \eta\nabla \log \frac{d\mu_{\xi_n}}{d\pi_n}(\boldsymbol{x}_n). \tag{15}$$

Thus, in practice, discrete SGLD can be implemented to approximate discrete Riemannian SGD. Furthermore, based on this approximation and Proposition 4, the continuous counterpart $\boldsymbol{x}_t$ of (15), as governed by the Riemannian SGD flow (12) (Lemma 1), satisfies the SDE

$$d\boldsymbol{x}_t = \nabla \log \frac{d\mu}{d\pi_t}(\boldsymbol{x}_t)dt + \sqrt{\eta}\Sigma_{\text{SGD}}^{\frac{1}{2}}(\boldsymbol{x}_t)dW_t, \tag{16}$$

is also the continuous limit of the discrete SGLD (14). This establishes a connection between the Riemannian SGD flow and discrete SGLD—that is, the Riemannian SGD flow in the Wasserstein space corresponds precisely to continuous SGLD. **Consequently, our Riemannian SGD flow provides a powerful framework for analyzing discrete SGLD or Riemannian SGD.** For example, combining Proposition 4 and Theorem 2 implies the convergence rate of discrete Riemannian SGD.

## 5.2 Convergence of Riemannian SGD Flow

Next, we examine the convergence rate of Riemannian SGD flow. As in Section 4, our analyses are respectively conducted with/without log-Sobolev inequality.

**Theorem 2.** *Let $\pi_t$ follows the Riemannian SGD flow (12) and $\mu$ defined in Proposition (3). Under Assumption 2, if $T \ge \frac{64L_1^4 D_{KL}(\pi_0\|\mu)}{4dL_1^2 L_2 + (d+1)^2 L_2^2}$, then by taking $\eta = \sqrt{\frac{D_{KL}(\pi_0\|\mu)}{T(4dL_1^2 L_2 + (d+1)^2 L_2^2)}}$, we have*

$$\frac{1}{\eta T}\int_0^{\eta T}\|\text{grad}D_{KL}(\pi_t \parallel \mu)\|_{\pi_t}^2 dt \le \frac{4D_{KL}(\pi_0 \parallel \mu)}{\eta T} = \mathcal{O}\left(\frac{1}{\sqrt{T}}\right). \tag{17}$$

*Besides that, if (5) is satisfied for $\mu$, $\eta = 1/\gamma T^\alpha$ with $0 < \alpha < 1$, and $T \ge (8L_1^2/\gamma)^{1/\alpha}$, then*

$$D_{KL}(\pi_{\eta T} \parallel \mu) \le \frac{1}{\gamma T^\alpha}\left[4dL_1^2 L_2 + (d+1)^2 L_2^2\right] = \mathcal{O}\left(\frac{1}{T^\alpha}\right). \tag{18}$$

---

[7]The approximation is verified by noting $\mathbb{E}_\pi[\langle\nabla f, \nabla \log \pi\rangle] = -\mathbb{E}_\pi[\Delta f]$ for continuous test function $f$, and combining Taylor's expansion, please check Appendix B for more details.

The proof of this theorem is provided in Appendix B.2. Notably, the presence of $\eta$ in the SDE leads to a convergence rate of order $\mathcal{O}(1/\sqrt{T})$ for the Riemannian SGD flow[8]. Under the log-Sobolev inequality, a global convergence rate of $\mathcal{O}(1/T)$ can be established (by taking $\alpha \to 1$). Therefore, the convergence rates proved in Theorem 2 align with those of the continuous SGD flow in Euclidean space [19, 33], as demonstrated in Appendix B.2.

**Remark 2.** *During the proof to Theorem 2, we assume* $\log \mu_\xi$ *is Lipschitz continuous. The assumption can be relaxed as* $\mathbb{E}_{\xi, \boldsymbol{x}_t}[\|\nabla \log \mu(\boldsymbol{x}_t) - \mathbb{E}_\xi[\nabla \log \mu(\boldsymbol{x}_t)]\|^2] \leq \sigma^2$, *for constant* $\sigma$ *and* $\boldsymbol{x}_t \sim \pi_t$ *in* (12)*, which is the standard "bounded variance" assumption in optimization on Euclidean space [19].*

In the remainder of this section, we further demonstrate the tightness of the derived convergence rate for the Riemannian SGD flow through the following example. Although this example does not satisfy the Lipschitz continuity condition on $\log \mu_\xi$ in Assumption 2, it does satisfy the condition of bounded variance discussed in the preceding remark. Consequently, the results in Theorem 2 remain valid.

**Example 1.** *Let* $\mu_{\boldsymbol{\xi}} \sim \mathcal{N}(\boldsymbol{\xi}, \boldsymbol{I})$, *with* $\boldsymbol{\xi} \in \{\boldsymbol{\xi}_1, \cdots, \boldsymbol{\xi}_N\}$, $\max_{1 \leq j \leq N} \|\boldsymbol{\xi}_j\| \leq C$ *for a constant* $C$.

Due to Proposition (3), we have $\mu \sim \mathcal{N}(\bar{\boldsymbol{\xi}}, \boldsymbol{I})$ with $\mathbb{E}[\boldsymbol{\xi}] = \bar{\boldsymbol{\xi}} = \sum_j \boldsymbol{\xi}_j / N$, which is the target measure of Riemannian SGD flow. Then, we have $\Sigma_{\text{SGD}} = \frac{1}{N} \sum_{j=1}^N (\boldsymbol{\xi}_j - \mathbb{E}[\boldsymbol{\xi}])(\boldsymbol{\xi}_j - \mathbb{E}[\boldsymbol{\xi}])^\top = \mathsf{Var}(\boldsymbol{\xi})$.

The assumptions (including log-Sobolev inequality) in Theorem 2 are all satisfied (see Lemma 3 in Appendix). Then, the corresponding SDE of Riemannian SGD flow (12) is

$$d\boldsymbol{x}_t = -(\boldsymbol{x}_t - \bar{\boldsymbol{\xi}})dt + (\sqrt{\eta}\mathsf{Var}^{\frac{1}{2}}(\boldsymbol{\xi}), \sqrt{2}\boldsymbol{I})dW_t, \tag{19}$$

which has the following closed-form solution $\boldsymbol{x}_t = \bar{\boldsymbol{\xi}} + e^{-t}(\boldsymbol{x}_0 - \bar{\boldsymbol{\xi}}) + e^{-t}\sqrt{\frac{\eta}{2}}\mathsf{Var}^{\frac{1}{2}}(\boldsymbol{\xi})W^{(1)}_{e^{2t}-1} + e^{-t}W^{(2)}_{e^{2t}-1}$, where $W^{(1)}_t$ and $W^{(2)}_t$ are standard independent Brownian motions. By taking $t = \eta T$, for any $\boldsymbol{x}_0$,

$$\boldsymbol{x}_{\eta T} \sim \mathcal{N}\left(\bar{\boldsymbol{\xi}} + e^{-\eta T}(\boldsymbol{x}_0 - \bar{\boldsymbol{\xi}}), (1 - e^{-2\eta T})(\eta\mathsf{Var}(\boldsymbol{\xi})/2 + \boldsymbol{I})\right). \tag{20}$$

Here, $e^{-\eta T} \approx 0$ for large $T$, and $\eta = \mathcal{O}(1/T^\alpha)$ with $0 < \alpha < 1$. Then, $\boldsymbol{x}_{\eta T} \approx \mathcal{N}(\bar{\boldsymbol{\xi}}, \frac{\eta}{2}\mathsf{Var}(\boldsymbol{\xi}) + \boldsymbol{I})$. The Riemannian gradient $\mathrm{grad}D_{KL}(\pi_{\eta T} \| \mu) = [(\boldsymbol{I} + \eta\mathsf{Var}(\boldsymbol{\xi})/2)^{-1} - \boldsymbol{I}](\boldsymbol{x}_{\eta T} - \bar{\boldsymbol{\xi}})$, which indicates

$$\|\mathrm{grad}D_{KL}(\pi_{\eta T} \| \mu)\|^2_{\pi_{\eta T}} = \mathrm{tr}\left((\boldsymbol{I} + \eta\mathsf{Var}(\boldsymbol{\xi})/2)^{-1} - \boldsymbol{I} + \eta\mathsf{Var}(\boldsymbol{\xi})/2\right) = \mathcal{O}\left(T^{-\alpha}\right). \tag{21}$$

On the other hand, the KL divergence between Gaussian measures [36] $\pi_{\eta T}, \mu$ can be calculated as

$$D_{KL}(\pi_{\eta T} \| \mu) \approx \frac{1}{2}[\log|\boldsymbol{I} + \eta\mathsf{Var}(\boldsymbol{\xi})/2| + \eta\mathrm{tr}(\mathsf{Var}(\boldsymbol{\xi}))/2] = \mathcal{O}\left(T^{-\alpha}\right). \tag{22}$$

As can be seen, the convergence rates in (21) and (22) are consistent with the proved convergence rates in Theorem 2, under $\alpha = 1/2$ and $\alpha \to 1$, respectively. Thus, the example (1) indicates the convergence rates in Theorem 2 are sharp.

## 6 Riemannian Stochastic Variance Reduction Gradient Flow

Next, we extend discrete algorithm SVRG [21] to its continuous counterpart in Wasserstein space.

### 6.1 Constructing Riemannian SVRG Flow

In practice, the objective in (11) often takes the form of a finite sum, where $\xi$ is uniformly distributed over $\xi_1, \cdots, \xi_N$, so that the objective becomes $\min_\pi F(\pi) = \min_\pi \mathbb{E}_\xi[f_\xi(\pi)] = \min_\pi \frac{1}{N} \sum_{j=1}^N f_{\xi_j}(\pi)$. Clearly, computing $\mathrm{grad}F(\pi)$ requires $\mathcal{O}(N)$ operations. The convergence rates in Theorems 1 and 4 imply that achieving $|\mathrm{grad}D_{KL}(\pi_t \| \mu)|^2_{\pi_t} \leq \epsilon$ for some $t$—that is, reaching an $\epsilon$-stationary point—requires computational complexities of $\mathcal{O}(N\epsilon^{-1})$ and $\mathcal{O}(\epsilon^{-2})$, respectively[9]. Therefore, for large $N$, Riemannian SGD flow offers an improvement over Riemannian GD flow.

---

[8]We emphasize that no constraints are imposed on the learning rate $\eta$ in Theorem 1; hence, the convergence rate of Riemannian GD remains as stated in Theorem 1.

[9]The computational complexity of continuous optimization is evaluated by implementing the corresponding discrete algorithm. Further details are provided in Appendix B.2

---

**Algorithm 2** Discrete Riemannian SVRG

---

**Input:** Exponential map $\mathrm{Exp}_\pi$, initialized $\pi_0$, learning rate $\eta$, epoch $I$, steps $M$ of each epoch.

1: Take $\pi_0^0 = \pi_0$;
2: **for** $i = 0, \cdots, I-1$ **do**
3:     Compute $\mathrm{grad}F(\pi_0^i) = \frac{1}{N}\sum_{j=1}^N \mathrm{grad}f_{\xi_j}(\pi_0^i)$
4:     **for** $n = 0, \cdots, M-1$ **do**
5:         Uniformly sample $\xi_n^i \in \{\xi_1, \cdots, \xi_N\}$ independent with $\pi_n^i$;
6:         Update $\pi_{n+1}^i = \mathrm{Exp}_{\pi_n^i}[-\eta(\mathrm{grad}f_{\xi_n^i}(\pi_n^i) - \Gamma_{\pi_0^i}^{\pi_n^i}(\mathrm{grad}f_{\xi_n^i}(\pi_0^i) - \mathrm{grad}F(\pi_0^i)))]$;
7:     **end for**
8:     $\pi_0^{i+1} = \pi_M^i$;
9: **end for**
10: **Return:** $\pi_N$.

---

In Euclidean space, considerable efforts have been devoted to further improving computational complexity, with methods such as SVRG [21, 56], SPIDER [18, 55], and SARAH [31]. Among these, the double-loop structure of SVRG represents a core idea shared across several variance-reduced approaches. For this reason, we focus specifically on SVRG in this paper.

As presented in Section 5, we begin with the discrete Riemannian SVRG method [52] outlined in Algorithm 2. Analogous to the Euclidean setting, line 6 of Algorithm 2 employs the tangent vector $\mathrm{grad}f_{\xi_n^i}(\pi_n^i) - \Gamma_{\pi_0^i}^{\pi_n^i}(\mathrm{grad}f_{\xi_n^i}(\pi_0^i) - \mathrm{grad}F(\pi_0^i)))$ which serves as a variance-reduced estimator of the Riemannian gradient $\mathrm{grad}F(\pi_n^i)$. This constitutes the core idea of SVRG-type algorithms. In this context, the mapping $\Gamma_{\pi_0^i}^{\pi_n^i}$ denotes parallel transport [1, 52]. Specifically, for the loss function $f_\xi(\pi) = D_{\mathrm{KL}}(\pi \parallel \mu_\xi)$, the difference of Riemannian gradients $\mathrm{grad}f_{\xi_n^i}(\pi_0^i) - \mathrm{grad}F(\pi_0^i)$ lies in the tangent space $\mathcal{T}_{\pi_0^i}\mathcal{P}$, while the exponential map $\mathrm{Exp}_{\pi_n^i}$ is defined on $\mathcal{T}_{\pi_n^i}\mathcal{P}$. To reconcile this discrepancy, the parallel transport $\Gamma_{\pi_0^i}^{\pi_n^i}: \mathcal{T}_{\pi_0^i}\mathcal{P} \to \mathcal{T}_{\pi_n^i}\mathcal{P}$ is applied, which maps vectors from $\mathcal{T}_{\pi_0^i}\mathcal{P}$ to $\mathcal{T}_{\pi_n^i}\mathcal{P}$ while preserving the Riemannian metric, i.e., $\|\Gamma_{\pi_0^i}^{\pi_n^i}(\boldsymbol{u})\|_{\pi_n^i}^2 = \|\boldsymbol{u}\|_{\pi_0^i}^2$ for any $\boldsymbol{u} \in \mathcal{T}_{\pi_0^i}$.

In Wasserstein space, we define $\Gamma_\mu^\pi(\boldsymbol{u}) = \boldsymbol{u} \circ T_{\pi\to\mu}$ for $\mu, \pi \in \mathcal{P}$ and $\boldsymbol{u} \in \mathcal{T}_\mu$, where $\circ$ is the composition operator, and $T_{\pi\to\mu}: \mathcal{X} \to \mathcal{X}$ satisfies $T_{\pi\to\mu}(\boldsymbol{x}) \sim \mu$ for $\boldsymbol{x} \sim \pi$. Thus,

$$\|\boldsymbol{u}\|_\mu^2 = \int \|\boldsymbol{u}\|^2 d\mu = \int \|\boldsymbol{u}(T_{\pi\to\mu}(\boldsymbol{x}))\|^2 d\pi(\boldsymbol{x}) = \int \|\boldsymbol{u} \circ T_{\pi\to\mu}\|^2 d\pi = \int \|\Gamma_\mu^\pi(\boldsymbol{u})\| d\pi, \quad (23)$$

which is consistent with the definition of parallel transport. Based on the preceding notations, we now proceed to construct the continuous Riemannian SVRG flow derived from Algorithm 2. The underlying rationale aligns with the approaches established in Propositions 2 and 4, namely, approximating the discrete Riemannian SVRG updates with a deterministic flow in the Wasserstein space. The formal result is stated in the following proposition.

**Proposition 5.** *Under Assumption 2, let $f_\xi(\pi) = D_{KL}(\pi \parallel \mu_\xi)$, the discrete Riemannian SVRG Algorithm 2 with $1 \le n \le \mathcal{O}(\lfloor 1/\eta \rfloor)$ approximates the Riemannian SVRG flow*

$$\frac{\partial}{\partial t}\pi_t(\boldsymbol{x}) = \nabla \cdot \left[\pi_t(\boldsymbol{x})\left(\nabla \log \frac{d\pi_t}{d\mu}(\boldsymbol{x}) - \frac{\eta}{2\pi_t(\boldsymbol{x})}\int \pi_{t_i,t}(\boldsymbol{y},\boldsymbol{x})\nabla_{\boldsymbol{x}}\cdot\Sigma_{\mathrm{SVRG}}(\boldsymbol{y},\boldsymbol{x})d\boldsymbol{y}\right.\right.$$
$$\left.\left. - \frac{\eta}{2\pi_t(\boldsymbol{x})}\int \pi_{t_i,t}(\boldsymbol{y},\boldsymbol{x})\Sigma_{\mathrm{SVRG}}(\boldsymbol{y},\boldsymbol{x})\nabla_{\boldsymbol{x}}\log\pi_{t_i,t}(\boldsymbol{y},\boldsymbol{x})d\boldsymbol{y}\right)\right], \quad (24)$$

*for $iM\eta = t_i \le t \le t_{i+1}$, $(\hat{\boldsymbol{x}}_{t_i}, \hat{\boldsymbol{x}}_t) \sim \pi_{t_i,t}$, $\hat{\boldsymbol{x}}_t \sim \pi_t$ in (24), since we have $\mathbb{E}[\|\boldsymbol{x}_n^i - \hat{\boldsymbol{x}}_{(iM+n)\eta}\|^2] \le \mathcal{O}(\eta)$, where $\boldsymbol{x}_n^i \sim \pi_n^i$ in Algorithm 2 for $\boldsymbol{x}_0^0 = \hat{\boldsymbol{x}}_0$. Here*

$$\Sigma_{\mathrm{SVRG}}(\boldsymbol{y},\boldsymbol{x}) = \mathbb{E}_\xi\left[(\nabla\log\mu_\xi(\boldsymbol{x}) - \nabla\log\mu_\xi(\boldsymbol{y}) + \nabla\mathbb{E}_\xi[\log\mu_\xi(\boldsymbol{y})] - \nabla\mathbb{E}_\xi\log\mu_\xi(\boldsymbol{x}))\right.$$
$$\left.(\nabla\log\mu_\xi(\boldsymbol{x}) - \nabla\log\mu_\xi(\boldsymbol{y}) + \nabla\mathbb{E}_\xi[\log\mu_\xi(\boldsymbol{y})] - \nabla\mathbb{E}_\xi\log\mu_\xi(\boldsymbol{x}))^\top\right]. \quad (25)$$

As in (4), the Riemannian SVRG flow defined in (5) also constitutes a deterministic curve in the Wasserstein space, which can be explained through similar discussion following Proposition 4.

Consequently, all three continuous flows in the Wasserstein space, (8), (12), and (24) describe deterministic trajectories, irrespective of any randomness in their discrete counterparts.

Moreover, similar to the discussion after Proposition 4, although the discrete Riemannian SVRG method presents implementation challenges, it can be effectively approximated through both the discrete SVRG Langevin dynamics [56] and the proposed Riemannian SVRG flow in (24). **Consequently, our formulation presents a valuable analytical framework for both SVRG Langevin dynamics and discrete Riemannian SVRG algorithms**. Further details are in Appendix C.1.

## 6.2 Convergence of Riemannian SVRG Flow

In this subsection, we analyze the convergence rate of the Riemannian SVRG flow defined in equation (24). The main result, stated in Theorem 3 below, adopts the same notations as those introduced in Proposition 5.

**Theorem 3.** *Let $\pi_t$ follows Riemannian SVRG flow* (24), *$\mu$ defined in Proposition* (3), *for sequences $\{t_i\}$ with $\Delta = t_1 - t_0 = \cdots = t_I - t_{I-1} = \mathcal{O}(1/\sqrt{\eta})$, and $\eta T = I\Delta$ to run Riemannian SVRG flow for $I$ epochs. Then for any $T$, if the $\nabla \log \mu_\xi$ is Lipschitz continuous with coefficient $L_2$ for all $\xi$, under proper $\pi_0$, we have*

$$\frac{1}{\eta T} \sum_{i=1}^{I} \int_{t_i}^{t_{i+1}} \|\mathrm{grad} D_{KL}(\pi_t \parallel \mu)\|_{\pi_t}^2 \, dt \leq \frac{2D_{KL}(\pi_0 \parallel \mu)}{\eta T}. \tag{26}$$

*By taking $\eta = \mathcal{O}(N^{-2/3})$, the computational complexity of Riemannian SVRG flow is of order $\mathcal{O}(N^{2/3}/\epsilon)$ to make $\min_{0 \leq t \leq \eta T} \|\mathrm{grad} D_{KL}(\pi_t \parallel \mu)\|_{\pi_t}^2 \leq \epsilon$.*

*Furthermore, when the log-Sobolev inequality* (5) *is satisfied for $\mu$, we have*

$$D_{KL}(\pi_{\eta T} \parallel \mu) \leq e^{-\gamma \eta T} D_{KL}(\pi_0 \parallel \mu), \tag{27}$$

*and it takes $\mathcal{O}((N + \gamma^{-1} N^{2/3}) \log \epsilon^{-1})$ computational complexity to make $D_{KL}(\pi_{\eta T} \parallel \mu) \leq \epsilon$.*

The proof of this theorem is deferred to Appendix C.1. As shown, for non-convex problems, the computational complexity required to reach an $\epsilon$-stationary point of the Riemannian SVRG flow is $\mathcal{O}(N^{2/3}\epsilon^{-1})$. This complexity can be lower than that of Riemannian GD flow, which is $\mathcal{O}(N\epsilon^{-1})$, and Riemannian SGD flow, which is $\mathcal{O}(\epsilon^{-2})$, when $\epsilon$ is sufficiently small (as clarified in Section 6.1).

On the other hand, under the Riemannian PL-inequality (i.e., log-Sobolev inequality), the computational complexities required to achieve $D_{\mathrm{KL}}(\pi_{\eta T} \parallel \mu) \leq \epsilon$ are $\mathcal{O}(\gamma^{-1} N \log \epsilon^{-1})$ and $\mathcal{O}(\gamma^{-1} \epsilon^{-1})$ for Riemannian GD and SGD flows. These can be improved by the Riemannian SVRG flow when $\gamma^{-1} \geq 1$ and $\mathcal{O}(\epsilon \log \epsilon^{-1}) \leq \mathcal{O}((\gamma N + N^{\frac{2}{3}})^{-1})$. Besides that, it is worth noting that, no matter with/without the log-Sobolev inequality, the proved convergence rates match the results in Euclidean space [33], and computational complexities match the discrete SVRG in Euclidean space [37].

In Theorem 3, the initial distribution $\pi_0$ is required to satisfy $\mathbb{E}_{\pi_{t_i,t}}[\mathrm{tr}(\nabla^2 \log (d\pi_t/d\mu)\Sigma_{\mathrm{SVRG}})] \leq \lambda_t \mathbb{E}_{\pi_{t_i,t}}[\mathrm{tr}(\Sigma_{\mathrm{SVRG}})]$ where $\lambda_t$ is a polynomial function of $t$. Since no constraint is imposed on the order of $\lambda_t$, which may grow arbitrarily large, this assumption can be readily satisfied in practice. Further technical details are provided in Appendix C.1.

As in Section 5.2, the convergence results in Theorem 3 do not require Lipschitz continuous $\log \mu_\xi$, so that Example 1 is suitable to be analyzed as follows.

As discussed in Section 6.1, the fundamental principle of SVRG lies in reducing the variance of stochastic gradient estimates at each update, thereby accelerating convergence. Interestingly, in Example 1, the induced noise covariance $\Sigma_{\mathrm{SVRG}} = 0$, which causes the Riemannian SVRG flow (24) degenerate into Riemannian GD flow (8). This indicates that the variance reduction technique completely eliminates gradient variance in this case. Notably, since $\nabla \log \mu_\xi(\boldsymbol{x}) = (\boldsymbol{x} - \boldsymbol{\xi})$ and $\nabla \log \mu(\boldsymbol{x}) = (\boldsymbol{x} - \bar{\boldsymbol{\xi}})$, the corresponding random vector $\boldsymbol{x}_n^i \in \mathbb{R}^d$ in the discrete Riemannian SVRG (line 6 of Algorithm 2) satisfies.

$$\boldsymbol{x}_{n+1}^i = \boldsymbol{x}_n^i - \eta \left( \nabla \log \frac{d\mu_{\xi_n^i}}{d\pi_n^i}(\boldsymbol{x}_n^i) - \nabla \log \frac{d\mu_{\xi_n^i}}{d\pi_0^i}(\boldsymbol{x}_0^i) + \nabla \log \frac{d\mu}{d\pi_0^i}(\boldsymbol{x}_0^i) \right) = \boldsymbol{x}_n^i + \eta \nabla \log \frac{d\mu}{d\pi_n^i}(\boldsymbol{x}_n^i),$$

which is exactly the discretion of Riemannian GD flow, but with $\mathcal{O}(1)$ (instead of $\mathcal{O}(N)$ as we do not compute $\bar{\boldsymbol{\xi}}$ for each $n$) computational complexity for each update step in line 6 of Algorithm

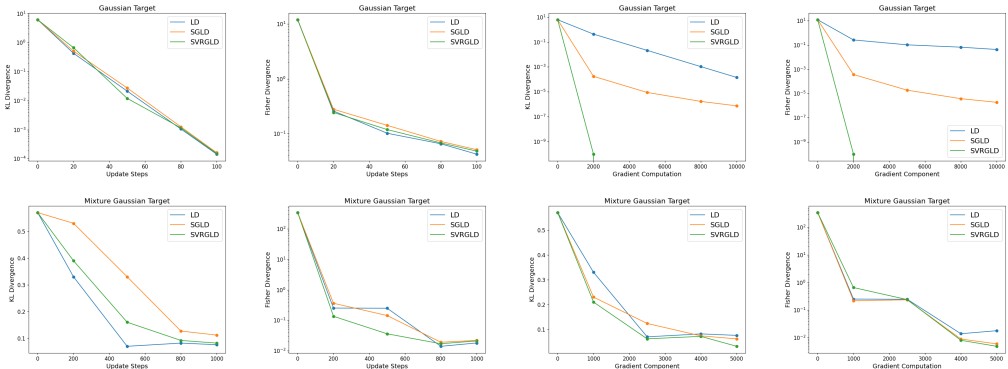

Figure 3: The convergence rates measured by KL divergence and Fisher divergence (Riemannian gradient norm) under different optimization methods.

2. This explains the improved computational complexity of Riemannian SVRG flow. Note that the corresponded SDE of (24) is $d\boldsymbol{x}_t = -(\boldsymbol{x}_t - \bar{\boldsymbol{\xi}})dt + \sqrt{2}dW_t$, with closed-form solution $\boldsymbol{x}_{\eta T} = \bar{\boldsymbol{\xi}} + e^{-\eta T}(\boldsymbol{x}_0 - \bar{\boldsymbol{\xi}}) + e^{-\eta T}W_{e^{2\eta T}-1}$. Then, we can prove the convergence rates of $D_{KL}(\pi_{\eta T} \parallel \mu)$ ($\boldsymbol{x}_{\eta T} \sim \pi_{\eta T}$) and its first order criteria are all of order $\mathcal{O}(e^{-\eta T})$, where the global convergence rate matches the result in Theorem 3.

## 7 Experiments

In this section, we empirically evaluate the proposed sampling algorithms on two examples.

**Gaussian.** In this case, we set $N = 100$, $d = 2$, $\mu_{\boldsymbol{\xi}_i} \sim (\boldsymbol{\xi}_i, \boldsymbol{I})$ with each $\boldsymbol{\xi}_i \sim \mathcal{N}(\boldsymbol{0}, \boldsymbol{I})$. Then, we get the target distribution $\mu \sim \mathcal{N}(\bar{\boldsymbol{\xi}}, \boldsymbol{I})$. Within this framework, we report the KL divergence $D_{KL}(\pi_{\eta T} \parallel \mu)$ and the Fisher divergence $\|\text{grad} D_{KL}(\pi_{\eta T} \parallel \mu)\|^2$ (first order criteria), where the $\pi_{\eta T}$ is obtained by Riemannian GD flow, Riemannian SGD flow or SVRG flow. In this case, due to Example 1, the two criteria can be explicitly estimated. The results are summarized in Figure 3.

**Mixture Gaussian.** In this case, we set $N = 5$, $d = 2$ with $\mu_{\boldsymbol{\xi}_i} \sim \frac{1}{2}\mathcal{N}(\boldsymbol{\xi}_{i,1}, \boldsymbol{I}) + \frac{1}{2}\mathcal{N}(\boldsymbol{\xi}_{i,2}, \boldsymbol{I})$, where $\boldsymbol{\xi}_{i,k} \sim \mathcal{N}(0, \boldsymbol{I})$. Unfortunately, the proposed flows can not be explicitly computed. Therefore, we implement their discrete versions as in Algorithms 1, 2. Then we report the KL divergence $D_{KL}(\pi_{\eta T} \parallel \mu)$ and Fisher divergence $\|\text{grad} D_{KL}(\pi_{\eta T} \parallel \mu)\|^2$ with $\pi_{\eta T}$ are approximated by the discrete Riemannian GD, SGD, and SVRG Algorithms. Here, the KL divergence and Riemannian gradient are estimated by density estimation as in [], with 1000 independent samples. The results are summarized in Figure 3.

As can be seen, under the same update steps, Riemannian SVRG and Riemannian GD have better convergence results than Riemannian SGD (Theorem 1, 2, and 3), while under the same gradient computations, Riemannian SGD and Riemannian SVRG (especially Riemannian SVRG) are sharper (see discussion after Theorem 3). These results are consistent with theoretical conclusions.

## 8 Conclusion

Based on the principles of Riemannian manifold optimization, this paper investigates continuous Riemannian SGD and SVRG flows for minimizing KL divergence within the Wasserstein space. We establish convergence rates for these stochastic flows that align with known results in Euclidean settings. Our technical approach involves constructing SDEs in Euclidean space by taking the limit of vanishing step sizes in discrete Riemannian optimization methods, where the corresponding Fokker-Planck equations characterize the desired curves in Wasserstein space. This framework provides new theoretical insights into continuous stochastic Riemannian optimization and demonstrates the utility of continuous methods as analytical tools for studying discrete optimization algorithms.

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

# A Proofs in Section 4

**Proposition 1.** *The Riemannian gradient of $F(\pi) = D_{KL}(\pi \parallel \mu)$ in Wasserstein space is*

$$\mathrm{grad}F(\pi) = \mathrm{grad}D_{KL}(\pi \parallel \mu) = \nabla \log \frac{d\pi}{d\mu}. \tag{7}$$

*Proof.* For any curve $\pi_t \in \mathcal{P}$ with $\pi_0 = \pi$, we have,

$$
\begin{aligned}
\lim_{t\to 0} \frac{D_{KL}(\pi_t \parallel \mu) - D_{KL}(\pi_0 \parallel \mu)}{t} &= \mathrm{Dev}D_{KL}(\pi_0 \parallel \mu)[\boldsymbol{v}_0] \\
&= \frac{\partial}{\partial t} D_{KL}(\pi_0 \parallel \mu) \mid_{t=0} \\
&= \int (1 + \log \pi_t) \frac{\partial}{\partial t} \pi_t d\boldsymbol{x} \mid_{t=0} + \int \frac{\partial}{\partial t} \pi_t \log \mu d\boldsymbol{x} \mid_{t=0} \\
&= \int \langle \nabla \log \pi - \nabla \log \mu, \boldsymbol{v}_0 \rangle d\pi \\
&= \left\langle \boldsymbol{v}_0, \nabla \log \frac{d\pi}{d\mu} \right\rangle_\pi.
\end{aligned} \tag{28}
$$

Thus we prove our conclusion due to the definition of Riemannian gradient. $\qquad\square$

**Proposition 2.** *Under Assumption 1 and $1 \le n \le \mathcal{O}(\lfloor 1/\eta \rfloor)$, the discrete Riemannian GD (1) approximates continuous Riemannian gradient flow $\pi_t$*

$$\frac{\partial}{\partial t} \pi_t = \nabla \cdot (\pi_t \mathrm{grad} D_{KL}(\pi_t \parallel \mu)) = \nabla \cdot \left( \pi_t \nabla \log \frac{d\pi_t}{d\mu} \right), \tag{8}$$

*by $\mathbb{E}[\|\boldsymbol{x}_n - \hat{\boldsymbol{x}}_{n\eta}\|^2] \le \mathcal{O}(\eta)$, where $\boldsymbol{x}_n \sim \pi_n$ in (1), $\hat{\boldsymbol{x}}_{n\eta} \sim \pi_{n\eta}$ in (8), for $\boldsymbol{x}_0 = \hat{\boldsymbol{x}}_0$.*

*Proof.* For any $f$, due to the definition of $\pi_{n+1}$ in (3), we have

$$\mathbb{E}_{\pi_{n+1}}[f(\boldsymbol{x})] = \mathbb{E}_{\pi_n}[f(\boldsymbol{x} - \eta \mathrm{grad}F(\pi_n(\boldsymbol{x})))], \tag{29}$$

so that for $\boldsymbol{x}_n \sim \pi_n$ and $\boldsymbol{x}_{n+1} \sim \pi_{n+1}$, we must have

$$\boldsymbol{x}_{n+1} = \boldsymbol{x}_n - \eta \mathrm{grad}F(\pi_n(\boldsymbol{x}_n)) = \boldsymbol{x}_n - \eta \nabla \log \frac{d\pi_n}{d\mu}(\boldsymbol{x}_n). \tag{30}$$

On the other hand, let us define

$$\bar{\boldsymbol{x}}_{n+1} = \bar{\boldsymbol{x}}_n + \eta \nabla_{\boldsymbol{x}} \log \mu(\bar{\boldsymbol{x}}_n) + \sqrt{2\eta} \epsilon_n, \tag{31}$$

where $\epsilon_n \sim \mathcal{N}(0, \boldsymbol{I})$, and $\bar{\boldsymbol{x}}_0 = \boldsymbol{x}_0$. Next, let us show $\boldsymbol{x}_n$ approximates $\bar{\boldsymbol{x}}_n$. For any test function $f \in C^2$ with (spectral norm) bounded Hessian, we have

$$
\begin{aligned}
\mathbb{E}[f(\boldsymbol{x}_{n+1})] &= \mathbb{E}[f(\boldsymbol{x}_n)] + \eta \mathbb{E}[\langle \nabla f(\boldsymbol{x}_n), \nabla_{\boldsymbol{x}} \log \mu(\boldsymbol{x}_n) - \nabla_{\boldsymbol{x}} \log \pi_n(\boldsymbol{x}_n) \rangle] + \mathcal{O}(\eta^2) \\
&= \mathbb{E}[f(\boldsymbol{x}_n)] + \eta \mathbb{E}[\langle \nabla f(\boldsymbol{x}_n), \nabla_{\boldsymbol{x}} \log \mu(\boldsymbol{x}_n) \rangle] + \eta \mathbb{E}[\Delta f(\boldsymbol{x}_n)] + \mathcal{O}(\eta^2) \\
&= \mathbb{E}[f(\boldsymbol{x}_n)] + \eta \mathbb{E}[\langle \nabla f(\boldsymbol{x}_n), \nabla_{\boldsymbol{x}} \log \mu(\boldsymbol{x}_n) \rangle] + \eta \mathbb{E}\left[ \epsilon_n^\top \nabla^2 f(\boldsymbol{x}_n) \epsilon_n \right] + \mathcal{O}(\eta^2) \\
&= \mathbb{E}\left[ f\left( \boldsymbol{x}_n + \eta \nabla_{\boldsymbol{x}} \log \mu(\boldsymbol{x}_n) + \sqrt{2\eta} \epsilon_n \right) \right] + \mathcal{O}(\eta^2).
\end{aligned} \tag{32}
$$

due to the definition of $f$ and $D_{KL}(\pi_n \parallel \mu) < \infty$. Then, let us define the following

$$f_{\delta,C}(\boldsymbol{x}) = \begin{cases} \|\boldsymbol{x}\|^2 & \|\boldsymbol{x}\|^2 \le C - \delta; \\ u_\delta(\boldsymbol{x}) & C - \delta \le \|\boldsymbol{x}\|^2 \le C; \\ C & C > \|\boldsymbol{x}\|^2, \end{cases} \tag{33}$$

where $\delta > 0$, $C > 0$ and $u_\delta(\boldsymbol{x})$ is a quadratic function of $\boldsymbol{x}$ to make the above $f_{\delta,C}$ smooth. Then

$$
\begin{aligned}
\mathbb{E}[f_{\delta,C}(\boldsymbol{x}_{n+1} - \bar{\boldsymbol{x}}_{n+1})] &= \mathbb{E}\left[ f_{\delta,C}\left( \boldsymbol{x}_n + \eta \nabla_{\boldsymbol{x}} \log \mu(\boldsymbol{x}_n) + \sqrt{2\eta} \epsilon_n - \bar{\boldsymbol{x}}_{n+1} \right) \right] + \mathcal{O}(\eta^2) \\
&= \mathbb{E}[f_{\delta,C}(\boldsymbol{x}_n + \eta \nabla_{\boldsymbol{x}} \log \mu(\boldsymbol{x}_n) - \bar{\boldsymbol{x}}_n - \eta \nabla_{\boldsymbol{x}} \log \mu(\bar{\boldsymbol{x}}_n))] + \mathcal{O}(\eta^2) \\
&\le (1+\eta)\mathbb{E}[f_{\delta,C}(\boldsymbol{x}_n - \bar{\boldsymbol{x}}_n)] + \left(1 + \frac{1}{\eta}\right) \eta^2 \mathbb{E}[f_{\delta,C}(\nabla_{\boldsymbol{x}} \log \mu(\boldsymbol{x}_n) - \nabla_{\boldsymbol{x}} \log \mu(\bar{\boldsymbol{x}}_n))] + \mathcal{O}(\eta^2) \\
&\le (1 + \mathcal{O}(\eta))\mathbb{E}[f_{\delta,C}(\boldsymbol{x}_n - \bar{\boldsymbol{x}}_n)] + \mathcal{O}(\eta^2),
\end{aligned} \tag{34}
$$

where the last two inequalities are respectively from Young's inequality $\|\boldsymbol{a} + \boldsymbol{b}\|^2 \leq (1+\eta)\|\boldsymbol{a}\|^2 + (1+1/\eta)\|\boldsymbol{b}\|^2$ for any $\eta > 0$, the Lipschitz continuity of $\nabla_{\boldsymbol{x}} \log \mu$, and the definition of $f_{\delta,C}$. Then, by recursively using the above inequality, we have

$$\mathbb{E}[f_{\delta,C}(\boldsymbol{x}_n - \bar{\boldsymbol{x}}_n)] \leq \mathcal{O}(\eta) \tag{35}$$

when $n \leq \mathcal{O}(1/\eta)$. By taking $\delta \to 0$, $C \to \infty$, and applying Fatou's Lemma [41], we get

$$\mathbb{E}\left[\|\boldsymbol{x}_n - \bar{\boldsymbol{x}}_n\|^2\right] = \mathbb{E}\left[\varliminf_{\delta \to 0, C \to \infty} f_{\delta,C}(\boldsymbol{x}_n - \bar{\boldsymbol{x}}_n)\right] \leq \varliminf_{\delta \to 0, C \to \infty} \mathbb{E}\left[f_{\delta,C}(\boldsymbol{x}_n - \bar{\boldsymbol{x}}_n)\right] \leq \mathcal{O}(\eta). \tag{36}$$

Next, we should show the $\bar{\boldsymbol{x}}_n$ (so that $\boldsymbol{x}_n$) is a discretion of the following SDE (continuous Langevin dynamics). Let us define

$$d\hat{\boldsymbol{x}}_t = \nabla \log \mu(\hat{\boldsymbol{x}}_t)dt + \sqrt{2}dW_t. \tag{37}$$

For any given $\bar{\boldsymbol{x}}_0 = \hat{\boldsymbol{x}}_0$, similar to (53), we can prove

$$\begin{aligned}
\mathbb{E}\left[\left\|\hat{\boldsymbol{x}}_{(n+1)\eta} - \bar{\boldsymbol{x}}_{n+1}\right\|^2\right] &\leq \mathbb{E}\left[\left\|\hat{\boldsymbol{x}}_{n\eta} + \eta\nabla\log\mu(\hat{\boldsymbol{x}}_{n\eta}) - \bar{\boldsymbol{x}}_{n+1}\right\|^2\right] + \mathcal{O}(\eta^2) \\
&\leq (1+\eta)\mathbb{E}\left[\|\hat{\boldsymbol{x}}_{n\eta} - \bar{\boldsymbol{x}}_n\|^2\right] + \left(1 + \frac{1}{\eta}\right)\mathbb{E}\left[\|\nabla\log\mu(\hat{\boldsymbol{x}}_{n\eta}) - \nabla\log\mu(\bar{\boldsymbol{x}}_n)\|^2\right] + \mathcal{O}(\eta^2) \\
&\leq (1+\eta)\mathbb{E}\left[\|\hat{\boldsymbol{x}}_{n\eta} - \bar{\boldsymbol{x}}_n\|^2\right] + L_2^2\eta^2\left(1 + \frac{1}{\eta}\right)\mathbb{E}\left[\|\hat{\boldsymbol{x}}_{n\eta} - \bar{\boldsymbol{x}}_n\|^2\right] + \mathcal{O}(\eta^2) \\
&= (1 + \mathcal{O}(\eta))\mathbb{E}\left[\|\hat{\boldsymbol{x}}_{n\eta} - \bar{\boldsymbol{x}}_n\|^2\right] + \mathcal{O}(\eta^2).
\end{aligned} \tag{38}$$

By iteratively applying this inequality, we get $\mathbb{E}[\|\hat{\boldsymbol{x}}_{n\eta} - \boldsymbol{x}_n\|^2] \leq \mathcal{O}(\eta)$ for any $n \leq \mathcal{O}(1/\eta)$. Thus, by the triangle inequality

$$\mathbb{E}\left[\|\hat{\boldsymbol{x}}_{n\eta} - \boldsymbol{x}_n\|^2\right] \leq 2\mathbb{E}\left[\|\hat{\boldsymbol{x}}_{n\eta} - \bar{\boldsymbol{x}}_n\|^2\right] + 2\mathbb{E}\left[\|\bar{\boldsymbol{x}}_n - \boldsymbol{x}_n\|^2\right] \leq \mathcal{O}(\eta). \tag{39}$$

Thus, we prove our conclusion by applying Lemma 1 to (37). $\qquad\square$

**Remark 3.** *It is worthy to note that during our proof, we introduce the auxiliary sequence $\{\bar{\boldsymbol{x}}_n\}$ which is the discrete Langevin dynamics. We construct its continuous counterpart instead of $\boldsymbol{x}_n$, because the drift term of it is $\nabla_{\boldsymbol{x}} \log \mu$ which has verifiable continuity. However, if we directly analyze $\boldsymbol{x}_n$ with drift term $\nabla_{\boldsymbol{x}} \log \mu / \pi_n$, its continuity is non-verifiable.*

**Theorem 1.** *[[9]] Let $\pi_t$ follows the Riemannian gradient flow (8), then for any $T > 0$, we have*

$$\frac{1}{T}\int_0^T \|\mathrm{grad}D_{KL}(\pi_t \parallel \mu)\|_{\pi_t}^2 dt \leq \frac{D_{KL}(\pi_0 \parallel \mu)}{T}. \tag{9}$$

*Moreover, if the log-Sobolev inequality (5) (Riemannian PL inequality) is satisfied for $\mu$, then*

$$D_{KL}(\pi_t \parallel \mu) \leq e^{-2\gamma t}D_{KL}(\pi_0 \parallel \mu). \tag{10}$$

*Proof.* Similar to (28), we have

$$\frac{\partial}{\partial t}D_{KL}(\pi_t \parallel \mu) = \mathrm{Dev}D_{KL}(\pi_t \parallel \mu)[-\mathrm{grad}D_{KL}(\pi_t \parallel \mu)] = -\|\mathrm{grad}D_{KL}(\pi_t \parallel \mu)\|_{\pi_t}^2. \tag{40}$$

Thus we know the Riemannian gradient flow resulted $\pi_t$ is monotonically decreased w.r.t. $t$ Taking integral and from the non-negativity of KL divergence implies the conclusion.

On the other hand, if the Riemannian PL inequality (129) holds with coefficient $\gamma$, then the above equality further implies

$$\frac{\partial}{\partial t}D_{KL}(\pi_t \parallel \mu) \leq -2\gamma D_{KL}(\pi_t \parallel \mu). \tag{41}$$

So that taking integral implies the second conclusion. $\qquad\square$

## B  Proofs in Section 5

**Proposition 3.** *The global optima of problem (11) is $\mu \propto \exp\left(\mathbb{E}_\xi\left[\log\mu_\xi\right]\right)$*

*Proof.* The result is directly proved by Langrange's multiplier theorem. Due to the definition of KL divergence and (11), the optimal $\pi$ should satisfies

$$\frac{\partial}{\partial \pi} \mathcal{L}(\pi) = \frac{\partial}{\partial \pi} \left\{ \mathbb{E}_\xi \left[ D_{KL}(\pi \parallel \mu_\xi) \right] - \lambda \left( \int \pi(\boldsymbol{x}) d\boldsymbol{x} - 1 \right) \right\} = 0, \tag{42}$$

which results in

$$\int p(\xi) \int (1 + \log \pi(\boldsymbol{x})) - \log \mu_\xi(\boldsymbol{x}) - \lambda d\boldsymbol{x} d\xi = 0 \tag{43}$$

for some $\lambda > 0$, where $p(\xi)$ is the density of $\xi$. Change the order of integral, we know that for any $\boldsymbol{x}$,

$$\log \pi(\boldsymbol{x}) + (1 - \lambda) = \int p(\xi) \log \mu_\xi(\boldsymbol{x}) d\xi, \tag{44}$$

which indicates our conclusion under the condition of $\int \pi(\boldsymbol{x}) d\boldsymbol{x} = 1$. $\qquad\square$

## B.1 Proofs of Proposition 4

**Proposition 4.** *Under Assumption 2, let $f_\xi(\pi) = D_{KL}(\pi \parallel \mu_\xi)$, the discrete Riemannian SGD Algorithm 1 with $1 \le n \le \mathcal{O}(\lfloor 1/\eta \rfloor)$ approximates the continuous Riemannian stochastic gradient flow*

$$\frac{\partial}{\partial t} \pi_t = \nabla \cdot \left[ \pi_t \left( \nabla \log \frac{d\pi_t}{d\mu} - \frac{\eta}{2} \nabla \cdot \Sigma_{\mathrm{SGD}} - \frac{\eta}{2} \Sigma_{\mathrm{SGD}} \nabla \log \pi_t \right) \right], \tag{12}$$

*by $\mathbb{E}[\|\boldsymbol{x}_n - \hat{\boldsymbol{x}}_{n\eta}\|^2] \le \mathcal{O}(\eta)$, where $\boldsymbol{x}_n \sim \pi_n$ in Algorithm 1, $\hat{\boldsymbol{x}}_{n\eta} \sim \pi_{n\eta}$ in (12), for $\boldsymbol{x}_0 = \hat{\boldsymbol{x}}_0$. Here*

$$\Sigma_{\mathrm{SGD}}(\boldsymbol{x}) = \mathbb{E}_\xi [ (\nabla \log \mu_\xi(\boldsymbol{x}) - \nabla \mathbb{E}_\xi [\log \mu_\xi(\boldsymbol{x})]) (\nabla \log \mu_\xi(\boldsymbol{x}) - \nabla \mathbb{E}_\xi [\log \mu_\xi(\boldsymbol{x})])^\top ]. \tag{13}$$

*Proof.* Similar to the proof of Proposition 2, we can show that for

$$\bar{\boldsymbol{x}}_{n+1} = \bar{\boldsymbol{x}}_n + \eta \nabla \log \mu_{\xi_n}(\bar{\boldsymbol{x}}_n) + \sqrt{2\eta} \epsilon_n, \tag{45}$$

with $\epsilon_n \sim \mathcal{N}(0, \boldsymbol{I})$, we have $\boldsymbol{x}_n$ approximates $\bar{\boldsymbol{x}}_n$. That says, similar to (32), when $\boldsymbol{x}_0 = \bar{\boldsymbol{x}}_0$, we can prove

$$\mathbb{E}\left[ \|\boldsymbol{x}_n - \bar{\boldsymbol{x}}_n\|^2 \right] \le \mathcal{O}(\eta^2) \tag{46}$$

Next, we show $\bar{\boldsymbol{x}}_n$ in (45) is the discretion of stochastic differential equation

$$\begin{aligned}
d\hat{\boldsymbol{x}}_t &= \nabla \mathbb{E}_\xi \left[ \log \mu_\xi(\hat{\boldsymbol{x}}_t) \right] + \left( \sqrt{\eta} \Sigma_{\mathrm{SGD}}^{\frac{1}{2}}(\hat{\boldsymbol{x}}_t), \sqrt{2} \boldsymbol{I} \right) dW_t \\
&= \boldsymbol{b}(\hat{\boldsymbol{x}}_t) + \left( \sqrt{\eta} \Sigma_{\mathrm{SGD}}^{\frac{1}{2}}(\hat{\boldsymbol{x}}_t), \sqrt{2} \boldsymbol{I} \right) dW_t,
\end{aligned} \tag{47}$$

where $\Sigma_{\mathrm{SGD}}(\hat{\boldsymbol{x}}_t)$ is the covariance matrix

$$\Sigma_{\mathrm{SGD}}(\hat{\boldsymbol{x}}_t) = \mathbb{E}_\xi \left[ (\nabla \log \mu_\xi(\hat{\boldsymbol{x}}_t) - \nabla \mathbb{E}_\xi [\log \mu_\xi(\hat{\boldsymbol{x}}_t)]) (\nabla \log \mu_\xi(\hat{\boldsymbol{x}}_t) - \nabla \mathbb{E}_\xi [\log \mu_\xi(\hat{\boldsymbol{x}}_t)])^\top \right]. \tag{48}$$

To check this, for any test function $f \in C^2$ with bounded gradient and Hessian, due to $\bar{\boldsymbol{x}}_0 = \hat{\boldsymbol{x}}_0$, Dynkin's formula [32], we have

$$\begin{aligned}
\mathbb{E}\left[ f(\hat{\boldsymbol{x}}_{(n+1)\eta}) \right] &= \mathbb{E}\left[ f(\hat{\boldsymbol{x}}_{n\eta}) \right] + \int_{n\eta}^{(n+1)\eta} \mathbb{E}\left[ \mathcal{L} f(\hat{\boldsymbol{x}}_t) \right] dt \\
&= \mathbb{E}\left[ f(\hat{\boldsymbol{x}}_{n\eta}) \right] + \eta \mathbb{E}\left[ \mathcal{L} f(\hat{\boldsymbol{x}}_{n\eta}) \right] + \frac{1}{2} \int_{n\eta}^{(n+1)\eta} \int_{n\eta}^{t} \mathbb{E}\left[ \mathcal{L}^2 f(\hat{\boldsymbol{x}}_s) \right] ds dt \\
&= \mathbb{E}\left[ f(\hat{\boldsymbol{x}}_{n\eta}) \right] + \eta \mathbb{E}\left[ \mathcal{L} f(\hat{\boldsymbol{x}}_{n\eta}) \right] + \mathcal{O}(\eta^2),
\end{aligned} \tag{49}$$

where the last equality is due to the Lipschitz continuity of $\nabla \log \mu_\xi(\boldsymbol{x})$ indicates $\nabla^2 \log \mu_\xi(\boldsymbol{x})$ have an upper bounded spectral norm, and $\nabla \log \mu_\xi(\boldsymbol{x})$ itself is upper bounded. Noting that

$$\mathbb{E}[\mathcal{L} f(\hat{\boldsymbol{x}}_{n\eta})] = \mathbb{E}_{\hat{\boldsymbol{x}}_{n\eta}} \left[ \langle \boldsymbol{b}(\hat{\boldsymbol{x}}_{n\eta}), \nabla f(\hat{\boldsymbol{x}}_{n\eta}) \rangle \right] + \frac{\eta}{2} \mathbb{E}\left[ \mathrm{tr}\left( \Sigma_{\mathrm{SGD}}(\hat{\boldsymbol{x}}_{n\eta}) \nabla^2 f(\hat{\boldsymbol{x}}_{n\eta}) \right) \right] + \mathbb{E}\left[ \Delta f(\hat{\boldsymbol{x}}_{n\eta}) \right]. \tag{50}$$

Plugging these into (49), we get

$$\begin{aligned}
\mathbb{E}\left[ f(\hat{\boldsymbol{x}}_{(n+1)\eta}) \right] &= \mathbb{E}\left[ f(\hat{\boldsymbol{x}}_{n\eta}) \right] + \mathbb{E}_{\hat{\boldsymbol{x}}_{n\eta}} \left[ \langle \boldsymbol{b}(\hat{\boldsymbol{x}}_{n\eta}), \nabla f(\hat{\boldsymbol{x}}_{n\eta}) \rangle \right] + \frac{\eta}{2} \mathbb{E}\left[ \mathrm{tr}\left( \Sigma_{\mathrm{SGD}}(\hat{\boldsymbol{x}}_{n\eta}) \nabla^2 f(\hat{\boldsymbol{x}}_{n\eta}) \right) \right] \\
&\quad + \mathbb{E}\left[ \Delta f(\hat{\boldsymbol{x}}_{n\eta}) \right] + \mathcal{O}(\eta^2).
\end{aligned} \tag{51}$$

On the other hand, we can similarly prove that

$$\mathbb{E}\left[f(\hat{\boldsymbol{x}}_{n\eta} + \eta\nabla\log\mu_{\xi_n}(\hat{\boldsymbol{x}}_{n\eta}) + \sqrt{2\eta}\boldsymbol{\epsilon}_n)\right] = \mathbb{E}\left[f(\hat{\boldsymbol{x}}_{n\eta})\right] + \eta\mathbb{E}[\langle\boldsymbol{b}(\hat{\boldsymbol{x}}_{n\eta}), \nabla f(\hat{\boldsymbol{x}}_{n\eta})\rangle] + \eta\mathbb{E}[\Delta f(\hat{\boldsymbol{x}}_{n\eta})]$$

$$+ \frac{\eta^2}{2}\mathbb{E}\left[\operatorname{tr}\left[\left(\Sigma_{\mathrm{SGD}}(\hat{\boldsymbol{x}}_{n\eta}) + \boldsymbol{b}(\hat{\boldsymbol{x}}_{n\eta})\boldsymbol{b}^\top(\hat{\boldsymbol{x}}_{n\eta})\right)\nabla^2 f(\hat{\boldsymbol{x}}_{n\eta})\right]\right] + \mathcal{O}(\eta^3). \tag{52}$$

Thus we get

$$\sup_{\boldsymbol{x}}\left|\mathbb{E}^{\boldsymbol{x}}\left[f(\hat{\boldsymbol{x}}_{(n+1)\eta})\right] - \mathbb{E}^{\boldsymbol{x}}\left[f(\hat{\boldsymbol{x}}_{n\eta} + \eta\nabla\log\mu_{\xi_n}(\hat{\boldsymbol{x}}_{n\eta}) + \sqrt{2\eta}\boldsymbol{\epsilon}_n)\right]\right| = \mathcal{O}(\eta^2), \tag{53}$$

due to the Lipschitz continuity of $\log\mu_\xi$, where $\mathbb{E}^{\boldsymbol{x}}[f(\hat{\boldsymbol{x}}_t)] = \mathbb{E}[f(\hat{\boldsymbol{x}}_t) \mid \hat{\boldsymbol{x}}_0 = \boldsymbol{x}]$. Let $f_{\delta,C}(\boldsymbol{x})$ be the ones in (33), then similar to (34),

$$\mathbb{E}\left[f_{\delta,C}\left(\hat{\boldsymbol{x}}_{(n+1)\eta} - \bar{\boldsymbol{x}}_{n+1}\right)\right] \le \mathbb{E}\left[f_{\delta,C}\left(\hat{\boldsymbol{x}}_{n\eta} + \eta\nabla\log\mu_{\xi_n}(\hat{\boldsymbol{x}}_{n\eta}) + \sqrt{2\eta}\boldsymbol{\epsilon}_n - \bar{\boldsymbol{x}}_{n+1}\right)\right] + \mathcal{O}(\eta^2)$$

$$= \mathbb{E}\left[f_{\delta,C}\left(\hat{\boldsymbol{x}}_{n\eta} - \bar{\boldsymbol{x}}_n + \eta\nabla\log\mu_{\xi_n}(\hat{\boldsymbol{x}}_{n\eta}) - \eta\nabla\log\mu_{\xi_n}(\bar{\boldsymbol{x}}_n)\right)\right] + \mathcal{O}(\eta^2)$$

$$\le (1 + \mathcal{O}(\eta))\mathbb{E}\left[f_{\delta,C}\left(\hat{\boldsymbol{x}}_{n\eta} - \bar{\boldsymbol{x}}_n\right)\right] + \left(1 + \frac{1}{\eta}\right)\eta^2\mathbb{E}\left[f_{\delta,C}\left(\nabla_{\boldsymbol{x}}\log\mu_{\xi_n}(\hat{\boldsymbol{x}}_{n\eta}) - \nabla_{\boldsymbol{x}}\log\mu_{\xi_n}(\bar{\boldsymbol{x}}_n)\right)\right] + \mathcal{O}(\eta^2)$$

$$\le (1 + \mathcal{O}(\eta))\mathbb{E}\left[f_{\delta,C}\left(\hat{\boldsymbol{x}}_{n\eta} - \bar{\boldsymbol{x}}_n\right)\right] + \mathcal{O}(\eta^2)$$

$$\le \cdots$$

$$\le \mathcal{O}(\eta). \tag{54}$$

The above inequality holds for any $n \le 1/\eta$. Similar to (36), by taking $\delta \to 0, C \to \infty$, and applying Fatou's Lemma, we get

$$\mathbb{E}\left[\|\hat{\boldsymbol{x}}_{n\eta} - \bar{\boldsymbol{x}}_n\|^2\right] \le \mathcal{O}(\eta). \tag{55}$$

Then combining (46) with the above inequality, and by triangle inequality, we have

$$\mathbb{E}\left[\|\boldsymbol{x}_n - \hat{\boldsymbol{x}}_{n\eta}\|^2\right] \le \mathcal{O}(\eta). \tag{56}$$

Thus we prove our conclusion by combining Lemma 1. $\qquad\square$

## B.2 Convergence of (Riemannian) SGD Flow

In this subsection, we first give proof of the convergence rate of the SGD flow in Euclidean space, by transferring it into its corresponding stochastic ordinary equation.

Similar to Proposition 4, we can prove, in Euclidean space, the SGD flow of minimizing $F(\boldsymbol{x}) = \mathbb{E}_\xi[f_\xi(\boldsymbol{x})]$ takes the form of

$$d\boldsymbol{x}_t = -\nabla F(\boldsymbol{x}_t) + \sqrt{\eta}\Sigma_{\mathrm{SGD}}(\boldsymbol{x}_t)^{\frac{1}{2}}dW_t, \tag{57}$$

with $\Sigma_{\mathrm{SGD}}(\boldsymbol{x}_t) = \mathbb{E}_\xi\left[(\nabla f_\xi(\boldsymbol{x}_t) - \mathbb{E}_\xi[f_\xi(\boldsymbol{x})])(\nabla f_\xi(\boldsymbol{x}_t) - \mathbb{E}_\xi[f_\xi(\boldsymbol{x})])^\top\right]$. Then by Lemma 1, it's corresponded stochastic ordinary equation is

$$d\boldsymbol{x}_t = -\nabla F(\boldsymbol{x}_t) - \frac{\eta}{2}\nabla\cdot\Sigma_{\mathrm{SGD}}(\boldsymbol{x}_t) - \frac{\eta}{2}\Sigma_{\mathrm{SGD}}\nabla\log\pi_t(\boldsymbol{x}_t)dt. \tag{58}$$

Before providing our theorem, we clarify the definition of our computational complexity to continuous optimization methods. Owing to the connection between the continuous method and its discrete counterpart as in Proposition 2, 4, and 5. It requires $T$ discrete update steps to arrive $\hat{\boldsymbol{x}}_{\eta T}$. Therefore, the computational complexity of running $T$ discrete steps is said to be the computational complexity of continuous optimization methods measured under $\hat{\boldsymbol{x}}_{\eta T}$.

Next, let us check the convergence rates of (58) for general non-convex optimization or with PL inequality. It worth noting that for this problem, the convergence rate is measured by $\mathbb{E}[\|\nabla F(\boldsymbol{x}_t)\|^2]$ or $\mathbb{E}[F(\boldsymbol{x}_t)] - \inf_{\boldsymbol{x}} F(\boldsymbol{x})$, with/without PL inequality.

**Theorem 4.** *Let $\boldsymbol{x}_t$ defined in (58), then if $\mathbb{E}\left[\operatorname{tr}\left(\Sigma_{\mathrm{SGD}}(\boldsymbol{x}_t)\nabla^2 F(\boldsymbol{x}_t)\right)\right] \le \sigma^2$ [10] by taking $\eta = \sqrt{\frac{\mathbb{E}[F(\boldsymbol{x}_0) - \inf_{\boldsymbol{x}} F(\boldsymbol{x})]}{T\sigma^2}}$, we have*

$$\frac{1}{\eta T}\int_0^{\eta T}\mathbb{E}_{\boldsymbol{x}_t}\left[\|\nabla F(\boldsymbol{x}_t)\|^2\right]dt \le \frac{2(F(\boldsymbol{x}_0) - \inf_{\boldsymbol{x}} F(\boldsymbol{x}))}{\eta T}, \tag{59}$$

---

[10]This can be satisfied when $f_\xi(\boldsymbol{x})$ and its gradient are Lipschitz continuous. Because we have $\mathbb{E}[\operatorname{tr}(\Sigma_{\mathrm{SGD}}(\boldsymbol{x}_t)\nabla^2 F(\boldsymbol{x}_t))] \le \lambda_{\max}(\nabla^2 F(\boldsymbol{x}_t))\mathbb{E}[\operatorname{tr}(\Sigma_{\mathrm{SGD}}(\boldsymbol{x}_t))]$ due to the definition of $\Sigma_{\mathrm{SGD}}$.

On the other hand, if $F(\boldsymbol{x}_t)$ satisfies PL inequality (128), by taking $\eta = 1/\gamma T^\alpha$ with $0 < \alpha < 1$, we have

$$\mathbb{E}\left[F(\boldsymbol{x}_{\eta T}) - \inf_{\boldsymbol{x}} F(\boldsymbol{x})\right] \leq \frac{\sigma^2}{\gamma T^\alpha}. \tag{60}$$

Besides that, if $F(\boldsymbol{x})$ is in the form of finite sum, the computational complexity is of order $\mathcal{O}(\epsilon^{-2})$ to make $\mathbb{E}_{\boldsymbol{x}_t}[\|\nabla F(\boldsymbol{x}_t)\|^2] \leq \epsilon$ for some $t$, and $\mathcal{O}(\epsilon^{-1/\alpha})$ to make $\mathbb{E}_{\boldsymbol{x}_t}[F(\boldsymbol{x}_t) - \inf_{\boldsymbol{x}} F(\boldsymbol{x})] \leq \epsilon$ under PL inequality.

*Proof.* Due to the definition of $\boldsymbol{x}_t$,

$$\frac{\partial F(\boldsymbol{x}_t)}{\partial t} = \left\langle \nabla F(\boldsymbol{x}_t), -\nabla F(\boldsymbol{x}_t) - \frac{\eta}{2} \nabla \cdot \Sigma_{\text{SGD}}(\boldsymbol{x}_t) - \frac{\eta}{2} \Sigma_{\text{SGD}} \nabla \log \pi_t(\boldsymbol{x}_t) \right\rangle. \tag{61}$$

On the other hand, for $\boldsymbol{x}_t \sim \pi_t$,

$$\mathbb{E}_{\boldsymbol{x}_t}\left[\langle \nabla F(\boldsymbol{x}_t), \Sigma_{\text{SGD}}(\boldsymbol{x}_t) \nabla \log \pi_t(\boldsymbol{x}_t) \rangle\right] = \int \langle \nabla F(\boldsymbol{x}_t), \Sigma_{\text{SGD}}(\boldsymbol{x}_t) \nabla \pi_t(\boldsymbol{x}_t) \rangle \, d\boldsymbol{x}$$
$$= -\mathbb{E}_{\boldsymbol{x}_t}\left[\langle \nabla F(\boldsymbol{x}_t), \nabla \cdot \Sigma_{\text{SGD}}(\boldsymbol{x}_t) \rangle + \text{tr}\left(\Sigma_{\text{SGD}}(\boldsymbol{x}_t) \nabla^2 F(\boldsymbol{x}_t)\right)\right]. \tag{62}$$

Plugging this into (61), we get

$$\frac{\partial \mathbb{E}\left[F(\boldsymbol{x}_t)\right]}{\partial t} = -\mathbb{E}\left[\|\nabla F(\boldsymbol{x}_t)\|^2\right] + \frac{\eta}{2} \mathbb{E}\left[\text{tr}\left(\Sigma_{\text{SGD}}(\boldsymbol{x}_t) \nabla^2 F(\boldsymbol{x}_t)\right)\right] \leq -\mathbb{E}\left[\|\nabla F(\boldsymbol{x}_t)\|^2\right] + \eta \sigma^2. \tag{63}$$

Thus

$$\frac{1}{\eta T} \int_0^{\eta T} \mathbb{E}_{\boldsymbol{x}_t}[\|\nabla F(\boldsymbol{x}_t)\|^2] dt \leq \frac{\mathbb{E}\left[F(\boldsymbol{x}_0) - F(\boldsymbol{x}_{T\eta})\right]}{\eta T} + \eta \sigma^2$$
$$= 2\sqrt{\frac{\mathbb{E}\left[F(\boldsymbol{x}_0) - F(\boldsymbol{x}_{T\eta})\right] \sigma^2}{T}} \tag{64}$$
$$= \frac{2\mathbb{E}\left[F(\boldsymbol{x}_0) - F(\boldsymbol{x}_{T\eta})\right]}{\eta T},$$

by taking $\eta = \sqrt{\frac{\mathbb{E}[F(\boldsymbol{x}_0) - \inf_{\boldsymbol{x}} F(\boldsymbol{x})]}{T\sigma^2}}$. Then we prove our first conclusion.

Next, let us consider the global convergence rate under PL inequality. By applying (128) to (63), we get

$$\frac{\partial \mathbb{E}\left[F(\boldsymbol{x}_t)\right] - \inf_{\boldsymbol{x}} F(\boldsymbol{x})}{\partial t} \leq -2\gamma(\mathbb{E}\left[F(\boldsymbol{x}_t)\right] - \inf_{\boldsymbol{x}} F(\boldsymbol{x})) + \eta \sigma^2, \tag{65}$$

which implies

$$\mathbb{E}\left[F(\boldsymbol{x}_{\eta T})\right] - \inf_{\boldsymbol{x}} F(\boldsymbol{x}) \leq e^{-2\gamma\eta T}\left(\mathbb{E}\left[F(\boldsymbol{x}_0)\right] - \inf_{\boldsymbol{x}} F(\boldsymbol{x})\right) + \eta \sigma^2 \left(1 - e^{-2\gamma\eta T}\right), \tag{66}$$

by Gronwall inequality. Thus by taking $\eta = 1/\gamma T^\alpha$ with $0 < \alpha < 1$, we get second conclusion.

Under finite sum objective, when $\sqrt{T} = \mathcal{O}(\eta T) = \mathcal{O}(\epsilon^{-1})$, we have $\min_{0 \leq t \leq \eta T} \mathbb{E}_{\boldsymbol{x}_t}[\|\nabla F(\boldsymbol{x}_t)\|^2] \leq \epsilon$. Thus, due to Proposition 4, it takes $M = \mathcal{O}(\eta T/\eta) = \mathcal{O}(\epsilon^{-2})$ steps in Algorithm 1 to get the "$\epsilon$-stationary point".

Furthermore, under PL inequality, due to (66), it takes $T = \mathcal{O}(\epsilon^{-1/\alpha})$ steps to make $\mathbb{E}\left[F(\boldsymbol{x}_{\eta T})\right] - \inf_{\boldsymbol{x}} F(\boldsymbol{x}) \leq \epsilon$, which results in computational complexity of order $\mathcal{O}(\epsilon^{-1/\alpha})$. $\qquad\square$

The convergence rate of Riemannian SGD flow can be similarly proven as in the above theorem. Next, let us check it. We need a lemma termed as von Neumann's trace inequality [46] to prove Theorem 4.

**Lemma 2** (von Neumann's trace inequality). *For systematic matrices $\boldsymbol{A}$ and $\boldsymbol{B}$, let $\{\lambda_i(\boldsymbol{A})\}$ and $\{\lambda_i(\boldsymbol{B})\}$ respectively be their eigenvalues with descending orders. Then*

$$\text{tr}(\boldsymbol{A}\boldsymbol{B}) \leq \sum_{i=1}^n \lambda_i(\boldsymbol{A})\lambda_i(\boldsymbol{B}). \tag{67}$$

**Theorem 2.** *Let $\pi_t$ follows the Riemannian SGD flow (12) and $\mu$ defined in Proposition (3). Under Assumption 2, if $T \geq \frac{64L_1^4 D_{KL}(\pi_0\|\mu)}{4dL_1^2 L_2 + (d+1)^2 L_2^2}$, then by taking $\eta = \sqrt{\frac{D_{KL}(\pi_0\|\mu)}{T(4dL_1^2 L_2 + (d+1)^2 L_2^2)}}$, we have*

$$\frac{1}{\eta T} \int_0^{\eta T} \|\text{grad} D_{KL}(\pi_t \| \mu)\|_{\pi_t}^2 dt \leq \frac{4D_{KL}(\pi_0 \| \mu)}{\eta T} = \mathcal{O}\left(\frac{1}{\sqrt{T}}\right). \tag{17}$$

*Besides that, if* (5) *is satisfied for* $\mu$, $\eta = 1/\gamma T^{\alpha}$ *with* $0 < \alpha < 1$, *and* $T \geq \left(8L_1^2/\gamma\right)^{1/\alpha}$, *then*

$$D_{KL}(\pi_{\eta T} \parallel \mu) \leq \frac{1}{\gamma T^{\alpha}} \left[4dL_1^2 L_2 + (d+1)^2 L_2^2\right] = \mathcal{O}\left(\frac{1}{T^{\alpha}}\right). \tag{18}$$

*Proof.* During the proof, we borrow the notations of $H_1, H_2, H_3$ in Lemma 3. Next, we prove the results as in Theorem 4. That is

$$\frac{\partial}{\partial t} D_{KL}(\pi_t \parallel \mu) = \left\langle \mathrm{grad} D_{KL}(\pi_t \parallel \mu), -\mathrm{grad} D_{KL}(\pi_t \parallel \mu) + \frac{\eta}{2}\nabla \cdot \Sigma_{\mathrm{SGD}} + \frac{\eta}{2}\Sigma_{\mathrm{SGD}}\nabla \log \pi_t \right\rangle_{\pi_t}$$

$$= -\|\mathrm{grad} D_{KL}(\pi_t \parallel \mu)\|^2 + \left\langle \mathrm{grad} D_{KL}(\pi_t \parallel \mu), \frac{\eta}{2}\nabla \cdot \Sigma_{\mathrm{SGD}} + \frac{\eta}{2}\Sigma_{\mathrm{SGD}}\nabla \log \pi_t \right\rangle_{\pi_t}. \tag{68}$$

To begin with, we have

$$\left\langle \mathrm{grad} D_{KL}(\pi_t \parallel \mu), \frac{\eta}{2}\nabla \cdot \Sigma_{\mathrm{SGD}} \right\rangle_{\pi_t} = \int \left\langle \nabla \log \frac{d\pi_t}{d\mu}, \frac{\eta}{2}\nabla \cdot \Sigma_{\mathrm{SGD}} \right\rangle d\pi_t$$

$$\leq \frac{\eta H_3}{2}\|\mathrm{grad} D_{KL}(\pi_t \parallel \mu)\|_{\pi_t}^2 + \frac{\eta}{8H_3}\mathbb{E}_{\pi_t}\left[\|\nabla \cdot \Sigma_{\mathrm{SGD}}\|^2\right] \tag{69}$$

$$\leq \frac{\eta H_3}{2}\|\mathrm{grad} D_{KL}(\pi_t \parallel \mu)\|_{\pi_t}^2 + \frac{\eta H_2^2}{8H_3},$$

where the first inequality is due to Young's inequality and last one is from the Lemma 3. Then, we have

$$\left\langle \mathrm{grad} D_{KL}(\pi_t \parallel \mu), \frac{\eta}{2}\Sigma_{\mathrm{SGD}}\nabla \log \pi_t \right\rangle_{\pi_t} = \int \left\langle \nabla \log \frac{d\pi_t}{d\mu}, \frac{\eta}{2}\Sigma_{\mathrm{SGD}}\nabla \log \pi_t \right\rangle d\pi_t$$

$$= \int \left\langle \nabla \log \pi_t, \frac{\eta}{2}\Sigma_{\mathrm{SGD}}\nabla \log \pi_t \right\rangle d\pi_t - \int \left\langle \nabla \log \mu, \frac{\eta}{2}\Sigma_{\mathrm{SGD}}\nabla \log \pi_t \right\rangle d\pi_t$$

$$\leq \frac{\eta H_3}{2}\int \|\nabla \log d\pi_t\|^2 \, d\pi_t - \int \left\langle \nabla \log \mu, \frac{\eta}{2}\Sigma_{\mathrm{SGD}}\nabla \log \pi_t \right\rangle d\pi_t \tag{70}$$

$$= \frac{\eta H_3}{2}\|\mathrm{grad} D_{KL}(\pi_t \parallel \mu)\|_{\pi_t}^2 - \int \left\langle \nabla \log \mu, \frac{\eta}{2}\Sigma_{\mathrm{SGD}}\nabla \log \pi_t \right\rangle d\pi_t$$

$$- \frac{\eta H_3}{2}\int \|\nabla \log \mu\|^2 \, d\pi_t + \frac{\eta H_3}{2}\int \left\langle \nabla \log \mu, \frac{\eta}{2}\nabla \log \pi_t \right\rangle d\pi_t.$$

In the r.h.s of the above inequality, the sum of the second and the forth terms can be bounded as

$$-\int \left\langle \nabla \log \mu, \frac{\eta}{2}\Sigma_{\mathrm{SGD}}\nabla \log \pi_t \right\rangle d\pi_t + \frac{\eta H_3}{2}\int \left\langle \nabla \log \mu, \frac{\eta}{2}\nabla \log \pi_t \right\rangle d\pi_t$$

$$= -\int \left\langle \nabla \log \mu, \frac{\eta}{2}\Sigma_{\mathrm{SGD}}\nabla \log \pi_t \right\rangle d\pi_t + \frac{\eta H_3}{2}\int \left\langle \nabla \log \mu, \frac{\eta}{2}\nabla \log \pi_t \right\rangle d\pi_t$$

$$- \int \left\langle \nabla \log \mu, \frac{\eta}{2}\nabla \cdot \Sigma_{\mathrm{SGD}} \right\rangle d\pi_t + \int \left\langle \nabla \log \mu, \frac{\eta}{2}\nabla \cdot \Sigma_{\mathrm{SGD}} \right\rangle d\pi_t$$

$$= \frac{\eta}{2}\int \mathrm{tr}\left(\nabla^2 \log \mu \Sigma_{\mathrm{SGD}}\right) d\pi_t - \frac{\eta H_3}{2}\int \mathrm{tr}\left(\nabla^2 \log \mu\right) d\pi_t \tag{71}$$

$$+ \int \left\langle \nabla \log \mu, \frac{\eta}{2}\nabla \cdot \Sigma_{\mathrm{SGD}} \right\rangle d\pi_t$$

$$\overset{a}{\leq} \frac{\eta H_1 H_3}{2} + \frac{\eta H_1 H_3}{2} + \frac{\eta H_3}{2}\int \|\nabla \log \mu\| \, d\pi_t$$

$$\leq \frac{\eta H_1 H_3}{2} + \frac{\eta H_1 H_3}{2} + \frac{\eta H_3}{2}\int \|\nabla \log \mu\|^2 \, d\pi_t + \frac{\eta H_2^2}{8H_3},$$

where the inequality $a$ is from Lemma 2, 3, and the semi-positive definite property of $\Sigma_{\mathrm{SGD}}$. By plugging (69), (70), and (71) into (68), we have

$$\frac{\partial}{\partial t} D_{KL}(\pi_t \parallel \mu) \leq -(1 - \eta H_3)\|\mathrm{grad} D_{KL}(\pi_t \parallel \mu)\|_{\pi_t}^2 + \eta H_1 H_3 + \frac{\eta H_2^2}{4H_3}, \tag{72}$$

due to the value of $\eta$ makes $\eta H_3 \leq \frac{1}{2}$. Taking integral w.r.t. $t$ as in (64) implies

$$\frac{1}{2\eta T} \int_0^{\eta T} \|\mathrm{grad} D_{KL}(\pi_t \| \mu)\|_{\pi_t}^2 \, dt \leq \frac{D_{KL}(\pi_0 \| \mu)}{\eta T} + \eta H_1 H_3 + \frac{\eta H_2^2}{4H_3}$$

$$= 2\sqrt{\frac{D_{KL}(\pi_0 \| \mu)}{T} \left(H_1 H_3 + \frac{H_2^2}{4H_3}\right)} \quad (73)$$

$$= 2 D_{KL}(\pi_0 \| \mu) \frac{1}{\eta T},$$

so that we prove our conclusion due to the value of $H_1, H_2, H_3$. The second result can be similarly obtained as in Theorem 4 by inequality

$$D_{KL}(\pi_{\eta T} \| \mu) \leq e^{-\gamma \eta T} D_{KL}(\pi_0 \| \mu) + \eta \left(H_1 H_3 + \frac{H_2^2}{4H_3}\right)\left(1 - e^{-\gamma \eta T}\right), \quad (74)$$

which is obtained by applying log-Sobolev inequality to (73) and Gronwall inequality. Thus, taking $\eta = 1/\gamma T^\alpha$ with $0 < \alpha < 1$ implies our conclusion. $\qquad \square$

Similar to the proof of Theorem 2, we can get the computational complexity of Riemannian GD and SGD flow, i.e., $\mathcal{O}(N\epsilon^{-1})$ and $\mathcal{O}(\epsilon^{-2})$ respectively for non-convex problem to arrive $\epsilon$-stationary point, but $\mathcal{O}(N\gamma^{-1}\log\epsilon^{-1})$ and $\mathcal{O}(\epsilon^{-1})$ under (log-Sobolev inequality) Riemannian PL inequality.

The following is the lemma implied by Assumption 2.

**Lemma 3.** *Under Assumption 2, it holds* $\mathbb{E}_{\pi_t}[\mathrm{tr}^+(\nabla^2 \log \mu)] \leq dL_2 = H_1$ [11], $\sup_{\boldsymbol{x}} \|\nabla \cdot \Sigma_{\mathrm{SGD}}\| \leq 4(d+1)L_1 L_2 = H_2$, *and* $\sup_{\boldsymbol{x}} \lambda_{\max}(\Sigma_{\mathrm{SGD}}(\boldsymbol{x})) \leq 4L_1^2 = H_3$.

*Proof.* Due to Assumption 2, we have

$$\mathbb{E}_{\pi_t}[\mathrm{tr}^+(\nabla^2 \log \mu)] = \mathbb{E}_{\pi_t}\left[\mathrm{tr}^+\left(\mathbb{E}_\xi\left[\nabla^2 \log \mu_\xi\right]\right)\right]$$

$$\leq \mathbb{E}_{\pi_t}\left[d\left\|\mathbb{E}_\xi \nabla^2 \log \mu_\xi\right\|\right] \quad (75)$$

$$\leq dL_2,$$

where $\| \cdot \|$ here is the spectral norm of matrix. On the other hand, we notice

$$\nabla \cdot \Sigma_{\mathrm{SGD}} = \mathbb{E}_\xi\left[\mathrm{tr}\left(\nabla^2 \log \mu_\xi - \mathbb{E}_\xi\left[\nabla^2 \log \mu_\xi\right]\right)\left(\nabla \log \mu_\xi - \mathbb{E}\left[\nabla \log \mu_\xi\right]\right)\right]$$

$$+ \mathbb{E}_\xi\left[\left(\nabla^2 \log \mu_\xi - \mathbb{E}_\xi\left[\nabla^2 \log \mu_\xi\right]\right)\left(\nabla \log \mu_\xi - \mathbb{E}\left[\nabla \log \mu_\xi\right]\right)\right]. \quad (76)$$

Then by Assumption 2,

$$\sup_{\boldsymbol{x}} \|\nabla \cdot \Sigma_{\mathrm{SGD}}(\boldsymbol{x})\| \leq 4dL_1 L_2 + 4L_1 L_2 = 4(d+1)L_1 L_2. \quad (77)$$

Finally, due to Assumption 2, we have

$$\sup_{\boldsymbol{x}} \lambda_{\max}(\Sigma_{\mathrm{SGD}}(\boldsymbol{x})) \leq 4L_1^2. \quad (78)$$

$\qquad \square$

# C  Proofs in Section 6

**Proposition 5.** *Under Assumption 2, let* $f_\xi(\pi) = D_{KL}(\pi \| \mu_\xi)$, *the discrete Riemannian SVRG Algorithm 2 with* $1 \leq n \leq \mathcal{O}(\lfloor 1/\eta \rfloor)$ *approximates the Riemannian SVRG flow*

$$\frac{\partial}{\partial t} \pi_t(\boldsymbol{x}) = \nabla \cdot \left[\pi_t(\boldsymbol{x})\left(\nabla \log \frac{d\pi_t}{d\mu}(\boldsymbol{x}) - \frac{\eta}{2\pi_t(\boldsymbol{x})} \int \pi_{t_i,t}(\boldsymbol{y}, \boldsymbol{x}) \nabla_{\boldsymbol{x}} \cdot \Sigma_{\mathrm{SVRG}}(\boldsymbol{y}, \boldsymbol{x}) d\boldsymbol{y}\right.\right.$$

$$\left.\left. - \frac{\eta}{2\pi_t(\boldsymbol{x})} \int \pi_{t_i,t}(\boldsymbol{y}, \boldsymbol{x}) \Sigma_{\mathrm{SVRG}}(\boldsymbol{y}, \boldsymbol{x}) \nabla_{\boldsymbol{x}} \log \pi_{t_i,t}(\boldsymbol{y}, \boldsymbol{x}) d\boldsymbol{y}\right)\right], \quad (24)$$

*for* $iM\eta = t_i \leq t \leq t_{i+1}$, $(\hat{\boldsymbol{x}}_{t_i}, \hat{\boldsymbol{x}}_t) \sim \pi_{t_i,t}$, $\hat{\boldsymbol{x}}_t \sim \pi_t$ *in (24), since we have* $\mathbb{E}[\|\boldsymbol{x}_n^i - \hat{\boldsymbol{x}}_{(iM+n)\eta}\|^2] \leq \mathcal{O}(\eta)$, *where* $\boldsymbol{x}_n^i \sim \pi_n^i$ *in Algorithm 2 for* $\boldsymbol{x}_0^0 = \hat{\boldsymbol{x}}_0$. *Here*

$$\Sigma_{\mathrm{SVRG}}(\boldsymbol{y}, \boldsymbol{x}) = \mathbb{E}_\xi\left[\left(\nabla \log \mu_\xi(\boldsymbol{x}) - \nabla \log \mu_\xi(\boldsymbol{y}) + \nabla \mathbb{E}_\xi\left[\log \mu_\xi(\boldsymbol{y})\right] - \nabla \mathbb{E}_\xi \log \mu_\xi(\boldsymbol{x})\right)\right.$$

$$\left(\nabla \log \mu_\xi(\boldsymbol{x}) - \nabla \log \mu_\xi(\boldsymbol{y}) + \nabla \mathbb{E}_\xi\left[\log \mu_\xi(\boldsymbol{y})\right] - \nabla \mathbb{E}_\xi \log \mu_\xi(\boldsymbol{x})\right)^\top\right]. \quad (25)$$

---
[11] $\mathrm{tr}^+(*)$ stands for the sum of absolute value of $*$'s eigenvalues.

*Proof.* The proof is similar to the one of Proposition 4. Firstly, we choose $\Gamma_{\pi_0^i}^{\pi_n^i}$ as the transportation that preserves the correlation between $\boldsymbol{x}_0^i$ and $\boldsymbol{x}_n^i$, which means that for any function $\boldsymbol{u} \in \mathcal{T}_{\pi_0^i}$, we have

$$\Gamma_{\pi_0^i}^{\pi_n^i}(\boldsymbol{u}(\boldsymbol{x}_0^i)) = \boldsymbol{u}(\boldsymbol{x}_n^i), \tag{79}$$

so that $\pi_0^i \times \pi_n^i$ is the union distribution of $(\boldsymbol{x}_0^i, \boldsymbol{x}_n^i)$ defined as below. Concretely, we know that for $0 \leq n \leq M-1, 0 \leq i \leq I-1$, the corresponded $\boldsymbol{x}_n^i$ of discrete SVRG in Algorithm 2 satisfies

$$\boldsymbol{x}_{n+1}^i = \boldsymbol{x}_n^i + \eta \left( \nabla \log \frac{d\mu_{\xi_n^i}}{d\pi_n^i}(\boldsymbol{x}_n^i) - \nabla \log \frac{d\mu_{\xi_n^i}}{d\pi_0^i}(\boldsymbol{x}_0^i) + \nabla \log \frac{d\mu}{d\pi_0^i}(\boldsymbol{x}_0^i) \right). \tag{80}$$

In the rest of this proof, we neglect the subscript $i$ to simplify the notations. Similar to Proposition 4, we can prove $\boldsymbol{x}_{n+1}$ is approximated by

$$\bar{\boldsymbol{x}}_{n+1} = \bar{\boldsymbol{x}}_n + \eta \left( \nabla \log \mu_{\xi_n}(\bar{\boldsymbol{x}}_n) - \nabla \log \mu_{\xi_n}(\boldsymbol{x}_0) + \nabla \log \mu(\boldsymbol{x}_0) \right) + \sqrt{2\eta}\boldsymbol{\epsilon}_n, \tag{81}$$

with $\boldsymbol{\epsilon}_n \sim \mathcal{N}(0, \boldsymbol{I})$ by noting the Lipchitz continuity of $\nabla \log \mu_\xi$. As in Proposition 4, the approximation error is similarly proven as

$$\mathbb{E}\left[ \|\bar{\boldsymbol{x}}_n - \boldsymbol{x}_n\|^2 \mid \boldsymbol{x}_0, \bar{\boldsymbol{x}}_0 \right] \leq \mathcal{O}(\eta) + \mathcal{O}(\|\boldsymbol{x}_0 - \bar{\boldsymbol{x}}_0\|^2). \tag{82}$$

Next, our goal is showing that the discrete dynamics $\bar{\boldsymbol{x}}_n$ is approximated by the following stochastic differential equation

$$d\hat{\boldsymbol{x}}_t = \nabla \mathbb{E}_\xi \left[ \log \mu_\xi(\hat{\boldsymbol{x}}_t) \right] dt + \left( \sqrt{\eta} \Sigma_{\mathrm{SVRG}}^{\frac{1}{2}}(\hat{\boldsymbol{x}}_{t_i}, \hat{\boldsymbol{x}}_t), \sqrt{2}\boldsymbol{I} \right) dW_t$$
$$= \boldsymbol{b}(\hat{\boldsymbol{x}}_t)dt + \left( \sqrt{\eta} \Sigma_{\mathrm{SVRG}}^{\frac{1}{2}}(\hat{\boldsymbol{x}}_{t_i}, \hat{\boldsymbol{x}}_t), \sqrt{2}\boldsymbol{I} \right) dW_t; \qquad iM\eta = t_i \leq t \leq t_{i+1} = (i+1)M\eta, \tag{83}$$

where $M$ is steps for each epoch, and $\Sigma_{\mathrm{SVRG}}^{\frac{1}{2}}(\hat{\boldsymbol{x}}_{t_i}, \boldsymbol{x}_t)$ is defined in (25). Similar to (49), for $(n+iM)\eta = t \in [t_i, t_{i+1}]$. We write $\hat{\boldsymbol{x}}_{(n+iM)\eta}$ as $\hat{\boldsymbol{x}}_{n\eta}$ in the rest of this proof to simplify the notation. We can prove that for any $f \in C^2$ with bounded gradient and Hessian under the condition of given $\hat{\boldsymbol{x}}_{t_i}$,

$$\mathbb{E}^{\hat{\boldsymbol{x}}_{t_i}}\left[ f(\hat{\boldsymbol{x}}_{(n+1)\eta}) \right] = \mathbb{E}^{\hat{\boldsymbol{x}}_{t_i}}\left[ f(\hat{\boldsymbol{x}}_{n\eta}) \right] + \eta \mathbb{E}^{\hat{\boldsymbol{x}}_{t_i}}\left[ \langle \boldsymbol{b}(\hat{\boldsymbol{x}}_{n\eta}), \nabla f(\hat{\boldsymbol{x}}_{n\eta}) \rangle \right] + \eta \mathbb{E}^{\hat{\boldsymbol{x}}_{t_i}}\left[ \Delta f(\hat{\boldsymbol{x}}_{n\eta}) \right]$$
$$+ \frac{\eta^2}{2} \mathbb{E}^{\hat{\boldsymbol{x}}_{t_i}}\left[ \mathrm{tr}\left[ (\Sigma_{\mathrm{SVRG}}(\hat{\boldsymbol{x}}_{t_i}, \hat{\boldsymbol{x}}_{n\eta})) \nabla^2 f(\hat{\boldsymbol{x}}_{n\eta}) \right] \right] + \mathcal{O}(\eta^2) \tag{84}$$

On the other hand

$$\mathbb{E}^{\hat{\boldsymbol{x}}_{t_i}}\left[ f \left( \hat{\boldsymbol{x}}_{n\eta} + \eta \left( \nabla \log \mu_{\xi_n}(\hat{\boldsymbol{x}}_{n\eta}) - \nabla \log \mu_{\xi_n}(\hat{\boldsymbol{x}}_{t_i}) + \nabla \log \mu(\hat{\boldsymbol{x}}_{t_i}) \right) \right) + \sqrt{2\eta}\boldsymbol{\epsilon}_n \right]$$
$$= \mathbb{E}^{\hat{\boldsymbol{x}}_{t_i}}\left[ f(\hat{\boldsymbol{x}}_{n\eta}) \right] + \eta \mathbb{E}^{\hat{\boldsymbol{x}}_{t_i}}\left[ \langle \boldsymbol{b}(\hat{\boldsymbol{x}}_{n\eta}), \nabla f(\hat{\boldsymbol{x}}_{n\eta}) \rangle \right] + \eta \mathbb{E}^{\hat{\boldsymbol{x}}_{t_i}}\left[ \Delta f(\hat{\boldsymbol{x}}_{n\eta}) \right] \tag{85}$$
$$+ \frac{\eta^2}{2} \mathbb{E}^{\hat{\boldsymbol{x}}_{t_i}}\left[ \mathrm{tr}\left[ \left( \Sigma_{\mathrm{SVRG}}(\hat{\boldsymbol{x}}_{t_i}, \hat{\boldsymbol{x}}_{n\eta}) + \boldsymbol{b}\boldsymbol{b}^\top \right) \nabla^2 f(\hat{\boldsymbol{x}}_{n\eta}) \right] \right] + \mathcal{O}(\eta^3),$$

where we use the fact that

$$\mathbb{E}^{\hat{\boldsymbol{x}}_{t_i}}\left[ \nabla \log \mu_{\xi_n}(\hat{\boldsymbol{x}}_{n\eta}) - \nabla \log \mu_{\xi_n}(\hat{\boldsymbol{x}}_{t_i}) + \nabla \log \mu(\hat{\boldsymbol{x}}_{t_i}) \right] = \mathbb{E}^{\hat{\boldsymbol{x}}_{t_i}}\left[ \nabla \log \mu_{\xi_n}(\hat{\boldsymbol{x}}_{n\eta}) \right] = \boldsymbol{b}(\hat{\boldsymbol{x}}_{n\eta}). \tag{86}$$

Then similar to (54) in the proof of Proposition 4, by taking $f$ as $f_{\delta,C}$ in (33), and combining (82), we prove

$$\mathbb{E}\left[ \|\hat{\boldsymbol{x}}_{n\eta} - \boldsymbol{x}_n\|^2 \mid \hat{\boldsymbol{x}}_{t_i}, \boldsymbol{x}_0 \right] \leq \mathcal{O}(\eta) + \mathcal{O}\left( \|\hat{\boldsymbol{x}}_{t_i} - \boldsymbol{x}_0\|^2 \right). \tag{87}$$

By noting that $\hat{\boldsymbol{x}}_0 = \boldsymbol{x}_0^0$, recursively using the above inequality over $i$, and taking expectation, we prove that

$$\mathbb{E}\left[ \left\| \hat{\boldsymbol{x}}_{(iM+n)\eta} - \boldsymbol{x}_n^i \right\|^2 \right] \leq \mathcal{O}(\eta). \tag{88}$$

On the other hand, we know that given $\hat{\boldsymbol{x}}_{t_i} = \boldsymbol{y}$, the conditional probability $\hat{\boldsymbol{x}}_t \mid \hat{\boldsymbol{x}}_{t_i}$ follows conditional density $\pi_{t|t_i}(\boldsymbol{x})$ satisfies

$$\frac{\partial}{\partial t}\pi_{t|t_i}(\boldsymbol{x} \mid \boldsymbol{y}) = \nabla \cdot \left[ \pi_{t|t_i}(\boldsymbol{x} \mid \boldsymbol{y}) \left( \nabla \log \frac{d\pi_t}{d\mu}(\boldsymbol{x}) - \frac{\eta}{2}\nabla_{\boldsymbol{x}} \cdot \Sigma_{\mathrm{SVRG}}(\boldsymbol{y}, \boldsymbol{x}) - \frac{\eta}{2}\Sigma_{\mathrm{SVRG}}(\boldsymbol{y}, \boldsymbol{x})\nabla_{\boldsymbol{x}} \log \pi_{t_i,t}(\boldsymbol{y}, \boldsymbol{x}) \right) \right]. \tag{89}$$

Then, multiplying $\pi_{t_i}(\boldsymbol{y})$ to the above equality, applying equality $\nabla_{\boldsymbol{x}} \log \pi_{t_i,t}(\boldsymbol{y}, \boldsymbol{x}) = \nabla_{\boldsymbol{x}} \log \pi_{t|t_i}(\boldsymbol{x} \mid \boldsymbol{y})$, and taking integral over $\boldsymbol{y}$ implies our conclusion. $\qquad\square$

## C.1 Convergence of Riemannian SVRG Flow

Similar to the results of Riemannian SGD in Section B.2, we first give the convergence rate of continuous SVRG flow (stochastic ODE) in the Euclidean space, which helps understanding the results in Wasserstein space. First, to minimize $F(\boldsymbol{x}) = 1/N \sum_{j=1}^{N} f_{\xi_j}(\boldsymbol{x})$, by generalizing the formulation in (24), the SVRG flow in Euclidean space is

$$d\boldsymbol{x}_t = -\nabla F(\boldsymbol{x}_t) - \frac{\eta}{2}\nabla \cdot \Sigma_{\mathrm{SVRG}}(\boldsymbol{x}_{t_i}, \boldsymbol{x}_t) - \frac{\eta}{2}\Sigma_{\mathrm{SVRG}}(\boldsymbol{x}_{t_i}, \boldsymbol{x}_t)\nabla \log \pi_t(\boldsymbol{x}_t)dt, \qquad t_i \leq t \leq t_{i+1}; \quad (90)$$

with $\Sigma_{\mathrm{SVRG}}$ defined as

$$\Sigma_{\mathrm{SVRG}}(\boldsymbol{y}, \boldsymbol{x}) = \mathbb{E}_{\xi, \boldsymbol{x}_{t_i}}\left[(\nabla f_\xi(\boldsymbol{x}) - \nabla f_\xi(\boldsymbol{y}) + \nabla F(\boldsymbol{y}) - \nabla F(\boldsymbol{x}))(\nabla f_\xi(\boldsymbol{x}) - \nabla f_\xi(\boldsymbol{y}) + \nabla F(\boldsymbol{y}) - \nabla F(\boldsymbol{x}))^\top\right],$$
$$(91)$$

and $\pi_t$ is the density of $\boldsymbol{x}_t$, $\xi$ is uniform distribution over $\{\xi_i\}$. Then the convergence rate and computational complexity of (90) is presented in the following Theorem.

**Theorem 5.** *For $\boldsymbol{x}_t$ in (90), learning rate $\eta = \mathcal{O}(N^{-2/3})$, $\Delta = t_1 - t_0 = \cdots = t_I - t_{I-1} = \mathcal{O}(1/\sqrt{\eta})$, and $\eta T = I\Delta$, if $f_\xi(\boldsymbol{x})$ is Lipschitz continuous with coefficient $L_1$ and has Lipschitz continuous gradient with coefficient $L_2$, then*

$$\frac{1}{\eta T}\sum_{i=1}^{I}\int_{t_i}^{t_{i+1}}\mathbb{E}\left[\|\nabla F(\boldsymbol{x}_t)\|^2\right]dt \leq \frac{2(\mathbb{E}\left[F(\boldsymbol{x}_0)\right] - \inf_{\boldsymbol{x}} F(\boldsymbol{x}))}{\eta T}, \qquad (92)$$

*On the other hand, by properly taking hyperparameters, the computational complexity of SVRG flow is of order $\mathcal{O}(N^{2/3}/\epsilon)$, when $\min_{0 \leq t \leq \eta T}\mathbb{E}\left[\|\nabla F(\boldsymbol{x}_t)\|^2\right] \leq \epsilon$. Further more, when $F(\boldsymbol{x})$ satisfies PL inequality (128) with coefficient $\gamma$, we have*

$$\mathbb{E}\left[F(\boldsymbol{x}_{\eta T}) - \inf F(\boldsymbol{x})\right] \leq e^{-\gamma \eta T}\mathbb{E}\left[F(\boldsymbol{x}_0) - \inf_{\boldsymbol{x}} F(\boldsymbol{x})\right]. \qquad (93)$$

*The computational complexity of SVRG flow is of order $\mathcal{O}((N + \gamma^{-1}N^{2/3})\log \epsilon^{-1})$ to make $\mathbb{E}\left[F(\boldsymbol{x}_{\eta T}) - \inf F(\boldsymbol{x})\right] \leq \epsilon$.*

*Proof.* Let us define the Lyapunov function, for $t \in [t_i, t_{i+1}]$ (with out loss of generality, let $i = 0$),

$$R_t(\boldsymbol{x}) = F(\boldsymbol{x}) + \frac{c_t}{2}\|\boldsymbol{x} - \boldsymbol{x}_{t_0}\|^2. \qquad (94)$$

Then we have for $\boldsymbol{x}_t$ defined in (90)

$$dR_t(\boldsymbol{x}_t) = \langle \nabla F(\boldsymbol{x}_t), d\boldsymbol{x}_t \rangle + \frac{c_t'}{2}\|\boldsymbol{x}_t - \boldsymbol{x}_{t_0}\|^2 dt + c_t \langle \boldsymbol{x}_t - \boldsymbol{x}_{t_0}, d\boldsymbol{x}_t \rangle. \qquad (95)$$

Due to (90), $\mathbb{E}\left[\|\boldsymbol{x} - \mathbb{E}[\boldsymbol{x}]\|^2\right] \leq \mathbb{E}\left[\|\boldsymbol{x}\|^2\right]$, and Lemma 2 we have

$$\mathbb{E}\left[\left\langle \nabla F(\boldsymbol{x}_t), \frac{d\boldsymbol{x}_t}{dt}\right\rangle\right] = -\mathbb{E}\left[\|\nabla F(\boldsymbol{x}_t)\|^2\right] + \frac{\eta}{2}\mathbb{E}\left[\mathrm{tr}\left(\Sigma_{\mathrm{SVRG}}(\boldsymbol{x}_{t_0}, \boldsymbol{x}_t)\right)\nabla^2 F(\boldsymbol{x}_t)\right]$$

$$\leq -\mathbb{E}\left[\|\nabla F(\boldsymbol{x}_t)\|^2\right] + \frac{\eta\lambda_{\max}(\nabla^2 F(\boldsymbol{x}_t))}{2}\mathbb{E}\left[\mathrm{tr}(\Sigma_{\mathrm{SVRG}}(\boldsymbol{x}_{t_0}, \boldsymbol{x}_t))\right]$$

$$= -\mathbb{E}\left[\|\nabla F(\boldsymbol{x}_t)\|^2\right] + \frac{\eta\lambda_{\max}(\nabla^2 F(\boldsymbol{x}_t))}{2}\mathbb{E}\left[\|\nabla f_\xi(\boldsymbol{x}_t) - \nabla f_\xi(\boldsymbol{x}_{t_0}) + \nabla F(\boldsymbol{x}_{t_0}) - \nabla F(\boldsymbol{x}_t)\|^2\right] \quad (96)$$

$$\leq -\mathbb{E}\left[\|\nabla F(\boldsymbol{x}_t)\|^2\right] + \frac{\eta L_2}{2}\mathbb{E}\left[\|\nabla f_\xi(\boldsymbol{x}_t) - \nabla f_\xi(\boldsymbol{x}_{t_0})\|^2\right]$$

$$\leq -\mathbb{E}\left[\|\nabla F(\boldsymbol{x}_t)\|^2\right] + \frac{\eta L_2^2}{2}\mathbb{E}\left[\|\boldsymbol{x}_t - \boldsymbol{x}_{t_0}\|^2\right].$$

On the other hand, by Young's inequality, for some $\beta > 0$,

$$\mathbb{E}\left[\left\langle \boldsymbol{x}_t - \boldsymbol{x}_{t_0}, \frac{d\boldsymbol{x}_t}{dt}\right\rangle\right] = \mathbb{E}\left[-\langle \boldsymbol{x}_t - \boldsymbol{x}_{t_0}, \nabla F(\boldsymbol{x}_t)\rangle\right] + \frac{\eta}{2}\mathbb{E}\left[\mathrm{tr}\left(\Sigma_{\mathrm{SVRG}}(t_0, \boldsymbol{x}_t)\right)\right]$$
$$(97)$$
$$\leq \frac{\beta}{2}\mathbb{E}\left[\|\boldsymbol{x}_t - \boldsymbol{x}_{t_0}\|^2\right] + \frac{1}{2\beta}\mathbb{E}\left[\|\nabla F(\boldsymbol{x}_t)\|^2\right] + \frac{\eta L_2}{2}\mathbb{E}\left[\|\boldsymbol{x}_t - \boldsymbol{x}_{t_0}\|^2\right].$$

By plugging (96) and (97) into (95), we have

$$\frac{\partial}{\partial t}\mathbb{E}\left[R_t(\boldsymbol{x}_t)\right] \leq -\left(1 - \frac{c_t}{2\beta}\right)\mathbb{E}\left[\|\nabla F(\boldsymbol{x}_t)\|^2\right] + \left[\frac{(\beta + \eta L_2)c_t}{2} + \frac{\eta L_1 L_2}{2} + \frac{c_t'}{2}\right]\mathbb{E}\left[\|\boldsymbol{x}_t - \boldsymbol{x}_{t_0}\|^2\right]. \quad (98)$$

By making
$$(\beta + \eta L_2)c_t + \eta L_1 L_2 + c'_t = 0, \tag{99}$$

we get
$$c_{t_1} = e^{-(\beta+\eta L_2)(t_1-t_0)}c_{t_0} - \frac{\eta L_1 L_2}{\beta + \eta L_2}\left(1 - e^{-(\beta+\eta L_2)(t_1-t_0)}\right). \tag{100}$$

By taking $c_{t_0} = \sqrt{\eta}, c_{t_1} = 0, \beta = \sqrt{\eta}$ we get
$$t_1 - t_0 = \frac{\log\left(1 + \sqrt{\eta}/L_2 + 1/L_1 L_2\right)}{\sqrt{\eta} + \eta L_2} \leq \mathcal{O}\left(\frac{1}{\sqrt{\eta}}\right), \tag{101}$$

when $\eta \to 0$. On the other hand, due to $c'_t < 0$ and (99), we have

$$
\begin{aligned}
\frac{1}{2(t_1-t_0)}\int_{t_0}^{t_1}\mathbb{E}\left[\|\nabla F(\boldsymbol{x}_t)\|^2\right]dt &= \left(\frac{1}{t_1-t_0}\right)\int_{t_0}^{t_1}\left(1 - \frac{c_{t_0}}{2\beta}\right)\mathbb{E}\left[\|\nabla F(\boldsymbol{x}_t)\|^2\right]dt \\
&\leq \left(\frac{1}{t_1-t_0}\right)\int_{t_0}^{t_1}\left(1 - \frac{c_t}{2\beta}\right)\mathbb{E}\left[\|\nabla F(\boldsymbol{x}_t)\|^2\right]dt \\
&\leq \frac{\mathbb{E}\left[R_{t_0}(\boldsymbol{x}_{t_0})\right] - \mathbb{E}\left[R_{t_1}(\boldsymbol{x}_{t_1})\right]}{t_1 - t_0} \\
&= \frac{\mathbb{E}\left[F(\boldsymbol{x}_{t_0})\right] - \mathbb{E}\left[F(\boldsymbol{x}_{t_1})\right]}{t_1 - t_0}.
\end{aligned} \tag{102}
$$

Thus, for any $T = t_I$ and $\Delta = t_1 - t_0 = \cdots = t_I - t_{I-1}$ defined in (101), we have
$$\frac{1}{I\Delta}\sum_{i=1}^{I}\int_{t_i}^{t_{i+1}}\mathbb{E}\left[\|\nabla F(\boldsymbol{x}_t)\|^2\right]dt \leq \frac{2(\mathbb{E}\left[F(\boldsymbol{x}_t)\right] - \inf_{\boldsymbol{x}} F(\boldsymbol{x}))}{T}, \tag{103}$$

which leads to the required convergence rate. Next, let us check the computational complexity of it. Due to $I\Delta = \eta T = \mathcal{O}(\epsilon^{-1})$, $\Delta = \mathcal{O}(1/\sqrt{\eta})$, and Proposition 5, we should running the SVRG for $I = \mathcal{O}(\sqrt{\eta}/\epsilon)$ epochs with $M = \Delta/\eta$ steps in each epoch in Algorithm 2. Besides, note that for each epoch, the computational complexity is of order $\mathcal{O}(M + N) = \mathcal{O}(\Delta/\eta + N)$. Then, by taking $\eta = \mathcal{O}(N^{-2/3})$, the computational complexity of SVRG flow is of order

$$I\mathcal{O}(M+N) = I\mathcal{O}(\Delta/\eta + N) = \mathcal{O}\left(\frac{\sqrt{\eta}}{\epsilon}\left(\eta^{-\frac{3}{2}} + N\right)\right) = \mathcal{O}\left(\frac{N^{\frac{2}{3}}}{\epsilon}\right). \tag{104}$$

Next, let us check the results when $F(\boldsymbol{x})$ satisfies PL inequality with coefficient $\gamma$. We will follow the above notations in the rest of this proof. For any $t \in [t_0, t_1]$, we can reconstruct the $c_t$ in Lyapunov function (95) such that
$$(2\gamma + \beta + \eta L_2)c_t + \eta L_1 L_2 + c'_t = 0, \tag{105}$$

which implies
$$c_{t_1} = e^{-(2\gamma+\beta+\eta L_2)(t_1-t_0)}c_{t_0} - \frac{\eta L_1 L_2}{2\gamma + \beta + \eta L_2}\left(1 - e^{-(2\gamma+\beta+\eta L_2)(t-t_0)}\right). \tag{106}$$

Similarly, by taking $c_{t_0} = \sqrt{\eta}, c_{t_1} = 0, \beta = \sqrt{\eta}$, we get
$$t_1 - t_0 = \frac{\log\left(1 + 2\gamma/\sqrt{\eta}L_1 L_2 + \sqrt{\eta}/L_2 + 1/L_1 L_2\right)}{2\gamma + \sqrt{\eta} + \eta L_2} = \min\left\{\mathcal{O}\left(\frac{1}{\sqrt{\eta}}\right), \mathcal{O}\left(\frac{1}{\gamma}\right)\right\}, \tag{107}$$

when $\eta \to 0$. Plugging this into (98), combining PL inequality and the monotonically decreasing property of $c_t$, we have

$$
\begin{aligned}
\frac{\partial}{\partial t}\mathbb{E}\left[R_t(\boldsymbol{x}_t) - \inf_{\boldsymbol{x}} F(\boldsymbol{x})\right] &\leq -2\gamma\left(1 - \frac{c_t}{2\beta}\right)\left(\mathbb{E}\left[F(\boldsymbol{x}_t)\right] - \inf_{\boldsymbol{x}} F(\boldsymbol{x})\right) - \gamma c_t \mathbb{E}\left[\|\boldsymbol{x}_t - \boldsymbol{x}_{t_0}\|^2\right] \\
&\leq \gamma\left(\mathbb{E}\left[F(\boldsymbol{x}_t)\right] - \inf_{\boldsymbol{x}} F(\boldsymbol{x})\right) - \gamma c_t \mathbb{E}\left[\|\boldsymbol{x}_t - \boldsymbol{x}_{t_0}\|^2\right] \\
&= -\gamma\mathbb{E}\left[R_t(\boldsymbol{x}_t) - \inf_{\boldsymbol{x}} F(\boldsymbol{x})\right].
\end{aligned} \tag{108}
$$

Hence
$$\mathbb{E}\left[F(\boldsymbol{x}_{t_1}) - \inf_{\boldsymbol{x}} F(\boldsymbol{x})\right] \leq e^{-\gamma(t_1-t_0)}\mathbb{E}\left[F(\boldsymbol{x}_{t_0}) - \inf_{\boldsymbol{x}} F(\boldsymbol{x})\right]. \tag{109}$$

Then for any $T = t_m$ and $\Delta = t_{i+1} - t_i = \cdots = t_1 - t_0$, we have
$$\mathbb{E}\left[F(\boldsymbol{x}_T) - \inf_{\boldsymbol{x}} F(\boldsymbol{x})\right] \leq e^{-\gamma T}\mathbb{E}\left[F(\boldsymbol{x}_0) - \inf_{\boldsymbol{x}} F(\boldsymbol{x})\right], \tag{110}$$

which implies the exponential convergence rate of SVRG flow under PL inequality.

On the other hand, let us check the computational complexity of it under PL inequality. As can be seen, to make $\mathbb{E}[F(\boldsymbol{x}_{t_1}) - \inf_{\boldsymbol{x}} F(\boldsymbol{x})] \leq \epsilon$, we should take $I\Delta = \eta T = \mathcal{O}(\gamma^{-1} \log \epsilon^{-1})$. Due to (107), the SVRG flow should be conducted for $I = \max\{\mathcal{O}(\sqrt{\eta}\gamma^{-1} \log \epsilon^{-1}), \mathcal{O}(\log \epsilon^{-1})\}$ epochs with $M = \Delta/\eta$ steps in each epoch, and the computational complexity for each epoch is of order $\mathcal{O}(M + N) = \mathcal{O}(\Delta/\eta + N)$. So that the total computational complexity is of order

$$I\mathcal{O}(\Delta/\eta + N) = \max\{\mathcal{O}(\sqrt{\eta}\gamma^{-1} \log \epsilon^{-1}), \mathcal{O}(\log \epsilon^{-1})\}(\min\{\mathcal{O}(1/\sqrt{\eta}), \mathcal{O}(1/\gamma)\}/\eta + N), \quad (111)$$

If $\gamma^{-1} \geq N^{1/3}$, we take $\eta = \mathcal{O}(N^{-2/3})$, the above equality becomes $\mathcal{O}((N + \gamma^{-1} N^{2/3}) \log \epsilon^{-1})$. On the other hand, when $\gamma^{-1} \leq N^{1/3}$, and $\eta = \mathcal{O}(N^{-2/3})$, the above equality is also $\mathcal{O}((N + \gamma^{-1} N^{2/3}) \log \epsilon^{-1})$. $\qquad\square$

Next, we prove the convergence rate of Riemannian SVRG flow as mentioned in main body of this paper. The following theorem is the formal statement of Theorem 3.

**Theorem 6.** *Let $\pi_t$ in (24), $\Delta = t_1 - t_0 = \cdots = t_I - t_{I-1} = \mathcal{O}(1/\sqrt{\eta})$, and $\eta T = I\Delta$ for $I$ epochs. Then, if Assumption 2 and $\mathbb{E}_{\pi_{t_i,t}}[\mathrm{tr}(\nabla^2 \log(d\pi_t/d\mu)\Sigma_{\mathrm{SVRG}})] \leq \lambda_t \mathbb{E}_{\pi_{t_i,t}}[\mathrm{tr}(\Sigma_{\mathrm{SVRG}})]$ holds for any $t$ and $\lambda_t$ is a polynomial of $t$, then*

$$\frac{1}{\eta T} \sum_{i=1}^{I} \int_{t_i}^{t_{i+1}} \|\mathrm{grad}D_{KL}(\pi_t \parallel \mu)\|_{\pi_t}^2 \, dt \leq \frac{2D_{KL}(\pi_0 \parallel \mu)}{\eta T}. \quad (112)$$

*On the other hand, by taking $\eta = \mathcal{O}(N^{-2/3})$, the computational complexity of Riemannian SVRG flow is of order $\mathcal{O}(N^{2/3}/\epsilon)$ when $\min_{0 \leq t \leq \eta T} \|\mathrm{grad}D_{KL}(\pi_t \parallel \mu)\|_{\pi_t}^2 \leq \epsilon$.*

*Furthermore, when $F(\pi) = D_{KL}(\pi \parallel \mu)$ satisfies log-Sobolev inequality (5), we have*

$$D_{KL}(\pi_{\eta T} \parallel \mu) \leq e^{-\gamma \eta T} D_{KL}(\pi_0 \parallel \mu). \quad (113)$$

*Besides that, it takes $\mathcal{O}((N + \gamma^{-1} N^{2/3}) \log \epsilon^{-1})$ computational complexity to make $D_{KL}(\pi_{\eta T} \parallel \mu) \leq \epsilon$.*

*Proof.* Let us consider the Lyapunov function for union probability measure $\pi_{t_0,t}$ with $t_0 \leq t \leq t_1$

$$R_t(\pi_{t_0,t}) = D_{KL}(\pi_t \parallel \mu) + \frac{c_t}{2} \int \|\boldsymbol{y} - \boldsymbol{x}\|^2 \pi_{t_0,t}(\boldsymbol{y}, \boldsymbol{x}) = D_{KL}(\pi_t \parallel \mu) + \frac{c_t}{2} G(\pi_{t_0,t}). \quad (114)$$

Then

$$\frac{\partial}{\partial t} R_t(\pi_{t_0,t}) = \langle \mathrm{grad}D_{KL}(\pi_t \parallel \mu), \boldsymbol{h}(\pi_t) \rangle_{\pi_t} + \frac{c_t'}{2} G(\pi_{t_0,t}) + \frac{c_t}{2} \frac{\partial}{\partial t} G(\pi_{t_0,t}), \quad (115)$$

where $\partial \pi_t / \partial t = \nabla \cdot (\pi_t \boldsymbol{h}(\pi_t))$ is used to simplify the notations. Then due to (24), Lemma 2 and similar induction to (96)

$$
\begin{aligned}
\langle \mathrm{grad}D_{KL}(\pi_t \parallel \mu), \boldsymbol{h}(\pi_t) \rangle_{\pi_t} = & -\|\mathrm{grad}D_{KL}(\pi_t \parallel \mu)\|_{\pi_t}^2 \\
& + \int \left\langle \nabla \log \frac{d\pi_t}{d\mu}(\boldsymbol{x}), \frac{\eta}{2} \int \pi_{t_0,t}(\boldsymbol{y}, \boldsymbol{x}) \nabla_{\boldsymbol{x}} \cdot \Sigma_{\mathrm{SVRG}}(\boldsymbol{y}, \boldsymbol{x}) \right\rangle d\boldsymbol{y}d\boldsymbol{x} \\
& + \int \left\langle \nabla \log \frac{d\pi_t}{d\mu}(\boldsymbol{x}), \frac{\eta}{2} \int \pi_{t_0,t}(\boldsymbol{y}, \boldsymbol{x}) \Sigma_{\mathrm{SVRG}}(\boldsymbol{y}, \boldsymbol{x}) \nabla_{\boldsymbol{x}} \log \pi_{t_0,t}(\boldsymbol{y}, \boldsymbol{x}) \right\rangle d\boldsymbol{y}d\boldsymbol{x} \\
= & -\|\mathrm{grad}D_{KL}(\pi_t \parallel \mu)\|_{\pi_t}^2 + \frac{\eta}{2} \mathbb{E}_{\pi_{t_0,t}} \left[ \mathrm{tr}\left( \nabla^2 \log \frac{d\pi_t}{d\mu}(\boldsymbol{x}) \Sigma_{\mathrm{SVRG}}(\boldsymbol{y}, \boldsymbol{x}) \right) \right] \\
\leq & -\|\mathrm{grad}D_{KL}(\pi_t \parallel \mu)\|_{\pi_t}^2 + \frac{\eta \lambda_t}{2} \mathbb{E} \left[ \|\nabla \log \mu_\xi(\boldsymbol{x}_{t_0}) - \nabla \log \mu_\xi(\boldsymbol{x}_t)\|^2 \right] \\
\leq & -\|\mathrm{grad}D_{KL}(\pi_t \parallel \mu)\|_{\pi_t}^2 + \frac{\eta L_2 \lambda_t}{2} G(\pi_{t_0,t}),
\end{aligned}
$$
$$\quad (116)$$

where the last inequality is due to the Lipschitz continuity of $\nabla \log \mu_\xi(\boldsymbol{x})$. Similarly, by Fokker-Planck equation, we get

$$
\begin{aligned}
\frac{\partial}{\partial t} G(\pi_{t_0,t}) &= \frac{\partial}{\partial t} \mathbb{E}\left[\|\boldsymbol{x}_t - \boldsymbol{x}_{t_0}\|^2\right] \\
&= \mathbb{E}\left[\left\langle \boldsymbol{x}_t - \boldsymbol{x}_{t_0}, \frac{d\boldsymbol{x}_t}{dt} \right\rangle\right] \\
&= \mathbb{E}\left[\langle \boldsymbol{x}_t - \boldsymbol{x}_{t_0}, \boldsymbol{h}(\pi_t(\boldsymbol{x}_t)) \rangle\right] \\
&\leq \frac{1}{2\beta} \|\mathrm{grad} D_{KL}(\pi_t \parallel \mu)\|_{\pi_t}^2 + \frac{\beta}{2} G(\pi_{t_0,t}) + \frac{\eta}{2} \mathbb{E}_{\pi_{t_0,t}}\left[\mathrm{tr}\left(\Sigma_{\mathrm{SVRG}}(\boldsymbol{x}_{t_0}, \boldsymbol{x}_t)\right)\right] \\
&\leq \frac{1}{2\beta} \|\mathrm{grad} D_{KL}(\pi_t \parallel \mu)\|_{\pi_t}^2 + \left(\frac{\beta}{2} + \frac{\eta L_2}{2}\right) G(\pi_{t_0,t}).
\end{aligned}
\tag{117}
$$

Combining (115), (117) and (116), we get

$$
\frac{\partial}{\partial t} R_t(\pi_{t_0,t}) \leq -\left(1 - \frac{c_t}{2\beta}\right) \|\mathrm{grad} D_{KL}(\pi_t \parallel \mu)\|_{\pi_t}^2 + \left[\frac{(\beta + \eta L_2)c_t}{2} + \frac{\eta \lambda_t L_2}{2} + \frac{c_t'}{2}\right] G(\pi_{t_0,t}). \tag{118}
$$

By taking

$$
(\beta + \eta L_2)c_t + \eta \lambda_t L_2 + c_t' = 0, \tag{119}
$$

which implies

$$
c_{t_1} = e^{-(\beta + \eta L_2)(t_1 - t_0)} c_{t_0} - \int_{t_0}^{t_1} \eta L_2 \lambda_t e^{-(\beta + \eta L_2)(t - t_0)} dt. \tag{120}
$$

Without loss of generality, let

$$
\begin{aligned}
a_p &= \eta L_2 \int_{t_0}^{t_1} (t - t_0)^p e^{-(\beta + \eta L_2)(t - t_0)} dt \\
&= -\frac{\eta L_2}{\beta + \eta L_2} (t_1 - t_0)^p e^{-(\beta + \eta L_2)(t - t_0)} dt + \eta L_2 \int_{t_0}^{t_1} p(t - t_0)^{p-1} e^{-(\beta + \eta L_2)(t - t_0)} dt \\
&= p a_{p-1} - \frac{\eta L_2}{\beta + \eta L_2} (t_1 - t_0)^p e^{-(\beta + \eta L_2)(t - t_0)} \\
&= \cdots \\
&= -\frac{\eta L_2}{\beta + \eta L_2} \left(1 - e^{-(\beta + \eta L_2)(t_1 - t_0)} \mathrm{Poly}(t_1 - t_0, p)\right),
\end{aligned}
\tag{121}
$$

where $\mathrm{Poly}(t_1 - t_0, p)$ is a $p$-th order polynomial of $t_1 - t_0$. W.l.o.g, let

$$
\lambda_t = \lambda_{t_0} + \mathrm{Poly}(t - t_0, p) = \lambda_{t_0} + \sum_{i=1}^{p} b_i(t - t_0)^p. \tag{122}
$$

Then we have

$$
\begin{aligned}
\int_{t_0}^{t_1} \eta L_2 \lambda_t e^{-(\beta + \eta L_2)(t - t_0)} dt &= \lambda_{t_0} a_0 + \sum_{i=1}^{p} a_i b_i \\
&= \frac{\eta L_2 \lambda_{t_0}}{\beta + \eta L_2} \left(1 - e^{-(\beta + \eta L_2)(t_1 - t_0)}\right) - \frac{\eta L_2}{\beta + \eta L_2} \left(\sum_{i=1}^{p} b_i + e^{-(\beta + \eta L_2)(t_1 - t_0)} \mathrm{Poly}(t_1 - t_0, p)\right) \\
&= \frac{\eta L_2}{\beta + \eta L_2} \left(\lambda_{t_0} - \sum_{i=1}^{p} b_i - \mathrm{Poly}(t_1 - t_0, p) e^{-(\beta + \eta L_2)(t_1 - t_0)}\right)
\end{aligned}
\tag{123}
$$

By invoking $c_{t_0} = \sqrt{\eta}, c_{t_1} = 0, \beta = \sqrt{\eta}$, and the above equality into (120) we get

$$
t_1 - t_0 = \frac{1}{\sqrt{\eta} + \eta L_2} \log \frac{1 + \sqrt{\eta} L_2 + L_2 \mathrm{Poly}(t_1 - t_0, p)}{L_2(\lambda_0 - \sum_{i=1}^{p} b_i)}, \tag{124}
$$

which implies $t_1 - t_0 = \mathcal{O}(1/\sqrt{\eta})$ as in (101) (note that the value of $b_i$ are automatically adjusted to make the above equality meaningful), by taking $\eta \to 0$. Thus, similar to (102), we get

$$
\frac{1}{2(t_1 - t_0)} \int_{t_0}^{t_1} \|\mathrm{grad} D_{KL}(\pi_t \parallel \mu)\|^2 dt \leq \frac{D_{KL}(\pi_{t_0} \parallel \mu) - D_{KL}(\pi_{t_1} \parallel \mu)}{t_1 - t_0}. \tag{125}
$$

Then the two conclusions are similarly obtained as in Theorem 5 by taking $\eta = \mathcal{O}(N^{-2/3})$. $\qquad \square$

Notably, the imposed "proper" condition on $\pi_0$ can be implied by the polynomial upper bound to the spectral norm of Hessian. This is because, from Lemma 2 and semi-positive definite property of $\Sigma_{\text{SVRG}}$, we know

$$\mathbb{E}_{\pi_{t_i,t}}\left[\nabla^2\log\left(d\pi_t/d\mu\right)\Sigma_{\text{SVRG}}\right] \leq \mathbb{E}_{\pi_{t_i,t}}\left[\lambda_{\max}\left(\nabla^2\log\left(d\pi_t/d\mu\right)\right)\text{tr}(\Sigma_{\text{SVRG}})\right]. \tag{126}$$

Then, we may take

$$\lambda_t = \frac{\mathbb{E}_{\pi_{t_i,t}}\left[\lambda_{\max}\left(\text{tr}(\nabla^2\log\left(d\pi_t/d\mu\right))\text{tr}(\Sigma_{\text{SVRG}})\right)\right]}{\mathbb{E}_{\pi_{t_i,t}}\left[\text{tr}(\Sigma_{\text{SVRG}})\right]}. \tag{127}$$

Due to the formulation of Riemannian SVRG (24), the density $\pi_t$ only depends on $\pi_0$ and $\mu$. Therefore, under properly chosen $\pi_0$, the obtained $\pi_t$ can satisfy the imposed condition on the spectral norm. Besides that, we do not impose any restriction on the order of $\lambda_t$, while the polynomial function class can approximate any function so that it can be extremely large. Therefore, the imposed polynomial order of $\lambda_t$ can be easily satisfied.

## D   More than Riemannian PL inequality

We observe that Theorems 1, 2, and 3 demonstrate the nice log-Sobolev inequality (5) improves the convergence rates into global ones. Therefore, we briefly discuss the condition in this section. As mentioned in Section 3, the log-Sobolev inequality is indeed the Riemannian PL inequality in Riemannian manifold. To see this, for a function $f$ defined on $\mathbb{R}^d$, the PL inequality is that for any global minima $\boldsymbol{x}^*$ of it, we have

$$2\gamma(f(\boldsymbol{x}) - f(\boldsymbol{x}^*)) \leq \|\nabla f(\boldsymbol{x})\|^2 \qquad \text{PL} \tag{128}$$

holds for any $\boldsymbol{x}$. The PL inequality indicates that all local minima are global minima. It is a nice property that guarantees the global convergence in Euclidean space [23]. Naturally, we can generalize it into Wasserstein space. To this end, let $F(\pi) = D_{KL}(\pi \parallel \mu)$, the sole global minima is $\pi = \mu$ with $F(\mu) = 0$. Then, in Wasserstein space, the PL inequality (128) is generalized to

$$2\gamma D_{KL}(\pi \parallel \mu) \leq \|\text{grad}D_{KL}(\pi \parallel \mu)\|_\pi^2, \qquad \text{Riemannian PL.} \tag{129}$$

which is log-Sobolev inequality (5) due to (7). Thus, our Theorems 1, 2, and 3 indicate that Riemannian PL inequality guarantees the global convergence on manifold.

Moreover, if $f$ in (128) has Lipschitz continuous gradient, the properties of quadratic growth (QG) and error bound (EB) are equivalent to the PL inequality [23]

$$f(\boldsymbol{x}) - f(\boldsymbol{x}^*) \geq \frac{\gamma}{2}\|\boldsymbol{x} - \boldsymbol{x}^*\|^2 \qquad \text{QG,} \tag{130}$$

$$\|\nabla f(\boldsymbol{x})\| \geq \gamma\|\boldsymbol{x} - \boldsymbol{x}^*\| \qquad \text{EB,} \tag{131}$$

so that global convergence. In Wasserstein space, the two properties are generalized as

$$D_{KL}(\pi \parallel \mu) \geq \frac{\gamma}{2}\mathsf{W}_2^2(\pi, \mu) \qquad \text{Riemannian QG,} \tag{132}$$

$$\left\|\nabla\log\frac{d\pi}{d\mu}\right\|_\pi \geq \gamma\mathsf{W}_2(\pi, \mu) \qquad \text{Riemannian EB.} \tag{133}$$

Then, the relationship between the three properties in Wasserstein space is illustrated by the following proposition. This proposition is from [34], and we prove it to make this paper self-contained.

**Proposition 6.** *[Otto-Vallani][34] Let $F(\pi)$ be $D_{KL}(\pi \parallel \mu)$, then Riemannian PL $\Rightarrow$ Riemannian QG, Riemannian PL $\Rightarrow$ Riemannian EB.*

This proposition indicates that Riemannian PL inequality implies the other two conditions, but not vice-versa. This is because the Lipschitz continuity of Riemannian gradient i.e., $\|\text{grad}D_{KL}(\pi \parallel \mu)\|_\pi^2 \leq L\mathsf{W}_2^2(\pi, \mu)$ does not hold as in Euclidean space [45] (Wasserstein distance is weaker than the KL divergence).

*Proof.* If Riemannian PL $\Rightarrow$ Riemannian QG then Riemannian PL $\Rightarrow$ Riemannian EB is naturally proved. Next, we prove the first claim. Let $G(\pi) = \sqrt{F(\pi)}$, then by chain-rule and Riemannian PL inequality,

$$\|\text{grad}G(\pi)\|_\pi^2 = \left\|\frac{\text{grad}D_{KL}(\pi \parallel \mu)}{2D_{KL}^{\frac{1}{2}}(\pi \parallel \mu)}\right\|_\pi^2 \geq \frac{\gamma}{2}. \tag{134}$$

Then let us consider

$$\begin{cases} \dfrac{\partial}{\partial t}\pi_t = \mathrm{Exp}_{\pi_t}[-\mathrm{grad}G(\pi_t)] = \nabla \cdot (\pi_t \mathrm{grad}G(\pi_t)), \\ \pi_0 = \pi \end{cases} \tag{135}$$

where the last equality can be similarly proved as in (2). Then

$$\begin{aligned} G(\pi_0) - G(\pi_T) &= \int_0^T \frac{\partial}{\partial t}G(\pi_t)dt \\ &= \int_0^T \mathrm{Dev}G(\pi_t)[-\mathrm{grad}G(\pi_t)]dt \\ &= \int_0^T \|-\mathrm{grad}G(\pi_t)\|_{\pi_t}^2 \, dt \\ &\geq \int_0^T \frac{\gamma}{2}dt \\ &= \frac{\gamma T}{2}. \end{aligned} \tag{136}$$

Due to this, and $G(\pi) \geq 0$, there exists some $T$ such that $\pi_T = \mu$. Since $\pi_t$ is a curvature satisfies $\pi_0 = \pi$ and $\pi_T = \mu$, due to the definition of Wasserstein distance is geodesic distance, we have

$$W_2(\pi, \mu) = W_2(\pi_0, \pi_T) \leq \int_0^T \|\mathrm{grad}G(\pi_t)\|_{\pi_t}dt. \tag{137}$$

Thus,

$$G(\pi_0) = G(\pi_0) - G(\pi_T) = \int_0^T \|-\mathrm{grad}G(\pi_t)\|_{\pi_t}^2 \, dt \geq \sqrt{\frac{\gamma}{2}}\int_0^T \geq \|\mathrm{grad}G(\pi_t)\|_{\pi_t}dt \geq \sqrt{\frac{\gamma}{2}}W_2(\pi, \mu), \tag{138}$$

which implies our conclusion. □