# OpenReview forum: "Continuous-time Riemannian SGD and SVRG Flows on Wasserstein Probabilistic Space"
_NeurIPS.cc/2025/Conference — NeurIPS 2025 poster_

### Official Review · Reviewer_oUnb · 2025-06-19

**Clarity:** 1
**Significance:** 2
**Originality:** 1
**Rating:** 3
**Confidence:** 4

**Summary:**

This paper introduces a continuous‐time analogue of stochastic gradient descent (SGD) and stochastic variance–reduced gradient (SVRG) methods formulated on the manifold of probability measures endowed with the second‐order Wasserstein metric.  Building on the well‐known correspondence between Riemannian gradient flow in this space and Langevin diffusion, the authors construct Riemannian SGD and SVRG flows.  By interpreting the discrete Riemannian SGD and SVRG updates as transformations of random vectors in Euclidean space, and then taking the learning‐rate limit, they derive the corresponding stochastic differential equations (SDEs).  Applying the Fokker–Planck equation translates these SDEs into deterministic “probability‐measure flows” on the Riemannian manifold. The authors establish some convergence guarantees under Lipschitz‐gradient assumptions and they prove that the Riemannian SGD flow attains a first‐order stationary rate of $O(1/\sqrt{T})$ (and $O(1/T)$ under a log‐Sobolev/PL inequality), while the Riemannian SVRG flow achieves $O(N^{2/3}/T)$ (and exponential convergence under PL).  These match the best-known Euclidean continuous‐time rates The authors show that standard discrete SGLD and variance‐reduced Langevin MCMC algorithms coincide as time‐discretizations of their proposed flows, thereby linking the continuous‐time structure underlying these sampling methods.

**Questions:**

* If the authors are able, then adding some numerical examples to the paper could significantly improve this work and help to solidify the application of this approach.
* The language and grammar used throughout the paper is very challenging to read. This would need to be significantly improved before this work is published in NeurIPS.
* Could the authors comment on how their theoretical results could (or perhaps could not) be applied to functionals other than KL?

**Ethical Concerns:**

["NO or VERY MINOR ethics concerns only"]

**Final Justification:**

The authors have addressed the questions that I asked, and if these changes are included in the paper, then I believe it warrants an increase in my score. However, as I have noted to the authors, I am still concerned about the readability of the paper and the limited range of experimental results provided.

**Limitations:**

There are no ethical or societal issues that the authors need to address.

**Quality:**

2

**Strengths And Weaknesses:**

Quality

The paper exhibits a high level of mathematical rigour: all constructions and convergence proofs are presented with precise assumptions—such as Lipschitz continuity and log-Sobolev or Polyak–Łojasiewicz inequalities—and are thoroughly detailed in the appendices. It also provides a comprehensive treatment of both nonconvex regimes and the faster rates attainable under PŁ conditions, unifying stochastic gradient and variance-reduced flows in a single framework. However, its reliance on strong technical conditions may be a small limit of its wider applicability. Verifying Lipschitz-gradient or log-Sobolev constants for complex target measures is often challenging in practice. Moreover, the absence of any empirical validation leaves open questions about the practical sharpness of the theoretical rates and the behaviour of the flows outside simplified settings.

Clarity

While the manuscript is structured in a logical sequence, from stating the important preliminaries through to SGD and SVRG flows, the presentation suffers from awkward phrasing, grammatical errors, and convoluted sentence structures that permeate pretty much throughout every section. Definitions of measure-flow notation and details about Riemannian manifolds are technically sound, but the poor use of English frequently forces the reader to pause and re-read passages to extract the intended meaning.  These linguistic challenges, combined with the dense theoretical material and very little intuitive exposition, presents a significant barrier to comprehension for all but the most specialised experts in geometric measure theory and continuous-time stochastic dynamics.

Significance

By proposing a continuous-time analogue of stochastic gradient and variance-reduced methods on the Wasserstein manifold, the authors provide a new theoretical bridge between optimisation flows and Langevin-type sampling algorithms. This unification could lead to new algorithmic designs that blend optimisation and sampling perspectives. Yet, the paper stops short of connecting these continuous flows to concrete algorithmic implementations or demonstrating improvements over existing discrete methods. As a result, its immediate impact on practical machine-learning tasks remains speculative until accompanied by empirical studies.

Originality

Methodologically, the use of the Fokker–Planck framework to lift Euclidean SDE limits into deterministic manifold flows represents an interesting idea for the Riemannian setting (but not unique for the Euclidean setting), and the insight that SGLD and variance-reduced Langevin samplers arise as discretisations of the same underlying flows gives a new view on this relationship. While these theoretical connections are interesting, many of the resulting convergence rates mirror those already known in the Euclidean setting, making some of the performance guarantees anticipated by analogy. The focus on KL-divergence minimisation also narrows the scope, leaving open the question of how readily the framework extends to other divergences or objective functions. Furthermore, there has been a lot of recent work in this area, and it is not entirely clear how much of this work is original in the context of other recent papers.

---

> ### Author Rebuttal · Authors · 2025-07-29
>
> We deeply grateful for your careful reviewing and valuable comments. We address your concerns as below.
>
> **Q1**: Strong technical conditions (Lipschitz-gradient or log-Sobolev constants).
>
> **A1**: For the Lipschitz-gradient Assumption 1 and 2, they are **only applied to bridge the gap between discrete and continuous methods** (Proposition 2, 4, 5), while our convergence results in Theorem 1, 2, 3 **do not require** them. Besides, the Lipschitz assumption is standard to bridge the discrete and continuous differential equation (as we did) in numerical analysis (see section of convergence in [4]).
>
> For log-Sobolev inequality, we also characterize the convergence rates of our methods without it in Theorems 1, 2, 3, evaluated by the norm of Riemannian gradient as in **standard non-convex optimization**. Since log-Sobolev inequality roughly implies "strongly-convexity" as clarified in our paper.
>
> **Q2**: The absence of any empirical validation.
>
> **A2**: Thanks for your valuable advice. We have made an example (Example 1) of sampling on Gaussian to illustrate the tightness of our theoretical results.
>
> **However, we recognize that numerical results will make our conclusions more convincing**. We evaluate different sampling algorithms on two experiments. **Since the NeurIPS rebuttal do not allow us to convey these results by image**, we present them in the following tables (results are averaged by 5 independent runs).
>
> Concretely, we set the each target distribution $\mu_{\xi}$ as **Gaussian or mixture-Gaussian**, and conduct sampling by the proposed algorithms. Notably, our convergence rates are respectively evaluated by update steps ($T$) and gradient computation (Riemannian SGD and SVRG has less gradient computations than Riemannian GD). The results are summarized in Tables 1-4, the KL divergence and Fisher divergence are estimated by combining MCMC and an estimated density as in [1].
>
> As can be seen, **under the same update steps**, Riemannian SVRG and Riemannian GD have better convergence results than Riemannian SGD (see Theorem 1, 2, 3), while **under the same gradient computations**, Riemannian SGD and Riemannian SVRG (the two algorithms have less gradient computations in each update step) have better convergence results (see discussion after Theorem 3). These results are consistent with our conclusion in paper. **We promise to add these numerical results in the revised version**, and we sincerely hope you may reconsider the concern on numerical results.
>
> **Table1. KL and Fisher divergences on Gaussian measured by update steps**
> | **KL divergence \ Update Steps** | 0    | 20   | 50     | 80      | 100     | **Fisher divergence \ Update Steps** | 0    | 20    | 50    | 80   | 100   |
> | -------------------------------- | ---- | ---- | ------ | ------- | ------- | -------------------------------- | ---- | ----- | ----- | ------ | ------ |
> | LD           | 6.12 | **0.43** | 0.0214 | **1.06e-3** | **1.44e-4** | LD      | 11.9 | **0.259** | **0.102** | **0.0655** | **0.0424** |
> | SGLD                 | 6.12 | 0.51 | 0.0274 | 1.26e-3 | 1.62e-4 | SGLD         | 11.9 | 0.279 | 0.142 | 0.0723 | 0.0512 |
> | SVRGLD               | 6.12 | 0.67 | **0.012** | 1.16e-3 | 1.52e-4  | SVRGLD            | 11.9 | 0.244 | 0.119 | 0.0679 | 0.0482 |
>
>
> **Table 2. KL and Fisher divergences on Gaussian measured by gradient computation**
> | **KL divergence \ Gradient Computations** | 0 | 2000 | 5000 | 8000 | 10000 | **Fisher divergence \ Gradient Computations** | 0 | 2000 | 5000 | 8000 | 10000 |
> | -------------------------------- | -------- | ------- | ------ | ------- | ------- | ---------------------------- | ------ | ------- | ------- | ------ |------ |
> | LD                    | 6.12 | **0.43** | 0.0214 | 1.06e-3 | 1.44e-4 | LD  | 11.9 | 0.259 | 0.102 | 0.0655 | 0.0424 |
> | SGLD                  | 6.12 | 1.76e-4 | 8.99e-6 | 1.71e-6 | 7.56e-7 |SGLD   | 11.9 | 3.64e-4 | 1.88e-5 | 3.73e-6 | 1.87e-6 |
> | SVRGLD                | 6.12 | **<1e-16** | **<1e-16** | **<1e-16** | **<1e-16** | SVRGLD   | 11.9 | **<1e-16** | **<1e-16** | **<1e-16** | **<1e-16** |
>
>  **Table 3. KL divergence and Fisher divergenceson on mixture Gaussian measured by update steps**
> |  **KL divergence \ Update Steps**  | 0 | 200 | 500 | 800 | 1000 |  **Fisher divergence \ Update Steps**     | 0 | 200 | 500 | 800 | 1000 |
> | -------------------------------- | -------- | ------- | ------ | ------- | ------- | ---------------------------- | ------ | ------- | ------- | ------ |------ |
> | LD            | 0.57 | **0.33** | **0.07** | **0.082** | **0.076** |LD                | 341.2 | 0.252 | 0.248 | **0.014** | **0.018** |
> |SGLD        | 0.57 | 0.53 | 0.33 | 0.127 | 0.112 |SGLD            | 341.2 | 0.362 | **0.143** | 0.019 | 0.022 |
> |SVRGLD        | 0.57 | 0.39 | 0.16 | 0.092 | 0.082 |SVRGLD           | 341.2 | **0.135** | 0.036 | 0.017 | 0.021 |
>
>
> **Table 4. KL divergence and Fisher divergenceson on mixture Gaussian measured by gradient computations**
> | **Gradient Computations** | 0 | 1000 | 2500 | 4000 | 5000 | **Gradient Computations** | 0 | 1000 | 2500 | 4000 | 5000 |
> | -------------------------------- | -------- | ------- | ------ | ------- | ------- | ---------------------------- | ------ | ------- | ------- | ------ |------ |
> | LD                   | 0.57 | 0.33 | 0.07 | 0.082 | 0.076 | LD                    | 341.2 | 0.252 | 0.248 | 0.014 | 0.018 |
> | SGLD                 | 0.57 | 0.23 | 0.124 | 0.074 | 0.062 |SGLD                | 341.2 | **0.219** | **0.237** | 0.009 | 0.006 |
> |SVRGLD                | 0.57 | **0.21** | **0.062** | **0.072** | **0.032** |SVRGLD               | 341.2 | 0.659 | 0.241 | **0.0081** | **0.0049** |
>
> **Q3**: The language and grammar used throughout the paper is very challenging to read.
>
> **A3**: We really grateful for your carefully reviewing all these materials with heavy mathematical notations. We **promise** we will revise our paper as follows to make our paper more readable.
>
> We will carefully check the paper word by word to fix all unproper phrasing, gramma issues, and convoluted sentence structures.
>
> We will rewrite the part of theoretical background, add more intuitive exposition to make these materials friendly to all audiences.
>
>
> **Q4**: Yet, the paper stops short of connecting these continuous flows to concrete algorithmic implementations or demonstrating improvements over existing discrete methods.
>
> **A4**: By combining our proved convergence rates of continuous methods and the proved gap between discrete and continuous methods can address your concerns. Roughly speaking, by Proposition 6, we know the Wasserstein distance is upper bounded by KL divergence (proved to be converge) $\frac{\gamma}{2}W_{2}^{2}(\pi, \mu) \leq D_{KL}(\pi\parallel \mu)$. Since Propositions 2, 4, 5 imply $W_{2}^{2}(\hat{\pi}_{n}, \pi_{n\eta}) \leq E[||x_{n} - \hat{x}_{n\eta}||^{2}]\leq \mathcal{O}(\eta)$ ($\hat{\pi}_{n}$ and $\pi_{n\eta}$ are respectively obtained by proposed discrete and continuous algorithms), we get the convergence rates of the discrete methods, which further demonstrates the improvements over existing discrete methods. **We promise to add these analysis to in the revised version.**
>
> **Q5**: The resulting convergence rates mirror those already known in the Euclidean setting.
>
> **A5**: Though the convergence rates match the ones in Euclidean space, **but the proof techniques are more complicated**. For example, it takes us a lot efforts to construct the Lyapunov function in (114), which is critical to prove the convergence rate. The construction relies techniques from both optimization and stochastic analysis, which are more sophisticated compared with the scheme in Euclidean space.
>
> **Q6**: Extensions to other objective functions more than KL divergence.
>
> **A6**: Our framework has the potential to be extended to other objectives. As can be seen, our core idea in Figure 2 **does not depend on the optimization objective**, which guarantees the extensions of flows to the other objectives. However, it worth noting that a **practical algorithm** may only be obtained under KL divergence.
>
> Taking Riemannian gradient as an example, it takes the formulation of $\frac{\partial{\pi_{t}}}{\partial{t}} = \nabla\cdot(\pi_{t}\mathrm{grad} F(\pi_{t}))$. The flow can be implemented in practice only if $\mathrm{grad} F(\pi_{t})$ is efficiently computed. However, this is not true for many objectives. For example, then $F(\pi) = W_{2}^{2}(\pi, \mu)$, $\mathrm{grad} F(\pi) = -2(T_{\pi\rightarrow \mu} - \mathrm{id})$ (see Theorem 1.4.11 in [1]), the $ T_{\pi\rightarrow \mu}$ is the optimal transportation map between $\pi$ and $\mu$, which is $\mathcal{O}(e^{d})$ complex in general.
>
> However, for KL divergence, $\frac{\partial{\pi_{t}}}{\partial{t}} =  -\nabla\cdot(\pi_{t}\nabla\log{\mu}) + \Delta\pi_{t}$, corresponds to SDE $dx_{t} = \nabla\log{\mu(x_{t})}dt + \sqrt{2}dW_{t}$, which makes it practical. This explains why the existing gradient flow in Wasserstein space is mainly discussed under KL divergence [1, 2, 3].
>
> **Q7**: Furthermore, there has been a lot of recent work in this area, and it is not entirely clear how much of this work is original in the context of other recent papers.
>
> **A7**: As clarified in line 72, the existing literature focus on discrete Langevin dynamics and its convergence rates under different criteria [1, 2, 3]. The main contributions in this paper **the continuous Riemannian SGD and SVRG flows** are not explored yet.
>
> **Q8-10**: The numerical verification; Grammar issues; The extensions to other divergence.
>
> **A8-10**: The answers refer to A2, A3, and A6 in above.
>
>  Ref:
>
> [1] Kinoshita et al., Improved Convergence Rate of Stochastic Gradient Langevin Dynamics with Variance Reduction and its Application to Optimization
>
> [2] Cheng et al., Convergence of langevin mcmc in kl-divergence
>
> [3] Chewi et al., Log-concave sampling
>
> [4] https://en.wikipedia.org/wiki/Numerical_methods_for_ordinary_differential_equations

---

> > ### Author Response · Authors · 2025-08-03
> > **Looking forward to see your feedback**
> >
> > Dear reviewer:
> >
> > Thank you again for your time reviewing our paper. We would appreciate it if you could confirm that our responses address your concerns. We would also be happy to engage in further discussions to address any other questions that you might have.
> >
> > Best regards

---

> > ### Comment · Reviewer_oUnb · 2025-08-05
> >
> > Dear Authors,
> >
> > Thank you for taking the time to respond to my questions. I'm very pleased to see that you have put a lot of effort into answering both my questions and the questions raised by the other reviewers. A quick follow-up point.
> >
> > * It's great that you will now be able to include the numerical examples in the revised paper. However, the two examples chosen are relatively simple, e.g., two-dimensional. I realise that there isn't enough time in the rebuttal period to develop further numerical results. However, I think there is a missed opportunity here to identify specific model classes where this approach would be better than simpler methods. Do you have any thoughts on which model types would expect this approach to perform significantly better than simpler gradient-based methods, e.g., SGD?
> >
> > I'm willing to increase my score, but I would note that I think for the NeurIPS conference, readability is very important. There is no way for me to check that this is improved for the final version, so that will have to be a decision for the AC.

---

> > > ### Author Response · Authors · 2025-08-05
> > >
> > > Thanks for your replying, we really appreciate for your insightful comments. In fact, high-dimensional sampling is a really hard problem, especially for our second case i.e., sampling from mixture-Gaussian, which can be viewed as a hard non-convex optimization problem [1]. We would like to extend our experiments to more high-dimensional problmes in the revised version as you suggested. As for the cases suitable for our methods, from our empirical observations, we think our methods should be applied to the log-concave sampling (our Example 1) and large-scale finite-sum problems. **As for the writting, we promise to carefully polish our language to make the paper more readable.**

---

> > > > ### Comment · Reviewer_oUnb · 2025-08-05
> > > >
> > > > Thank you.

---

### Official Review · Reviewer_ahtD · 2025-06-21

**Clarity:** 3
**Significance:** 3
**Originality:** 3
**Rating:** 5
**Confidence:** 3

**Summary:**

This paper studies the problem of minimizing the KL divergence where the target distribution is stochastic. A solution to solve this problem is for instance the Stochastic Gradient Langevin dynamic (SGLD). In this work, this problem is studied from the Wasserstein gradient flow point of view. It is proposed to solve this problem through stochastic Riemannian gradient descent, and it is shown that this algorithm approximates the trajectory of a flow satisfying a continuity equation given in closed-form. Moreover, it is also shown that SGLD can be seen as a discretization of this continuity equation. Then, the convergence of the continuous flow is shown under mild assumption, and the PL inequality. Then, the same work is done for the stochastic Riemannian variance reduction gradient flow.

**Questions:**

In Section 3 and 4 giving preliminaries on the Wasserstein space and on Wasserstein gradient of the KL divergence, there are no assumptions of regularity on densities. For instance, I think the KL divergence has only subgradients on the Wasserstein space, and only assuming enough regularity, see e.g. [1] (Theorem 10.4.9) or [2]. I believe this should be clarified.

In Remark 1, it is stated "Unfortunately, unlike our method, this method is not applied to the stochastic optimization problem in Section 5, because the aforementioned minimization problem can not involve stochastic gradient as expected." However, stochastic proximal algorithm also exist, and could be used to solve the same problem as Section 5, equation (11). While it may not have been done yet, I don't think the statement is true.

In this work, you analyze only the convergence of the continuous dynamic. Would it be possible to study also the convergence of the discrete schemes?

Other objectives are solved using a stochastic Wasserstein gradient descent, e.g. barycenters [3,4] or the Sliced-Wasserstein distance [5]. Could the analysis done in this work be extended to these objectives?

It could be nice to add experiments demonstrating the rates, even in the Gaussian case where everything may be known in closed-form?

**Typos**
- Line 113: "As can be seen, with this lemma and (2), we can directly get the direction of $\pi_t$ corresponded to a SDE": sentence not very clear, "corresponded"
- Line 126: "we seek to corresponding the discrete dynamics"
- Line 132-133: "the corresponded F-P" -> corresponding
- Line 181: "Thus, the target distribution becomes $\mu$ Proposition (3) in the sequel."
- Equation (23): lack a square on $\|\Gamma_\mu^\pi(u)\|$


[1] Ambrosio, L., Gigli, N., & Savaré, G. (2008). Gradient flows: in metric spaces and in the space of probability measures. Springer Science & Business Media.

[2] Xu, Y., & Li, Q. (2024). Forward-Euler time-discretization for Wasserstein gradient flows can be wrong. arXiv preprint arXiv:2406.08209.

[3] Backhoff-Veraguas, J., Fontbona, J., Rios, G., & Tobar, F. (2022). Bayesian learning with Wasserstein barycenters. ESAIM: Probability and Statistics, 26, 436-472.

[4] Chewi, S., Maunu, T., Rigollet, P., & Stromme, A. J. (2020, July). Gradient descent algorithms for Bures-Wasserstein barycenters. In Conference on Learning Theory (pp. 1276-1304). PMLR.

[5] Liutkus, A., Simsekli, U., Majewski, S., Durmus, A., & Stöter, F. R. (2019, May). Sliced-Wasserstein flows: Nonparametric generative modeling via optimal transport and diffusions. In International Conference on machine learning (pp. 4104-4113). PMLR.

**Ethical Concerns:**

["NO or VERY MINOR ethics concerns only"]

**Final Justification:**

This works derive continuity equations for the continuous counterpart of the stochastic gradient descent and stochastic variance reduction gradient descent on the Wasserstein space. In my opinion, these results are quite interesting, and the authors made connections with known algorithms such as SGLD and SVRGLD, showed the convergence of the flows under mild assumptions, and provided an example in the Gaussian setting.

The original submission is however not very clear, not always very rigorous, and lacks numerical experiments. Nonetheless, I think some of the unclear things have been clarified in the rebuttal, and experiments were provided. Therefore, I believe that the main concerns have been adressed, and I would be inclined to suggest acceptance.

**Limitations:**

yes

**Quality:**

3

**Strengths And Weaknesses:**

The point of view of analyzing the minimization of the KL divergence with a stochastic target is a nice contribution, which is complementary to works done on SGLD and SVRG. The paper is pretty clear, with a first section analyzing the minimization of the KL divergence under no stochasticity, and then diving into the stochastic versions of the problems. Moreover, there are examples on Gaussian allowing to verify the rates. Overall, I believe this is an interesting article, but it also has some weaknesses that I will describe below.


**Strengths**:
- A clear analysis of the minimization of the KL with and without stochasticity
- Analysis of stochastic gradient descent and stochastic variance reduction gradient on the Wasserstein space, with convergence results
- The strategy to get a continuity equation in the stochastic case is interesting
- Example on Gaussian


**Weaknesses**:
- I believe some technical assumptions are lacking (e.g. the analysis is restricted to absolutely continuous distributions, but it is never mentioned...)
- There are no numerical results demonstrating the theoretical results

---

> ### Author Rebuttal · Authors · 2025-07-29
>
> We thank for your valuable comment and address your concerns as follows.
>
> **Q1**: I believe some technical assumptions are lacking (e.g. the analysis is restricted to absolutely continuous distributions, but it is never mentioned...)
>
> **A1**: Thanks for pointing it. In this paper we assume the target distribution $\mu\propto \exp(-V(x))$ in line 102. Then the KL $D_{KL}(\pi\parallel \mu)\propto \int V(x)d\pi(x) - \int \log{\pi(x)}d\pi(x)$, so that the KL divergence exists whenever $\pi(x)$ is absolutely continuous to Lebesgue measure. This is true when the initialization distribution $\pi_{0}$ satisfies this condition, since the discussed $\pi_{t}$ satisfies the F-P equation (2). We will add this discussion in the revised version to make our conclusion more precises.
>
> **Q2**: There are no **numerical results** demonstrating the theoretical results
>
> **A2**: Thanks for your valuable advice. We have made an example (Example 1 in our paper) of sampling on Gaussian distributions to illustrate the tightness of our theoretical results.
>
> **However, we recognize that numerical results will make our conclusions more convincing**. We evaluate different sampling algorithms on two experiments. **Since the NeurIPS rebuttal do not allow us to convey these results by image**, we present them in the following tables (results are averaged by 5 independent runs).
>
> Concretely, we set the each target distribution $\mu_{\xi}$ as **Gaussian or mixture-Gaussian**, and conduct sampling by the proposed algorithms. Notably, our convergence rates are respectively evaluated by update steps ($T$) and gradient computation (Riemannian SGD and Riemannian SVRG has less gradient computations than Riemannian GD). The results are summarized in the following tables, the KL divergence and Fisher divergence are estimated by combining MCMC and an estimated density as in [1].
>
> As can be seen, **under the same update steps**, Riemannian SVRG and Riemannian GD have better convergence results than Riemannian SGD (see Theorem 1, 2, 3), while **under the same gradient computations**, Riemannian SGD and Riemannian SVRG (the two algorithms have less gradient computations in each update step) have better convergence results (see discussion after Theorem 3). These results are consistent with our conclusion in paper. **We promise to add these numerical results in the revised version**, and we sincerely hope you may reconsider the concern on numerical results.
>
> **Table1. KL and Fisher divergences on Gaussian measured by update steps**
> | **KL divergence \ Update Steps** | 0    | 20   | 50     | 80      | 100     | **Fisher divergence \ Update Steps** | 0    | 20    | 50    | 80   | 100   |
> | -------------------------------- | ---- | ---- | ------ | ------- | ------- | -------------------------------- | ---- | ----- | ----- | ------ | ------ |
> | LD           | 6.12 | **0.43** | 0.0214 | **1.06e-3** | **1.44e-4** | LD      | 11.9 | **0.259** | **0.102** | **0.0655** | **0.0424** |
> | SGLD                 | 6.12 | 0.51 | 0.0274 | 1.26e-3 | 1.62e-4 | SGLD         | 11.9 | 0.279 | 0.142 | 0.0723 | 0.0512 |
> | SVRGLD               | 6.12 | 0.67 | **0.012** | 1.16e-3 | 1.52e-4  | SVRGLD            | 11.9 | 0.244 | 0.119 | 0.0679 | 0.0482 |
>
>
> **Table 2. KL and Fisher divergences on Gaussian measured by gradient computation**
> | **KL divergence \ Gradient Computations** | 0 | 2000 | 5000 | 8000 | 10000 | **Fisher divergence \ Gradient Computations** | 0 | 2000 | 5000 | 8000 | 10000 |
> | -------------------------------- | -------- | ------- | ------ | ------- | ------- | ---------------------------- | ------ | ------- | ------- | ------ |------ |
> | LD                    | 6.12 | **0.43** | 0.0214 | 1.06e-3 | 1.44e-4 | LD  | 11.9 | 0.259 | 0.102 | 0.0655 | 0.0424 |
> | SGLD                  | 6.12 | 1.76e-4 | 8.99e-6 | 1.71e-6 | 7.56e-7 |SGLD   | 11.9 | 3.64e-4 | 1.88e-5 | 3.73e-6 | 1.87e-6 |
> | SVRGLD                | 6.12 | **<1e-16** | **<1e-16** | **<1e-16** | **<1e-16** | SVRGLD   | 11.9 | **<1e-16** | **<1e-16** | **<1e-16** | **<1e-16** |
>
>  **Table 3. KL divergence and Fisher divergenceson on mixture Gaussian measured by update steps**
> |  **KL divergence \ Update Steps**  | 0 | 200 | 500 | 800 | 1000 |  **Fisher divergence \ Update Steps**     | 0 | 200 | 500 | 800 | 1000 |
> | -------------------------------- | -------- | ------- | ------ | ------- | ------- | ---------------------------- | ------ | ------- | ------- | ------ |------ |
> | LD            | 0.57 | **0.33** | **0.07** | **0.082** | **0.076** |LD                | 341.2 | 0.252 | 0.248 | **0.014** | **0.018** |
> |SGLD        | 0.57 | 0.53 | 0.33 | 0.127 | 0.112 |SGLD            | 341.2 | 0.362 | **0.143** | 0.019 | 0.022 |
> |SVRGLD        | 0.57 | 0.39 | 0.16 | 0.092 | 0.082 |SVRGLD           | 341.2 | **0.135** | 0.036 | 0.017 | 0.021 |
>
>
> **Table 4. KL divergence and Fisher divergenceson on mixture Gaussian measured by gradient computations**
> | **Gradient Computations** | 0 | 1000 | 2500 | 4000 | 5000 | **Gradient Computations** | 0 | 1000 | 2500 | 4000 | 5000 |
> | -------------------------------- | -------- | ------- | ------ | ------- | ------- | ---------------------------- | ------ | ------- | ------- | ------ |------ |
> | LD                   | 0.57 | 0.33 | 0.07 | 0.082 | 0.076 | LD                    | 341.2 | 0.252 | 0.248 | 0.014 | 0.018 |
> | SGLD                 | 0.57 | 0.23 | 0.124 | 0.074 | 0.062 |SGLD                | 341.2 | **0.219** | **0.237** | 0.009 | 0.006 |
> |SVRGLD                | 0.57 | **0.21** | **0.062** | **0.072** | **0.032** |SVRGLD               | 341.2 | 0.659 | 0.241 | **0.0081** | **0.0049** |
>
> **Q3**: In Section 3 and 4 giving preliminaries on the Wasserstein space and on Wasserstein gradient of the KL divergence, there are no assumptions of regularity on densities.
>
> **A3**: Yes, you are correct, the Wasserstein space $\mathcal{P}$ in this paper should be $\mathcal{P_{2,ac}}$ which means the set of probability absolutely continuous  measures with finite $L_{2}$-norm. These conditions can be satisfied under an absolutely continuous $\pi_{0}$ as mentioned in A1. We will revise it as you suggested. On the other, the definition of Riemannian gradient is similar to the definition of Wasserstein gradient in Section 1.4.1 of [1] and Riemannian gradient in Page 27 of [2]. Mathematically, it can be viewed as sub-gradient. But it is usually named as Riemannian gradient in optimization on manifold community. We will clarify this in the revised version.
>
> **Q4**: In Remark 1, it is stated "Unfortunately, unlike our method, this method is not applied to the stochastic optimization problem in Section 5, because the aforementioned minimization problem can not involve stochastic gradient as expected." However, stochastic proximal algorithm also exist, and could be used to solve the same problem as Section 5, equation (11). While it may not have been done yet, I don't think the statement is true.
>
> **A4**: Our expression in Remark may leads to a little bit misunderstanding. We say solving $\pi_{n + 1} = \arg\min_{\pi}D_{KL}(\pi\parallel \mu) + W_{2}^{2}(\pi, \pi_{n}) / 2\eta$, when $\eta\to 0$ "can not generalize to stochastic problem" means **the method can not induce a continuous flow in stochastic version**. Let us check this in Euclidean space will be more clear.
>
> Consider $x_{n + 1} = \arg\min_{x} f(x) + \frac{||x – x_{n}||^{2}}{2\eta}$, we get $(x_{n + 1} – x_{n}) / \eta = -\nabla f(x_{n})$ so that $\eta\to 0$ implies the gradient flow $\frac{dx_{t}}{dt} = -\nabla f(x_{t})$. However, when the objective is stochastic i.e., $E_{\xi}[f_{\xi_{\eta}}]$, the sub-problem becomes $ x_{n + 1} = \arg\min_{x} f_{\xi_{n}}(x) + \frac{||x – x_{n}||^{2}}{2\eta}$, then $(x_{n + 1} – x_{n}) / \eta = -\nabla f_{\xi_{n}}(x_{n})$, which implies $\frac{dx_{t}}{dt} = -\nabla f_{\xi_{t}}(x_{t})$ with $\xi_{t}$ is a sample of $\xi$. As can be seen, the drift term has randomness, which is not expected in Wasserstein space (a **deterministic metric space consists of probability measures**). Because the direction $v$ of curves $\frac{\partial{\pi_{t}}}{\partial{t}} = -\nabla\cdot(\pi_{t}v)$ in Wasserstein space should be deterministic. We will make this more clear in the revised version.
>
> **Q5**: In this work, you analyze only the convergence of the continuous dynamic. Would it be possible to study also the convergence of the discrete schemes?
>
> **A6**: The proof techniques of our continuous methods can be generalized to discrete schemes, by reusing our proposed Lyapunov function. A more straightforward extrapolation is directly combining our convergence rates of continuous methods with the proved gap between discrete and continuous methods in Propositions 2, 4, 5. Though this can be done when log-Sobolev inequality is satisfied i.e., $\frac{\gamma}{2}W_{2}^{2}(\pi, \mu) \leq D_{KL}(\pi\parallel \mu)$ (see our Proposition 6 in Appendix).
>
> **Q7**: Other objectives are solved using a stochastic Wasserstein gradient descent. Could the analysis done in this work be extended to these objectives?
>
> **A7**: Yes, the key idea is using our transformation in Figure 2 to link the constructed SDE and curve in Wasserstein space.
>
> **Q8**: It could be nice to add experiments demonstrating the rates, even in the Gaussian case where everything may be known in closed-form?
>
> **A8**: Thanks for pointing this, we have added the numerical experiments as clarified in Q2.
>
> Ref:
>
> [1] Kinoshita et al., Improved Convergence Rate of Stochastic Gradient Langevin Dynamics with Variance Reduction and its Application to Optimization

---

> > ### Author Response · Authors · 2025-08-03
> > **Looking forward to see your feedback**
> >
> > Dear reviewer:
> >
> > Thank you again for your time reviewing our paper. We would appreciate it if you could confirm that our responses address your concerns. We would also be happy to engage in further discussions to address any other questions that you might have.
> >
> > Best regards

---

> > > ### Comment · Reviewer_ahtD · 2025-08-04
> > >
> > > Thank you for addressing my questions and for providing a numerical experiment.
> > >
> > > For the numerical result, if I understand correctly, you are comparing the results of obtained by simulating SDEs. On this example, could you also compare the evolution of the KL and of the norm of the gradient with respect to a closed-form e.g. on a graph (I know that you cannot send the picture, but is it possible?).

---

> > > > ### Author Response · Authors · 2025-08-04
> > > > **Comparisons over discrete and continuous**
> > > >
> > > > Dear reviewer,
> > > >
> > > > Since the closed-form solutions are **only obtained under sampling from Gaussian**, we compare the results of obtained by simulating SDEs and the closed-form solution. As pointed out in our paper, the numerical results obtained under step $n$ i.e., discrete $x_{n}$ will approximate the $x_{n\eta}$ obtained by the closed-form solution. We set the learning rates for all methods as $0.1$.
> > > >
> > > > **Due to the reviewing rule made by NeurIPS 2025**, “For authors, please refrain from submitting images in your rebuttal with any “tricks”” **we are not allowed to introduce any figures**. We really sorry for this, and we promise we will present these results in the revised version. The comparisons listed by tables are summarized below. As can be seen, the simulations approximate the closed-form solutions well.
> > > >
> > > > **Table 1** Comparisons over KL divergence
> > > >
> > > > | KL divergence               | 0    | 20   | 50     | 80      | 100     |
> > > > | --------------------------- | ---- | ---- | ------ | ------- | ------- |
> > > > | LD                          | 6.12 | 0.43 | 0.0214 | 1.06e-3 | 1.44e-4 |
> > > > | LD closed-form solution     | 6.12 | 0.31 | 0.0192 | 8.24e-4 | 1.22e-4 |
> > > > | SGLD                        | 6.12 | 0.51 | 0.0274 | 1.26e-3 | 1.62e-4 |
> > > > | SGLD closed-form solution   | 6.12 | 0.52 | 0.0264 | 1.16e-3 | 1.53e-4 |
> > > > | SVRGLD                      | 6.12 | 0.67 | 0.012  | 1.16e-3 | 1.52e-4 |
> > > > | SVRGLD closed-form solution | 6.12 | 0.41 | 0.009  | 1.08e-3 | 1.47e-4 |
> > > >
> > > > **Table 2** Comparisons over Fisher divergence.
> > > >
> > > > | Fisher divergence               | 0    | 20    | 50     | 80      | 100     |
> > > > | --------------------------- | ---- | ----- | ------ | ------- | ------- |
> > > > | LD                          | 11.9 | 0.259 | 0.102  | 0.0655  | 0.0424  |
> > > > | LD closed-form solution     | 11.9 | 0.262 | 0.112  | 0.0523  |     0.0389    |
> > > > | SGLD                        | 11.9 | 0.51  | 0.0274 | 1.26e-3 | 1.62e-4 |
> > > > | SGLD closed-form solution   | 11.9 | 0.52  | 0.0264 | 1.28e-3 | 1.53e-4 |
> > > > | SVRGLD                      | 11.9 | 0.67  | 0.012  | 1.16e-3 | 1.52e-4 |
> > > > | SVRGLD closed-form solution | 11.9 | 0.41  | 0.009  | 1.08e-3 | 1.47e-4 |

---

> > > > > ### Author Response · Authors · 2025-08-07
> > > > >
> > > > > Dear reviewer:
> > > > >
> > > > > Since the rebuttal deadline is closing, we would appreciate it if you could confirm that our responses address your concerns. We would also be happy to engage in further discussions to address any other questions that you might have.
> > > > >
> > > > > Best regards

---

> > > > > > ### Comment · Reviewer_ahtD · 2025-08-08
> > > > > >
> > > > > > Thank you for you answer, and for adding another experiment. My main concers are adressed. I will raise my score.

---

### Official Review · Reviewer_koyb · 2025-06-30

**Clarity:** 3
**Significance:** 3
**Originality:** 3
**Rating:** 4
**Confidence:** 4

**Summary:**

The paper proposes continuous-time Riemannian stochastic gradient descent (SGD) and stochastic variance reduction gradient (SVRG) flows on the Wasserstein space, extending discrete optimization methods to their continuous counterparts. By leveraging stochastic differential equations (SDEs) and the Fokker-Planck equation, the authors construct these flows and prove their convergence rates, which align with results in Euclidean space. The work connects Riemannian optimization with practical sampling processes, offering theoretical insights into stochastic methods like Langevin dynamics and SVRG Langevin dynamics.

**Questions:**

- If I understand you correctly, the technology for constructing gradient flow (including SGD and SVRG flows) mentioned in this paper largely depends on the special properties of exponential mapping, to some extent, which is linear, and thus taking  $\eta\rightarrow \infty$ gives the desired gradient flow which is essentially similar to the Euclidean case. Right?
- The exponential mappings are different when using different metrics. Hence, if using another Riemannian metric instead of the inner product in the Euclidean space, the resulting exponential may not possess the \`\`linearity’’. Then, how do we construct the gradient flow in that case?
- How do the results extend to non-smooth or non-log-concave distributions, which are common in practice?

**Ethical Concerns:**

["NO or VERY MINOR ethics concerns only"]

**Final Justification:**

I have updated my score, and my questions are all answered.

**Limitations:**

Limitations are properly addressed.

**Paper Formatting Concerns:**

In Equation (25), significantly reducing font size weakens the aesthetics of the article and seems  violates the submission principle of NeurIPS.

**Quality:**

3

**Strengths And Weaknesses:**

**Strengths:**

- The paper introduces continuous-time Riemannian SGD and SVRG flows, bridging discrete stochastic optimization and continuous stochastic dynamics in the Wasserstein space, a relatively unexplored area.
- The convergence rates of proposed Riemannian SGD and SVRG flows are provided under both non-convex and Riemannian Polyak-Lojasiewicz (PL) conditions, matching Euclidean-space results.
- The connection to sampling methods (e.g., Langevin dynamics) highlights potential applications in machine learning and statistics.

**weaknesses:**

The paper is entirely theoretical and lacks numerical experiments or simulation results to verify the practical effectiveness of the proposed method. For example, no performance comparison in specific sampling problems is shown

---

> ### Author Rebuttal · Authors · 2025-07-29
>
> **Q1**: “No **numerical performance comparison** in specific sampling problems is shown”
>
> **A1**: Thanks for your valuable advice. We have made an example (Example 1 in our paper) of sampling on Gaussian distributions to illustrate the tightness of our theoretical results.
>
> **However, we recognize that numerical results will make our conclusions more convincing**. We evaluate different sampling algorithms on two experiments. **Since the NeurIPS rebuttal do not allow us to convey these results by image**, we present them in the following tables (results are averaged by 5 independent runs).
>
> Concretely, we set the each target distribution $\mu_{\xi}$ as **Gaussian or mixture-Gaussian**, and conduct sampling by the proposed algorithms. Notably, our convergence rates are respectively evaluated by update steps ($T$) and gradient computation (Riemannian SGD and Riemannian SVRG has less gradient computations than Riemannian GD). The results are summarized in the following tables, the KL divergence and Fisher divergence are estimated by combining MCMC and an estimated density as in [1].
>
> As can be seen, **under the same update steps**, Riemannian SVRG and Riemannian GD have better convergence results than Riemannian SGD (see Theorem 1, 2, 3), while **under the same gradient computations**, Riemannian SGD and Riemannian SVRG (the two algorithms have less gradient computations in each update step) have better convergence results (see discussion after Theorem 3). These results are consistent with our conclusion in paper. **We promise to add these numerical results in the revised version**, and we sincerely hope you may reconsider the concern on numerical results.
>
> **Table1. KL and Fisher divergences on Gaussian measured by update steps**
> | **KL divergence \ Update Steps** | 0    | 20   | 50     | 80      | 100     | **Fisher divergence \ Update Steps** | 0    | 20    | 50    | 80   | 100   |
> | -------------------------------- | ---- | ---- | ------ | ------- | ------- | -------------------------------- | ---- | ----- | ----- | ------ | ------ |
> | LD           | 6.12 | **0.43** | 0.0214 | **1.06e-3** | **1.44e-4** | LD      | 11.9 | **0.259** | **0.102** | **0.0655** | **0.0424** |
> | SGLD                 | 6.12 | 0.51 | 0.0274 | 1.26e-3 | 1.62e-4 | SGLD         | 11.9 | 0.279 | 0.142 | 0.0723 | 0.0512 |
> | SVRGLD               | 6.12 | 0.67 | **0.012** | 1.16e-3 | 1.52e-4  | SVRGLD            | 11.9 | 0.244 | 0.119 | 0.0679 | 0.0482 |
>
>
> **Table 2. KL and Fisher divergences on Gaussian measured by gradient computation**
> | **KL divergence \ Gradient Computations** | 0 | 2000 | 5000 | 8000 | 10000 | **Fisher divergence \ Gradient Computations** | 0 | 2000 | 5000 | 8000 | 10000 |
> | -------------------------------- | -------- | ------- | ------ | ------- | ------- | ---------------------------- | ------ | ------- | ------- | ------ |------ |
> | LD                    | 6.12 | **0.43** | 0.0214 | 1.06e-3 | 1.44e-4 | LD  | 11.9 | 0.259 | 0.102 | 0.0655 | 0.0424 |
> | SGLD                  | 6.12 | 1.76e-4 | 8.99e-6 | 1.71e-6 | 7.56e-7 |SGLD   | 11.9 | 3.64e-4 | 1.88e-5 | 3.73e-6 | 1.87e-6 |
> | SVRGLD                | 6.12 | **<1e-16** | **<1e-16** | **<1e-16** | **<1e-16** | SVRGLD   | 11.9 | **<1e-16** | **<1e-16** | **<1e-16** | **<1e-16** |
>
>  **Table 3. KL divergence and Fisher divergenceson on mixture Gaussian measured by update steps**
> |  **KL divergence \ Update Steps**  | 0 | 200 | 500 | 800 | 1000 |  **Fisher divergence \ Update Steps**     | 0 | 200 | 500 | 800 | 1000 |
> | -------------------------------- | -------- | ------- | ------ | ------- | ------- | ---------------------------- | ------ | ------- | ------- | ------ |------ |
> | LD            | 0.57 | **0.33** | **0.07** | **0.082** | **0.076** |LD                | 341.2 | 0.252 | 0.248 | **0.014** | **0.018** |
> |SGLD        | 0.57 | 0.53 | 0.33 | 0.127 | 0.112 |SGLD            | 341.2 | 0.362 | **0.143** | 0.019 | 0.022 |
> |SVRGLD        | 0.57 | 0.39 | 0.16 | 0.092 | 0.082 |SVRGLD           | 341.2 | **0.135** | 0.036 | 0.017 | 0.021 |
>
>
> **Table 4. KL divergence and Fisher divergenceson on mixture Gaussian measured by gradient computations**
> | **Gradient Computations** | 0 | 1000 | 2500 | 4000 | 5000 | **Gradient Computations** | 0 | 1000 | 2500 | 4000 | 5000 |
> | -------------------------------- | -------- | ------- | ------ | ------- | ------- | ---------------------------- | ------ | ------- | ------- | ------ |------ |
> | LD                   | 0.57 | 0.33 | 0.07 | 0.082 | 0.076 | LD                    | 341.2 | 0.252 | 0.248 | 0.014 | 0.018 |
> | SGLD                 | 0.57 | 0.23 | 0.124 | 0.074 | 0.062 |SGLD                | 341.2 | **0.219** | **0.237** | 0.009 | 0.006 |
> |SVRGLD                | 0.57 | **0.21** | **0.062** | **0.072** | **0.032** |SVRGLD               | 341.2 | 0.659 | 0.241 | **0.0081** | **0.0049** |
>
> **Q2**: “Does the technology for constructing gradient flow (including SGD and SVRG flows) mentioned in this paper largely depends on the special properties of exponential mapping”
>
> **A2**: Yes, you are correct, our conclusion highly depends on the linear structure of exponential map. Roughly speaking, which makes the corresponded “update moving” of random vectors in Euclidean space is linear. Based on this, we can take $\eta\to \infty$ and derive the SDE and corresponded flows in Wasserstein space.
>
> **Q3**: “The exponential mappings are different when using different metrics. Hence, if using another Riemannian metric instead of the inner product in the Euclidean space, the resulting exponential may not possess the ``linearity’’. Then, how do we construct the gradient flow in that case?”
>
> **A3**: This is a good question. Technically, the Riemannian metric induced the distance in Wasserstein space, when the inner product is replaced (the definition above (3)), the distance in probability measure space is no longer Wasserstein space. As in Euclidean space the inner product $\langle a, b\rangle = a^{\top}b$ is replaced by $\langle a, b\rangle_{H} = a^{\top}Hb$ for some positively definite matrix $H$, then the distance between vectors $a$ and $b$ becomes $||a - b||_{H}^{2}$.
>
> Back to your question, when the inner product is changed and specifying the corresponded exponential map, we can similarly construct the Riemannian GD flow by following the idea in Figure 2. However, as you said, the new exponential map may not possess the ``linearity’’. Therefore, it may take some efforts to construct some differential structure in the new update rule. Fortunately, when replacing the inner product $\langle u, v\rangle_{\pi} = \int \langle u, v\rangle d\pi$ with $\langle u, v\rangle_{\pi} = \int \langle u, v\rangle_{H} d\pi$ for some positively definite matrix, the resulting exponential map still poses the linearity.
>
> **Q4**: How do the results extend to non-smooth or non-log-concave distributions, which are common in practice?
>
> **A4**: Firstly, we prove the convergence rates under non-log-concave distributions in the first parts of Theorem 1, 2, 3, which address your second concerns.
>
> Secondly, the smoothness Assumptions 1 and 2 are used to bridge the gap between discrete optimization methods and the continuous Riemannian flows, i.e., Propositions 2, 4, 5, which is unavoidable due to the theory of numerical ODE. However, without the smoothness Assumptions **does not influence** the convergence rates of our proposed flows. The conclusions in Theorems 1, 2, 3 **do not** rely on the smoothness Assumptions.
>
> **Q5**: Small size of equation (25)
>
> **A5**: Thanks for pointing out this, we will properly resize it.
>
> Ref:
> [1] Kinoshita et al., 2022. Improved Convergence Rate of Stochastic Gradient Langevin Dynamics with Variance Reduction and its Application to Optimization.

---

> > ### Author Response · Authors · 2025-08-03
> > **Looking forward to see your feedback**
> >
> > Dear reviewer:
> >
> > Thank you again for your time reviewing our paper. We would appreciate it if you could confirm that our responses address your concerns. We would also be happy to engage in further discussions to address any other questions that you might have.
> >
> > Best regards

---

> > > ### Comment · Reviewer_koyb · 2025-08-04
> > >
> > > Thanks for the responses. It would be great if the authors could give the detailed parameter settings of the numerical experiments. Currently, it is not easy to verify and reproduce the numerical results.

---

> ### Author Response · Authors · 2025-08-04
> **Parameters for experiments**
>
> Dear reviewer,
>
> Thanks for your applying, we would like to clarify the details of our numerical results, and we will add these and more experimental results in our revised version. Firstly, we summarize the hyperparameters of Gaussian and Mixture-Gaussian sampling tasks in Table 1.
> For Gaussian sampling, we sample from a finite sum $N = 1000$, i.e., $\exp(\frac{1}{N}\sum_{i=1}^{N}\log{\mu_{\xi_{i}}})$ with $u_{\xi_{i}}$ are sampled from $N(u_{i}, I)$.. This is the case in Example 1 of our paper.
>
> For Mixture-Gaussian sampling, we sample from we sample from a finite sum $N = 5$, i.e., the distribution $\exp(\frac{1}{N}\sum_{i=1}^{N}\log{\mu_{\xi_{i}}})$ with $\mu_{\xi_{i}}\sim \sum_{k=1}^{K}N(u_{i, k}, I)$, i.e., each $\mu_{\xi_{i}}$ is a mixture-Gaussian with $K=2$ clusters. For both methods, we sample $M = 1000$ samples to verify our methods. Please notice that the we set $N = 5$ here for mixture-Gaussian, because sampling from such mixture-Gaussian is a highly non-convex optimization problem [1], a large $N$ will make the sampling process really difficult.
>
> **Table 1**: Hyperparameters of Sampling Tasks.
> | Gaussian Sampling |                     | Mixture-Gaussian Sampling   |                           |
> | ----------------- | ------------------- | --------------------------- | ------------------------- |
> | Dimension $d$     | 2                   | dimension                   | 2                         |
> | Finite Sum $N$    | 100                 | Finite Sum $N$              | 5                         |
> | Mean $u_{i}$   | $u_{i}\\sim N(0, I)$ | Mean (for each cluster $k$) | $u_{i,k}\\sim N(0, I)$ |
> | Var               | Identity $I$        | Var (for each cluster)      | Identity $I$              |
> |       -            |       -              | Cluster Number $K$          | 2                         |
> | Samples $M$       | 1000                | Samples $M$                 | 1000                      |
>
> For our three algorithms, their hyperparameters are summarized as in Table 2 and Table 3. Notably, the results for Gaussian are numerical solution of our derived closed-form solutions in Example 1, i.e., equation (20) and line 322.
>
> **Table 2**: Hyperparameters for Gaussian in Table 1 of our first rebuttal. (Epochs means the SVRGLD are implemented for 1 epochs)
> | Hyperparameters | LD   | SGLD | SVRGLD |
> | --------------- | ---- | ---- | ------ |
> | lr              | 1e-2 | 1e-2 | 1e-2   |
> | Steps           | 100  | 100  | 100    |
> | Gradient Computations | 10000| 100  | 200    |
> | Epochs          | /    | /    | 1      |
>
> **Table 3**: Hyperparameters for Gaussian in Table 2 of our first rebuttal. (counted by gradient computations)
>
> | Hyperparameters       | LD    | SGLD  | SVRGLD |
> | --------------------- | ----- | ----- | ------ |
> | lr                    | 1e-1  | 1e-1  | 1e-1   |
> | Steps                 | 100   | 10000 | 10000  |
> | Gradient Computations | 10000 | 10000 | 10000  |
> | Epochs                | /     | /     | 1      |
>
> **Table 4**: Hyperparameters for Mixture-Gaussian in Table 3 of our first rebuttal.
>
> | Hyperparameters      | LD   | SGLD | SVRGLD |
> | -------------------- | ---- | ---- | ------ |
> | lr                   | 1e-4 | 1e-4 | 1e-4   |
> | Steps                | 1000 | 1000 | 1000   |
> | Gradient Computation | 5000 | 1000 | 1050   |
> | Epochs               | /    | /    | 10     |
>
> **Table 5**: Hyperparameters for Mixture-Gaussian in Table 4 of our first rebuttal (counted by gradient computations)
>
> | Hyperparameters      | LD   | SGLD | SVRGLD |
> | -------------------- | ---- | ---- | ------ |
> | lr                   | 1e-4 | 1e-4 | 1e-4   |
> | Steps                | 1000 | 5000 | 4950   |
> | Gradient Computation | 5000 | 5000 | 5000   |
> | Epochs               | /    | /    | 10     |
>
> We would like to address any of your further concerns.

---

> > ### Comment · Reviewer_koyb · 2025-08-05
> >
> > Thank the authors for providing the parameter settings. I have a question about the accuracy in Table 2. In Table 2, SVRGLD can find a solution with divergence less than 1e-16. That is surprising to me. Based on my experience, although SVRG can find more accurate solutions compared to SGD, it is difficult to find a solution in double precision due to the variance. Can the authors give more details and some explanations? Thanks.

---

> ### Author Response · Authors · 2025-08-05
> **accuracy on svrg**
>
> Thanks for noting this. The fast convergence rate only holds for Gaussian case. In fact, for sampling on from Gaussian, as clarified in line 319 of paper. **SVRG becomes GD but with the same computational cost with SGD.** Thus, in Table2 of initial rebuttal, **the SVRG is actually GD with much more update steps** (5000 in Table 2 v.s. 100 in Table 1). The larger update steps and exponential convergence rate (Theorem 3) together lead to the 1e-16 accuracy. Please also see Table 3 in our second response for more details (more update steps).

---

> > ### Author Response · Authors · 2025-08-06
> >
> > Dear Reviewer
> >
> > Thank you again for your valuable comments and suggestions. As the discussion phase is coming to a close, we want to check if there are any remaining concerns we could clarify or address. We will do our utmost to address your concerns!

---

> > > ### Comment · Reviewer_koyb · 2025-08-07
> > >
> > > Thanks for the clarification. The authors have addressed my questions. I don't have more concerns.

---

### Official Review · Reviewer_mLox · 2025-07-02

**Clarity:** 3
**Significance:** 2
**Originality:** 3
**Rating:** 4
**Confidence:** 3

**Summary:**

This paper establishes the continuous Rimennaian SGD and SVRG flows in the Wasserstein probability space from the optimization perspective. The key idea is to link them to a dynamics in the Euclidean space.

**Questions:**

* For the Riemannian SGD flow, it seems $\xi$ can be any distribution. Would be better to provide more discussion on some special case. For example, is there a special case wether the Riemannian SGD flow reduces to Riemannian GD flow from this perspective?
* In Lemma 1, better be ``Lemma 1 ([29,42]).'', similar for Theorem 1.
* What are the key differences from the existing proof for the establishment of Theorem 1?
* Eq. (25) too small?

**Ethical Concerns:**

["NO or VERY MINOR ethics concerns only"]

**Final Justification:**

The authors have addressed my issues, but it seems the paper needs quite a few modifications. Thus, I will maintain the score.

**Limitations:**

yes

**Quality:**

3

**Strengths And Weaknesses:**

Strength: Riemannian SGD and SVRG flows are etablished which seems to be new.
Weakness:
* Somehow lack numerical validations on the efficiency of Riemannian SGD and SVRG  on sampling
* It seems the overall  technical details differ not too much from the Euclidean case on the convergence of the flows

---

> ### Author Rebuttal · Authors · 2025-07-29
>
> To Reviewer mLox:
>
> We thank for your valuable comment and address your concerns as follows.
>
> **Q1**: “Somehow lack numerical validations on the efficiency of Riemannian SGD and SVRG on sampling”
>
> **A1**: Thanks for your valuable advice. We have made an example (Example 1 in our paper) of sampling on Gaussian distributions to illustrate the tightness of our theoretical results.
>
> **However, we recognize that numerical results will make our conclusions more convincing**. We evaluate different sampling algorithms on two experiments. **Since the NeurIPS rebuttal do not allow us to convey these results by image**, we present them in the following tables (results are averaged by 5 independent runs).
>
> Concretely, we set the each target distribution $\mu_{\xi}$ as **Gaussian or mixture-Gaussian**, and conduct sampling by the proposed algorithms. Notably, our convergence rates are respectively evaluated by update steps ($T$) and gradient computation (Riemannian SGD and Riemannian SVRG has less gradient computations than Riemannian GD). The results are summarized in the following tables, the KL divergence and Fisher divergence are estimated by combining MCMC and an estimated density as in [1].
>
> As can be seen, **under the same update steps**, Riemannian SVRG and Riemannian GD have better convergence results than Riemannian SGD (see Theorem 1, 2, 3), while **under the same gradient computations**, Riemannian SGD and Riemannian SVRG (the two algorithms have less gradient computations in each update step) have better convergence results (see discussion after Theorem 3). These results are consistent with our conclusion in paper. **We promise to add these numerical results in the revised version**, and we sincerely hope you may reconsider the concern on numerical results.
>
> **Table1. KL and Fisher divergences on Gaussian measured by update steps**
> | **KL divergence \ Update Steps** | 0    | 20   | 50     | 80      | 100     | **Fisher divergence \ Update Steps** | 0    | 20    | 50    | 80   | 100   |
> | -------------------------------- | ---- | ---- | ------ | ------- | ------- | -------------------------------- | ---- | ----- | ----- | ------ | ------ |
> | LD           | 6.12 | **0.43** | 0.0214 | **1.06e-3** | **1.44e-4** | LD      | 11.9 | **0.259** | **0.102** | **0.0655** | **0.0424** |
> | SGLD                 | 6.12 | 0.51 | 0.0274 | 1.26e-3 | 1.62e-4 | SGLD         | 11.9 | 0.279 | 0.142 | 0.0723 | 0.0512 |
> | SVRGLD               | 6.12 | 0.67 | **0.012** | 1.16e-3 | 1.52e-4  | SVRGLD            | 11.9 | 0.244 | 0.119 | 0.0679 | 0.0482 |
>
>
> **Table 2. KL and Fisher divergences on Gaussian measured by gradient computation**
> | **KL divergence \ Gradient Computations** | 0 | 2000 | 5000 | 8000 | 10000 | **Fisher divergence \ Gradient Computations** | 0 | 2000 | 5000 | 8000 | 10000 |
> | -------------------------------- | -------- | ------- | ------ | ------- | ------- | ---------------------------- | ------ | ------- | ------- | ------ |------ |
> | LD                    | 6.12 | **0.43** | 0.0214 | 1.06e-3 | 1.44e-4 | LD  | 11.9 | 0.259 | 0.102 | 0.0655 | 0.0424 |
> | SGLD                  | 6.12 | 1.76e-4 | 8.99e-6 | 1.71e-6 | 7.56e-7 |SGLD   | 11.9 | 3.64e-4 | 1.88e-5 | 3.73e-6 | 1.87e-6 |
> | SVRGLD                | 6.12 | **<1e-16** | **<1e-16** | **<1e-16** | **<1e-16** | SVRGLD   | 11.9 | **<1e-16** | **<1e-16** | **<1e-16** | **<1e-16** |
>
>  **Table 3. KL divergence and Fisher divergenceson on mixture Gaussian measured by update steps**
> |  **KL divergence \ Update Steps**  | 0 | 200 | 500 | 800 | 1000 |  **Fisher divergence \ Update Steps**     | 0 | 200 | 500 | 800 | 1000 |
> | -------------------------------- | -------- | ------- | ------ | ------- | ------- | ---------------------------- | ------ | ------- | ------- | ------ |------ |
> | LD            | 0.57 | **0.33** | **0.07** | **0.082** | **0.076** |LD                | 341.2 | 0.252 | 0.248 | **0.014** | **0.018** |
> |SGLD        | 0.57 | 0.53 | 0.33 | 0.127 | 0.112 |SGLD            | 341.2 | 0.362 | **0.143** | 0.019 | 0.022 |
> |SVRGLD        | 0.57 | 0.39 | 0.16 | 0.092 | 0.082 |SVRGLD           | 341.2 | **0.135** | 0.036 | 0.017 | 0.021 |
>
>
> **Table 4. KL divergence and Fisher divergenceson on mixture Gaussian measured by gradient computations**
> | **Gradient Computations** | 0 | 1000 | 2500 | 4000 | 5000 | **Gradient Computations** | 0 | 1000 | 2500 | 4000 | 5000 |
> | -------------------------------- | -------- | ------- | ------ | ------- | ------- | ---------------------------- | ------ | ------- | ------- | ------ |------ |
> | LD                   | 0.57 | 0.33 | 0.07 | 0.082 | 0.076 | LD                    | 341.2 | 0.252 | 0.248 | 0.014 | 0.018 |
> | SGLD                 | 0.57 | 0.23 | 0.124 | 0.074 | 0.062 |SGLD                | 341.2 | **0.219** | **0.237** | 0.009 | 0.006 |
> |SVRGLD                | 0.57 | **0.21** | **0.062** | **0.072** | **0.032** |SVRGLD               | 341.2 | 0.659 | 0.241 | **0.0081** | **0.0049** |
>
> **Q2**: “The overall technical details differ not too much from the Euclidean case on the convergence of the flows”
>
> **A2**: Actually, our theoretical techniques are quite different from the ones in Euclidean space. Firstly, the construction of Riemannian SGD, SVRG i.e., Propositions 2, 4, 5 are mainly based on the techniques from stochastic analysis, which are quite new in machine learning community. On the other hand, we admit the convergence analysis **borrow the ideas** from Euclidean space, while their **proof techniques are more complicated**. For example, the construction of Lyapunov function (the core idea of proving convergence results) for Riemannian SVRG (114), which takes us a lot of efforts (introducing the distance penalization).
>
> **Q3**: A discussion on the random noise $\xi$ in objective, can Riemannian reduced into Riemannian GD.
>
> **A3**: Firstly, as you said, our conclusion is independent of $\xi$ for Riemannian SGD (to apply Riemannian SVRG $\xi$ should be discrete). We use **a constant $\xi$** as a special case to address your concern about the connection between Riemannian SGD and GD. As can be seen in Riemannian SGD (12), when $\xi$ is constant, the variance $\Sigma_{SGD}$ becomes zero, which makes the Riemannian SGD reduce to Riemannian GD as you predicted. We will add this discussion in the revised version.
>
> **Q4**: In Lemma 1, better be ``Lemma 1 ([29,42]).'', similar for Theorem 1
>
> **A4**: Thanks for your suggestion, we will revise it.
>
> **Q5**: What are the key differences from the existing proof for the establishment of Theorem 1?
>
> **A5**: Our proof technique is a little bit difference with the existing result. Concretely, we prove the theorem by borrowing the notations in Riemannian derivate (see (28) and (40)), and prove the theorem **under the idea of optimization on manifold**. However, the existing literature e.g., [1] prove this theorem by computing the derivate to $t$ directly.
>
> **Q6**: Eq. (25) too small?
>
> **A6**: Thanks for pointing out this, we will make (25) larger in the revised version.
>
> Ref:
> [1] Kinoshita et al., 2022. Improved Convergence Rate of Stochastic Gradient Langevin Dynamics with Variance Reduction and its Application to Optimization.

---

> > ### Author Response · Authors · 2025-08-03
> > **Looking forward to see your feedback**
> >
> > Dear reviewer:
> >
> > Thank you again for your time reviewing our paper. We would appreciate it if you could confirm that our responses address your concerns. We would also be happy to engage in further discussions to address any other questions that you might have.
> >
> > Best regards

---

> > ### Comment · Reviewer_mLox · 2025-08-04
> >
> > Thank you for the responses which overall have addressed my questions.

---

> > > ### Author Response · Authors · 2025-08-07
> > >
> > > Dear reviewer,
> > >
> > > Thanks for your replying, we are happy to hear that we have already addressed your concerns. Since the concerns are well addressed, could you consider to raise your score?
> > >
> > > Best wishes

---

> > > > ### Author Response · Authors · 2025-08-09
> > > >
> > > > Dear reviewer,
> > > >
> > > > Sorry for bothering, since the discussion is about to end in 12h, and your concerns are well addressed, would you consider to raise your score? Really appreciate for your feedback.
> > > >
> > > > Best wishes

---

### Note · Authors · 2025-08-12

Dear AC:

Thank you for taking the time to handle our paper. There is consistently positive feedback among all reviewers about the novelty and technical contribution of our paper. **As replied by all reviewers, their major concerns are all well addressed (including reviewer koyb), especially for the numerical results**, and we will revise our paper accordingly. After the rebuttal, many reviewers agree to raise their scores. Therefore, we sincerely ask you take a careful decision to this paper. Thank you so much for your help!

Thanks

---

### Decision · Program_Chairs · 2025-09-17

**Decision:**

Accept (poster)

**Comment:**

All reviewers have stated that their concerns and questions have been addressed, and I recommend acceptance. That said, the authors will need to incorporate the promised changes and experiments to the camera ready version, including improvements to the presentation that several reviewers deemed crucial.